# Beyond Single-Turn: A Survey on Multi-Turn Interactions with Large Language Models

**Yubo Li**                                    *yubol@andrew.cmu.edu*
**Xiaobin Shen**[†]                             *xiaobins@andrew.cmu.edu*
**Yidi Miao**[†]                                *yidim@andrew.cmu.edu*
**Xueying Ding**[‡]                             *xding2@andrew.cmu.edu*
**Xinyu Yao**[‡]                                *xinyuyao@andrew.cmu.edu*
**Ramayya Krishnan**                            *rk2x@andrew.cmu.edu*
**Rema Padman**                                 *rpadman@andrew.cmu.edu*

*Carnegie Mellon University*

**Reviewed on OpenReview:** *https://openreview.net/forum?id=UYNQXPevpF*

## Abstract

Recent advances in large language models (LLMs) have substantially improved single-turn task performance, yet real-world applications increasingly demand sophisticated multi-turn interactions. This survey provides a structured review of recent progress in evaluating and enhancing multi-turn LLM interactions. Centered on a task-family-oriented taxonomy, spanning (i) instruction following in domains such as mathematics and coding, and (ii) conversational engagement in role-play, healthcare, education, and adversarial jailbreak settings, we systematically examine the challenges of maintaining context, coherence, fairness, and responsiveness across prolonged dialogues. We organize existing benchmarks and datasets into coherent categories reflecting the evolving landscape of multi-turn dialogue evaluation, and review a broad spectrum of enhancement methodologies, including model-centric strategies (in-context learning, supervised fine-tuning, reinforcement learning, and architectural innovations), external integration approaches (memory augmentation, retrieval-based methods, and knowledge graphs), and agent-based techniques for collaborative interaction. Finally, we identify open challenges and promising directions for future research to further improve the robustness and effectiveness of multi-turn LLM interactions. The companion repository is available at `https://github.com/yubol-bobo/Awesome-Multi-Turn-LLMs`.

## 1 Introduction

The advent of Large Language Models (LLMs), exemplified by influential systems such as the GPT series (Radford et al., 2019; Brown et al., 2020; Achiam et al., 2023), PaLM (Chowdhery et al., 2023), and LLaMA (Touvron et al., 2023), has significantly reshaped numerous domains, from education and healthcare to customer service and software engineering. These powerful language models demonstrate remarkable proficiency in generating coherent and contextually relevant responses, achieving substantial gains in performance across various language understanding and generation benchmarks.

However, much of the early progress in both the evaluation and improvements of LLMs has been concentrated in single-turn settings, where models are tested on isolated prompts without considering prior conversational context. While this approach has yielded strong benchmark performance, it underexplores LLMs' broader potential in multi-turn dialogue—the setting that better reflects many real-world interactive uses. Figure 1 makes this distinction explicit by treating single-turn interaction as the one-step special case of a multi-turn trajectory: the single-turn setting has no realized prior history or evolving auxiliary state, whereas

---

†Equal contribution (co-second authors). ‡Equal contribution (co-third authors).

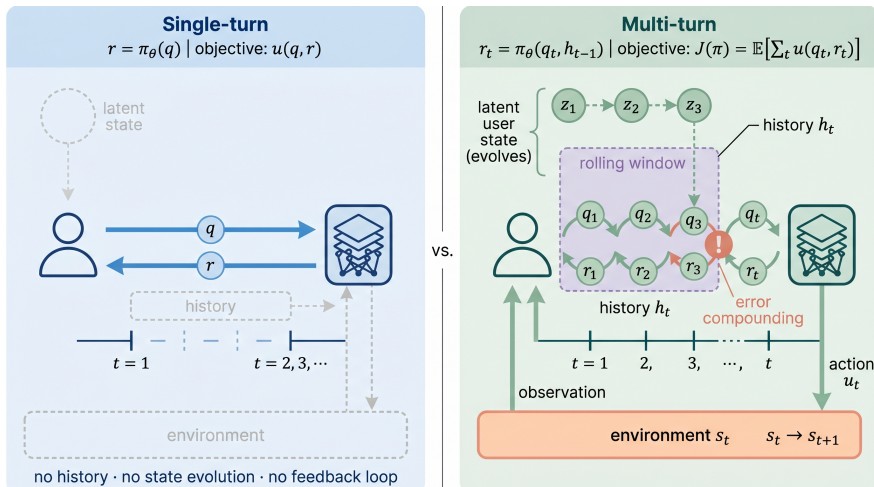

Figure 1: Single-turn vs. multi-turn LLM interaction. The left panel depicts a single-turn exchange as the one-step special case of the multi-turn structure shown on the right, with no realized prior history, auxiliary state, or environmental feedback. The multi-turn panel makes explicit the additional structure introduced by sequential interaction, including temporal unfolding across turns, rolling history $h_t$, evolving latent state $z_t$, environment feedback through $s_t$ and $u_t$, and the possibility of error compounding. See §2 for formal definitions of all notations.

the multi-turn setting exposes the temporal, stateful, and feedback-driven structure that defines sustained interaction. In practice, user needs rarely arise as isolated queries; meaningful and productive interactions typically unfold through sustained, multi-turn exchanges. Effective communication between humans and artificial intelligence inherently demands an understanding of conversational history, nuanced interpretation of previous exchanges, iterative refinement of goals, and adaptive response strategies. Single-turn interaction thus presents a significant limitation, restricting the deployment and utilization of the robust capabilities that LLMs possess.

Recognizing this crucial limitation, substantial research attention has recently shifted toward multi-turn interactions, focusing on enhancing LLM capabilities to sustain context, maintain consistency, handle ambiguity, and dynamically respond across sequential conversational turns. Multi-turn interactions introduce additional layers of complexity, such as managing dialogue coherence, maintaining alignment with user intentions, and addressing issues like cumulative errors, hallucinations, and contextual drift.

This emerging field presents rich opportunities and considerable challenges, prompting a rapidly expanding body of research dedicated to optimizing, evaluating, and deploying LLMs within multi-turn settings. Understanding these multi-turn dynamics and systematically addressing their inherent complexities is essential for improving LLMs' deployment in interactive settings, thereby significantly expanding their applicability and effectiveness in real-world scenarios.

**Scope** This survey focuses on *multi-turn interaction* with LLMs. Our core scope consists of settings in which an LLM participates in sequential interaction and is evaluated on its ability to maintain context, adapt across turns, and achieve task success over a dialogue trajectory. We organize this literature by task family, covering instruction following and conversational engagement in several high-impact domains.

We treat *LLM-based agents* as adjacent rather than central scope. Agentic systems often extend plain multi-turn interaction with tool use, explicit planning, environment manipulation, or multi-agent coordination over richer action spaces. These works are highly relevant when discussing improvement methods, and we therefore review them when they directly inform advances in multi-turn interaction, but we do not aim to survey the agent literature exhaustively or treat it as a core task family of this paper.

We also exclude *multimodal LLMs (MLLMs)* from the main scope of the survey. Although multimodal systems increasingly support multi-turn interaction, they involve substantially different observation spaces, task formulations, and evaluation protocols. We therefore limit our main analysis to non-multimodal settings in order to maintain a coherent and analytically focused scope.

Borderline works are included when they directly inform multi-turn interaction with LLMs, but if a paper's primary contribution lies in agent planning, environment control, or multimodal perception and action, we treat it as adjacent rather than core scope. We further formalize these distinctions in §2.

**Related Surveys**   Several prior surveys are closely related to our topic, but they differ in taxonomy, scope, and coverage emphasis. Foundational instruction-following evaluation papers (Zhou et al., 2023; Zeng et al., 2024b) primarily study single-turn alignment and robustness in static settings. Broader instruction-tuning surveys such as Zhang et al. (2026) substantively cover instruction following and supervised fine-tuning from a methodological perspective, rather than organizing the field through multi-turn task families. Dialogue-oriented surveys such as Yi et al. (2025) and domain reviews in healthcare (Valizadeh & Parde, 2022; Shi et al., 2024c) focus on dialogue systems within particular application areas. The closest prior work, Zhang et al. (2025a), centers on multi-turn capabilities, evaluation methods, and algorithmic improvements from a largely capability-oriented perspective.

In contrast, we focus on *multi-turn interaction* with LLMs and organize the field through a *task-family-oriented* taxonomy spanning instruction following and conversational engagement. This framing places greater emphasis on high-impact application domains such as healthcare and education, while integrating task structure, benchmark resources, improvement methods, and open challenges within a unified application-oriented view. Table 1 summarizes the main differences in taxonomy and in whether each survey substantively covers the content families and cross-cutting axes that anchor this survey, and Appendix B.1 provides broader pointers to adjacent survey literatures.

Table 1: Comparison with closely related surveys along taxonomy and coverage dimensions.

| Survey | Taxonomy | Primary Scope | IF | CE | Impr. | Chal. |
|---|---|---|---|---|---|---|
| Zhang et al., 2026 | Method | Instruction tuning / SFT | ✓ | ✗ | ✓ | ✓ |
| Yi et al., 2025 | Task | Dialogue task | ✗ | ✓ | ✓ | ✓ |
| Valizadeh & Parde, 2022 | Domain | Dialogue task (Healthcare) | ✗ | ✓ | ✓ | ✓ |
| Shi et al., 2024c | Domain | Dialogue task (Medical) | ✗ | ✓ | ✓ | ✓ |
| Zhang et al., 2025a | Capability | Multi-turn capability | ✗ | ✗ | ✓ | ✗ |
| Ours | Task family | Multi-turn task families | ✓ | ✓ | ✓ | ✓ |

*Legend:* ✓= Yes; ✗= No.    *Abbreviations:* IF = instruction-following task family; CE = conversational-engagement task family; Impr. = improvement methods; Chal. = open challenges.

Papers included in prior surveys may therefore be absent from our core taxonomy for several reasons: they may fall outside our scope (e.g., multimodal or embodied settings), lie primarily in agentic planning or environment control, fail to involve genuinely policy-dependent multi-turn interaction, fall outside our temporal coverage emphasis, or be reassigned in our taxonomy to adjacent sections such as improvements or boundary discussions. These boundary decisions follow the scope above and the formal distinctions in §2.

**Review Methodology**   This survey is a task-family-oriented review of *multi-turn interaction* with LLMs. The original manuscript was largely completed in April 2025, and the current version retrospectively documents the corpus-construction process used for that version while extending the survey to newly available papers through April 2026 under the same inclusion, exclusion, and boundary-setting rules. Rather than claiming a fully prospective systematic review, we adopt a PRISMA-ScR-inspired transparency protocol: Appendix A reports the search sources, keyword families, screening stages, inclusion and exclusion criteria, stage-level corpus counts, and the logic by which papers were assigned to the final corpus. This design improves methodological transparency while remaining consistent with the survey's task-family-oriented narrative synthesis.

**Main Contributions**  To address these gaps and facilitate greater research efforts in multi-turn LLM interactions, this survey presents a structured and detailed analysis that explicitly considers practical scenarios and characteristics of multi-turn deployments. We categorize multi-turn interactions by task family, exploring both mixed-topic instruction-following settings (§3.1) and more complex, open-ended conversational-engagement settings (§3.2) across several key domains where LLMs have demonstrated substantial impacts.

Beyond categorization, we contribute by detailing improvement methodologies across three crucial dimensions: (1) Model-Centric Approaches, directly refining and adapting LLMs to effectively handle sequential dialogue dynamics through strategies like in-context learning, supervised fine-tuning, reinforcement learning, and innovative architectures (§4.1); (2) External Integration Approaches, enhancing LLM performance by leveraging external resources such as memory structures, retrieval mechanisms, and knowledge graphs to overcome contextual limitations and maintain factual consistency (§4.2); and (3) Agent-Based Approaches, representing a growing line of work on proactive, iterative agents that interact individually or collaboratively, managing complexity and improving reasoning capabilities in extended interactions (§4.3).

Additionally, we thoroughly discuss open challenges (§5), proposing a clear taxonomy that categorizes these challenges into five major areas: Context Understanding, Complex Reasoning, Adaptation & Learning, Evaluations, and Ethical & Safety Issues. Finally, we summarize key insights, reflect on overarching themes, and provide perspectives on future directions in a dedicated conclusion (§6).

To the best of our knowledge, this survey is the first to organize multi-turn LLM interaction through a task-family-oriented taxonomy that jointly covers instruction-following and conversational-engagement settings while coupling that taxonomy with harmonized benchmark tables, improvement methods, and open challenges across seven application subdomains.

## 2 Background and Problem Formulation

### 2.1 A Sequential View of Multi-Turn Interaction

We model a multi-turn interaction as a trajectory

$$\tau = (q_1, r_1, q_2, r_2, \ldots, q_T, r_T),$$

where $q_t$ denotes the query or input context available to the LLM at turn $t$, and $r_t$ denotes the model's generated response or task-level action. In the non-multimodal settings considered in this survey, $q_t$ may include the current user utterance, retained dialogue history, retrieved evidence, tool outputs, and persistent or summarized memory state. In plain dialogue settings, $q_t$ typically reduces to the current user utterance together with the retained conversational history. Let

$$h_t = (q_1, r_1, \ldots, q_t)$$

be the interaction history observed before generating turn $t$. The LLM then acts as a conditional policy

$$\pi_\theta(r_t \mid h_t, z_t),$$

where $z_t$ denotes optional auxiliary state, such as retrieved evidence, memory records, tool or environment state, or structured task state. Because practical LLMs have finite context windows, long interactions may require context-window management through truncation, summarization, retrieval over prior conversation or external corpora, or persistent memory stores once the full history no longer fits in the prompt.

This formulation is intentionally broad. It covers pure dialogue benchmarks, tutoring and consultation, iterative coding assistance, and non-multimodal tool-mediated interaction. To make the statefulness of the problem explicit, we can view each turn as depending on a latent state $s_t$ that summarizes user goals, discourse state, task progress, relevant external context, and task or safety constraints. After the model produces $r_t$, the interaction evolves according to

$$s_{t+1} \sim P(\cdot \mid s_t, r_t), \qquad q_{t+1} \sim P(\cdot \mid s_{t+1}),$$

so the quality of a turn is determined not only by its local helpfulness, but also by how it shapes future turns. A generic trajectory-level objective can therefore be written as

$$J(\pi_\theta) = \mathbb{E}_{\tau \sim \pi_\theta} \left[ \sum_{t=1}^{T} u_t \right],$$

where $u_t$ can instantiate instruction satisfaction, factual correctness, pedagogical quality, safety, user satisfaction, or downstream task success.

## 2.2 Interactive Tasks vs. Static Dialogue Evaluation

Not all multi-turn benchmarks test the same problem setting. Throughout this survey, we distinguish three related but non-identical regimes.

At a high level, the distinction can be stated in terms of whether the future input is fixed or policy-dependent. In *static* settings, the benchmark provides a fixed collection of histories and targets,

$$\mathcal{D}_{\text{static}} = \{(h_t, y_t)\}_{t=1}^{N},$$

and the model is evaluated by how well it predicts or ranks $y_t$ given $h_t$. In *interactive* settings, by contrast, the next input depends on the model's response,

$$q_{t+1} \sim P(\cdot \mid h_t, r_t),$$

so evaluation concerns the quality of the full trajectory induced by the policy rather than only the quality of isolated next-turn completions. The intuition was introduced in Figure 1; here we sharpen it into the formal distinction between static and interactive evaluation.

**Static dialogue evaluation.** In static multi-turn evaluation, the conversation transcript or the next-turn targets are fixed in advance. The model conditions on a dialogue history, but its response does not causally influence how subsequent turns are generated. Benchmarks in the MT-Bench family (Zheng et al., 2023; Bai et al., 2024) largely fall into this regime: they are multi-turn because turns are sequentially dependent, but the evaluation is still offline.

**Interactive multi-turn tasks.** In interactive settings, the model response changes what information becomes available later. Asking a clarifying question may reveal missing constraints; a tutoring intervention may alter the student's next reply; a debugging suggestion may trigger new compiler feedback. Here the model is evaluated as a conversational policy over trajectories, not merely as a conditional generator over fixed transcripts. Formally, the benchmark no longer consists only of fixed $(h_t, y_t)$ pairs; instead, the model induces a rollout

$$\tau \sim \pi_\theta, \qquad \tau = (q_1, r_1, \ldots, q_T, r_T),$$

and performance is measured by trajectory-level success, cumulative utility, or final-task completion.

**Dynamic environment and agent settings.** Some works go further and let the model plan, call tools, manipulate external environments, or coordinate multiple agents over long horizons (Yao et al., 2023; Liu et al., 2024e). These settings overlap with multi-turn interaction, but their action spaces and environment dynamics are richer than those of plain dialogue. In this survey, we treat them as adjacent rather than core scope: they are highly relevant when discussing improvement methods, but our main task taxonomy centers on non-multimodal multi-turn interaction with LLMs.

This distinction also clarifies our use of the term *interactive task*. In this paper, a task is interactive if turn-level model decisions causally affect later observations, even when the environment is simply another human or a user simulator rather than a fully embodied world.

**Key terms.** We use *multi-turn interaction* to mean that model behavior at turn $t$ depends on earlier turns and can affect later ones, excluding collections of independent single-turn prompts that merely share a topic. An *utterance* is a single conversational contribution from one party, such as one user message or one model reply; a full turn may therefore be described at the level of the current utterance together with the relevant retained history. We use *memory* as an umbrella term for mechanisms that preserve or reconstruct relevant information beyond the model's immediate local generation state, including long-context prompting, summaries, episodic memory modules, external memory stores, and structured state trackers. *In-context learning (ICL)* adapts model behavior at inference time by conditioning on demonstrations, instructions, or explicit state descriptions placed in the prompt, without updating model parameters (Brown et al., 2020; Dong et al., 2024a). In multi-turn settings, this can include exemplar dialogues, state summaries, or graph-based prompts derived from dialogue history. *Retrieval-augmented generation (RAG)* augments generation with externally retrieved evidence (Lewis et al., 2020); in multi-turn settings, retrieval may target both external documents and earlier conversational context rather than only the current user query. Finally, we reserve *interactive task* for settings in which the model's turn-level choices affect what it will observe later, which distinguishes true multi-turn policy problems from offline evaluation on fixed dialogue logs.

## 3   Multi-Turn Interaction Task Families

In this survey, we categorize multi-turn interactions by task families rather than capabilities because real-world multi-turn scenarios inherently require several capabilities to operate together in service of a concrete objective. Although reasoning, memory, contextual understanding, and adaptability are all critical, they rarely function in isolation within extended conversations. Instead, these capabilities interact dynamically as the model tries to satisfy a user's goal over a dialogue trajectory. By focusing on well-defined task families such as multi-turn instruction following and conversational engagement, we better capture the practical complexity of these interactions and provide an intuitive framework for understanding how LLMs behave in real-world multi-turn settings.

Before diving into detailed discussions on multi-turn instruction-following and conversational-engagement task families, we briefly clarify the rationale behind this categorization. We distinguish these families primarily along two dimensions: user-intention clarity and task complexity. Multi-turn instruction-following task families typically involve clear, explicit instructions with well-defined user intentions, so performance is judged mainly by how precisely the model adheres to or successfully executes those instructions. By contrast, multi-turn conversational-engagement task families are often more open-ended: user intentions may initially be unclear, only partially specified, or dynamically evolving over the course of the dialogue. In these settings, the model extends beyond strict instruction compliance and instead acts as an assistant or consultant—for example, a health consultant, teaching assistant, or customer service representative. Such interactions may require proactive information seeking, synthesis across multiple topics, inference over implied user goals, and the use of external knowledge or tools to sustain contextually appropriate responses over time.

At the next level of the taxonomy, we intentionally group the literature using whichever subdivision best matches how the field clusters in practice. In the instruction-following family, the main subfamilies are general-purpose, mathematical, and coding-oriented settings. In the conversational-engagement family, some subgroups are interaction settings (for example, role-play and jailbreak), whereas others are application domains (for example, healthcare and education). We therefore use "task family" as the main organizing term and treat the lower level as a mixture of subfamilies, settings, and domains rather than as a single uniform ontology.

Figure 2 summarizes the task-family taxonomy used in this survey, while fuller section-by-section literature coverage is provided in the appendix (Table 11).

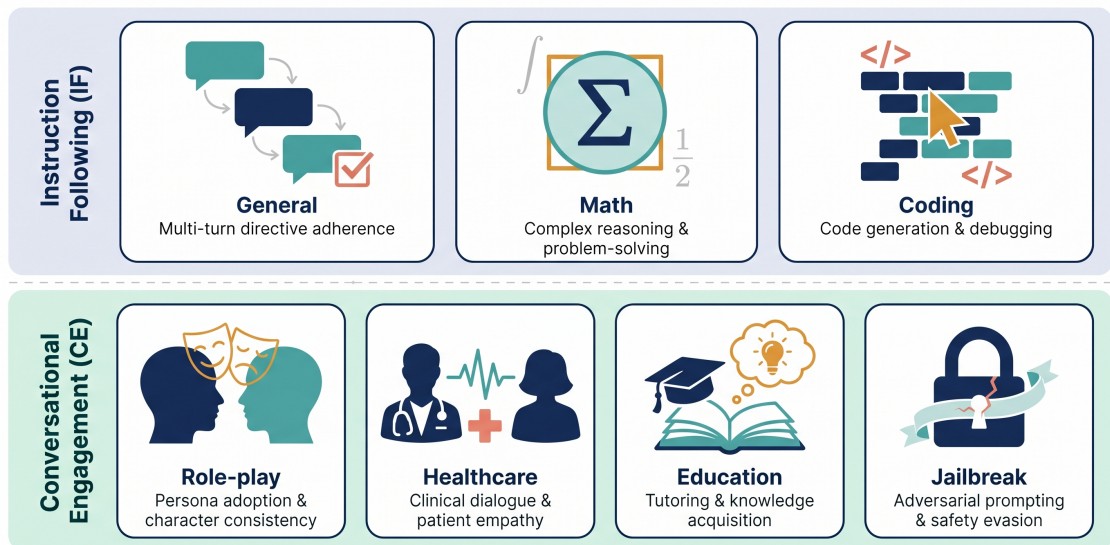

Figure 2: Top-level task-family taxonomy of multi-turn LLM interactions. Fuller section-by-section coverage is provided in the appendix (Table 11).

## 3.1 Instruction-Following Task Family

In this section, we focus on the multi-turn instruction-following task family, highlighting its distinctive characteristics, challenges, and recent benchmark developments. Inspired by the categorization used in MT-Bench (Zheng et al., 2023), we group existing multi-turn instruction-following benchmarks into three primary subfamilies: general-purpose instruction following, mathematics, and coding. As shown in Figure 2, most benchmarks naturally fall into the general category because they cover diverse task types. Nevertheless, specialized benchmarks targeting math and coding interactions have also emerged prominently, demonstrating evaluation dimensions and challenges that differ from general-purpose settings.

### 3.1.1 General-Purpose Instruction Following

Recent research has increasingly focused on evaluating the multi-turn interaction abilities of large language models (LLMs), aiming to capture the complexities of real-world dialogue that single-turn benchmarks, such as BIG-Bench (Srivastava et al., 2023), CSQA (Talmor et al., 2019), MMLU (Hendrycks et al., 2021), and GSM8K (Cobbe et al., 2021), often fail to address.

MT-Bench (Zheng et al., 2023) emerged as one of the first curated benchmarks designed to evaluate multi-turn instruction following capabilities in LLMs. It consists of 80 two-turn dialogues that span eight categories: writing, roleplay, information extraction, reasoning, math, coding, STEM knowledge, and social science. The benchmark evaluates model responses through pairwise comparisons, assessing correctness and helpfulness. A key contribution of MT-Bench (along with AlpacaEval (Li et al., 2023e)) is its systematic study of **LLM-as-a-judge**, where strong LLMs, such as GPT-4, serve as automated evaluators. The study shows that LLM judges achieve over 80% agreement with human evaluators, making them a scalable alternative to human assessments. It also examines potential biases and limitations of LLM judges and suggests mitigation strategies. Since its introduction, LLM-as-a-judge has become a widely adopted evaluation method, shaping benchmarking practices. However, MT-Bench is limited to two-turn dialogues and a relatively small dataset, underscoring the need for more comprehensive benchmarks.

Building on this foundation, MT-Bench++ (Sun et al., 2024) extends the original MT-Bench dataset by incorporating additional follow-up turns per dialogue, resulting in eight-turn interactions. These extended interactions, enriched with carefully crafted ellipsis and anaphora, further challenge models' abilities to maintain context and coherence over prolonged exchanges.

Increasing dialogue length alone was insufficient for comprehensive evaluation. MT-Bench-101 (Bai et al., 2024) introduces a fine-grained taxonomy for multi-turn dialogues. It categorizes interactions into aspects such as perceptivity (context understanding and memory), interactivity (eliciting clarifying questions), and adaptability (reasoning and reflection). With 4,208 turns across 1,388 dialogues and a three-tier ability taxonomy, this benchmark facilitates a detailed assessment of specific interaction skills. Evaluations on 21 prominent LLMs have revealed that even state-of-the-art chat-tuned models exhibit uneven performance across different turns and task types, and that standard alignment techniques (e.g., supervised fine-tuning and RLHF) do not guarantee consistent improvements in multi-turn settings.

MT-Eval (Kwan et al., 2024) takes a different approach by investigating the performance disparities between single-turn and multi-turn interactions. Using 1,170 multi-turn queries derived from human-chat transcripts, MT-Eval categorizes user tasks into four distinct types: follow-up (building upon previous responses), refinement (modifying prior requests), expansion (elaborating on earlier topics), and recollection (retrieving information from earlier turns). The study demonstrates that most LLMs suffer significant performance degradation in multi-turn scenarios, with errors compounding over successive exchanges and the temporal distance from the relevant context further exacerbating the decline. This interaction taxonomy has since been adopted by subsequent research, notably M2Lingual (Maheshwary et al., 2025), which extends multi-turn evaluation into the multilingual domain. M2Lingual applies these interaction categories across 12 diverse languages, revealing concerning cross-lingual brittleness in context retention abilities. The benchmark demonstrates that even advanced models struggle to maintain contextual understanding across language boundaries, with performance deteriorating more severely in non-English interactions, highlighting a critical gap in the multilingual capabilities of current LLMs for sustained dialogues.

M2Lingual is not the only line of work highlighting such cross-lingual challenges. Multi-IF (He et al., 2024), a benchmark for multi-turn and multilingual instructions, introduces a challenging evaluation set for LLMs involving multi-turn, verifiable writing instructions (e.g., style or format requirements) across eight languages. The Multi-IF dataset contains 4,501 dialogues created by expanding a single-turn English benchmark (IFEval (Zhou et al., 2023)) into more challenging multi-turn sequences and translating them into seven additional languages. Concurrently, FairMT-Bench (Fan et al., 2025) focuses explicitly on fairness in multi-turn dialogues. FairMT-Bench is the first comprehensive benchmark designed to evaluate fairness in open-domain multi-turn dialogues for LLMs, formulating a task taxonomy that targets fairness capabilities across three stages: context understanding, user interaction, and instruction trade-offs—challenges discussed further in §5. Based on this taxonomy, the authors constructed the FairMT-1K and FairMT-10K datasets, encompassing two major bias types (stereotype and toxicity) and six bias attributes (age, gender, race, religion, disability, and appearance), covering nearly all bias categories commonly addressed in fairness evaluation.

While most multi-turn benchmarks focus on context retention and reasoning across turns, FB-Bench (Li et al., 2025d) introduces a novel dimension by measuring LLMs' responsiveness to human feedback in multi-turn interaction settings. This Chinese-language benchmark evaluates two crucial aspects of feedback handling: error correction (the ability to fix mistakes when prompted) and response maintenance (the ability to maintain correct responses when challenged). FB-Bench spans diverse task categories, including mathematics, reasoning, coding, text extraction, text error correction, text creation, knowledge Q&A, and text translation. Findings from this benchmark reveal that leading LLMs demonstrate comparable capabilities in error correction across tasks, but their performance varies significantly in response maintenance. Moreover, the research indicates that hinting guidance substantially improves response quality, while exposure to misinformation or fabricated credentials often results in misleading outputs.

Li et al. (2025e) recently draw attention to the challenge of maintaining consistent responses in multi-turn LLM interactions, introducing a framework that assumes models should remain "firm" rather than "fickle" when faced with challenging follow-up prompts. Their work introduces the Position-Weighted Consistency (PWC) score, a novel metric that considers the temporal dimension of dialogue by assigning greater penalties to inconsistencies occurring in earlier interaction rounds. This approach reflects the real-world importance of early stability in establishing user trust. Through experiments with leading models across carefully designed challenge scenarios, the authors demonstrated that even high-performing LLMs can be swayed by various follow-up strategies, including emotional appeals and expert authority claims. To enhance response

stability, they propose the Confidence-Aware Response Generation (CARG) framework, which integrates model confidence signals into the generation process. Empirical results demonstrate that CARG significantly improves response consistency without compromising accuracy, highlighting the necessity of incorporating turn-based considerations into LLM evaluations.

In addition to task-oriented evaluations, a complementary line of work has explored abstract reasoning benchmarks designed to test LLMs' core reasoning capabilities in multi-turn settings. AQA-Bench (Yang et al., 2025c) evaluates LLMs in interactive environments requiring sequential decision-making, such as binary search and graph traversal (DFS, BFS). It emphasizes memory maintenance, procedural adherence, and planning over multiple turns, with both algorithmic and "embodied" (narrative) settings. WILT (Wason Inductive Logic Test) (Banatt et al., 2024) further assesses inductive reasoning by asking models to iteratively discover hidden rules through evidence gathering and hypothesis testing. These benchmarks are task-agnostic and explicitly designed to avoid memorization, targeting core skills such as logical consistency, strategic exploration, and hypothesis refinement.

Besides these abstract reasoning evaluations, other studies have addressed multi-turn interactions from specialized task perspectives. For instance, WebLINX (Lù et al., 2024) tackles the problem of conversational web navigation, where an LLM-based agent must follow user instructions via dialogue to accomplish tasks on real websites. Tasks range from booking tickets to finding information, requiring the agent to understand natural language commands and manipulate a web page accordingly (click links, fill forms, etc.) over multiple turns. MULTITURNINSTRUCT (Han et al., 2025a) proposes a systematic benchmark to probe LLMs' ability to handle sequential, potentially conflicting instructions in a conversation. SysBench (Qin et al., 2025) is a benchmark specifically designed to evaluate how well LLMs adhere to system-level instructions (the hidden directives guiding an AI assistant) in multi-turn interactions. Emphasizing clear and explicit user intentions, we categorize two recent recommendation-system works in this subsection: SAPIENT (Du et al., 2025a) and ECR (Zhang et al., 2024b). SAPIENT (Du et al., 2025a) is a multi-turn conversational recommender system that integrates a planning module for strategic dialogue management, while ECR (Zhang et al., 2024b) introduces an "empathetic" conversational recommender that enhances traditional recommendation dialogues with emotional awareness.

Two recent threads sharpen what "good" multi-turn instruction following means beyond simply answering correctly. Clarify When Necessary (Zhang & Choi, 2025) formalizes ambiguity resolution as deciding when to ask for clarification, what clarifying question to ask, and how to incorporate the new information into the final answer, while Teaching Language Models To Gather Information Proactively (Huang et al., 2025) similarly treats information gathering as a proactive interaction skill rather than a side effect of chat behavior. At the same time, Javaji et al. (2025) show that simply adding extra turns does not uniformly help: iterative prompting improves some tasks, but often fails when later turns do not add genuinely new information or when the model cannot effectively revise its earlier assumptions.

New benchmarks also make the structure of the conversation itself more explicit. StructFlowBench (Li et al., 2025b) models six inter-turn structural relations, TRUEBench (Park et al., 2025) targets multilingual productivity-assistant use cases with explicit and implicit constraints, and EvolIF (Jia et al., 2025a) introduces an evolving benchmark framework that evaluates models until user patience is exhausted rather than stopping at a fixed turn budget. TURNWISE (Graf et al., 2026) isolates the multi-turn performance gap by matching single-turn and multi-turn settings, and IHEval (Zhang et al., 2025f) evaluates whether models respect instruction hierarchies across system messages, user turns, conversation history, and tool outputs. Together, these studies suggest that failures in multi-turn instruction following often stem not only from missing knowledge, but also from weak control over clarification, structure, and instruction priority across turns. Table 2 summarizes the released benchmark and dataset resources discussed in IF-General, organized by resource scale, curation provenance, evaluator setup, and primary evaluation criteria. Across benchmark tables, "/" denotes a field that is not reported or not applicable in the source paper.

Table 2: Released benchmarks and datasets for multi-turn IF-General.

| Benchmark / Dataset | Dataset Size | | Data Curation | | Evaluator | | | | Evaluation Criteria |
|---|---|---|---|---|---|---|---|---|---|
| | Total #Dial. | Avg. #Turns/Dial. | Human-based | LLM-based | Rule-based | Human-as-a-judge[a] | LLM-as-a-judge | Agreement Check[b] | |
| MT-Bench (Zheng et al., 2023) | 80 | 2 | yes | no | no | no | yes | yes | Correctness, Helpfulness |
| MT-Bench++ (Sun et al., 2024) | 80 | 8 | no | yes | no | no | yes | no | Helpfulness, Relevance, Accuracy, Depth, Creativity, Detail |
| MT-Eval (Kwan et al., 2024) | 168 | 6.96 | no | yes | yes | no | yes | no | Quality, Constraint Following, Classification / Recollection Accuracy |
| WEBLINX (Lù et al., 2024) | 2337 | 43 | yes | no | yes | no | no | / | Intent Match, Element IoU, Text F1, Turn / Overall Score |
| AQA-Bench (Yang et al., 2025c) | / | [15,30][c] | / | / | yes | no | no | / | Goal Metric, Policy Metric |
| MT-Bench-101 (Bai et al., 2024) | 1388 | 3.03 | no | yes | no | no | yes | yes | Perceptivity, Adaptability, Interactivity |
| M2Lingual[d] (Maheshwary et al., 2025) | 1140 | 2.48 | no | yes | MT-Bench | | | | same as MT-Bench's |
| SysBench (Qin et al., 2025) | 500 | 5 | yes | yes | yes | no | yes | yes | Constraint Satisfaction Rate, Instruction Satisfaction Rate, Session Stability Rate |
| FB-Bench (Li et al., 2025d) | 591 | 2 | yes | yes | no | no | yes | yes | Error Correction, Response Maintenance |
| WILT (Banatt et al., 2024) | 50 | < 30 | / | / | yes | no | no | / | Hidden-Function Deduction Accuracy |
| Multi-IF (He et al., 2024) | 909 | 3 | no | yes | yes | no | no | / | Instruction- and Conversation-Level IF Accuracy |
| FairMT-Bench (Fan et al., 2025) | 10k | 4 | no | yes | no | no | yes | yes | Direct Bias, Implicit Bias |
| IHEval (Zhang et al., 2025f) | 3538 | 2 | yes | yes | yes | no | no | / | Accuracy under aligned and conflicting instruction hierarchies |
| StructFlowBench (Li et al., 2025b) | 155 | 4.15 | yes | yes | no | yes | yes | yes | Structural-flow Adherence, Multi-turn Constraint Following |
| MULTI TURN INSTRUCT (Han et al., 2025a) | ≈1100 | / | yes | yes | yes | no | no | / | Exact Match, Non-Matching Rate, BLEU |
| MT-Consistency (Li et al., 2025e) | 700 | 9 | yes | no | yes | no | yes | yes | $Acc_{init}$, $Acc_{avg}$, First Sway Round, PWC |
| TRUEBench (Park et al., 2025) | 2400 | / | yes | yes | yes | no | yes | / | Explicit / Implicit Constraint Satisfaction, Overall Pass Rate |
| EvolIF (Jia et al., 2025a) | 150 | /[e] | yes | yes | yes | no | yes | / | Flow-grounded Robustness, Failure Recovery, Process-centric Scores |
| TURNWISE (Graf et al., 2026) | 805[f] | [2,8] | yes | yes | no | no | yes | / | Pairwise Win Rate against Matched Single-turn Settings |

[a] Does not include human evaluation done only for agreement checking with an LLM judge.

[b] If LLM-as-a-judge is used, whether the paper reports agreement checking against human annotators on a subset.

[c] 15 for easy mode, 30 for hard mode.

[d] M2Lingual describes multilingual data generation involving 70+ languages, while the English multi-turn evaluation statistics used here are taken from the paper's MT-Eval-style setting.

[e] The length of the interaction is governed by a patience threshold (which was set to 3 in the paper).

[f] 805 comes from the released AlpacaEval evaluation set, which TURNWISE maps to in a 1:1 manner; the paper does not separately report 805 as a canonical TURNWISE size.

Together, these works underscore ongoing challenges in developing robust multi-turn instruction-following interactions in LLMs, particularly regarding context retention, dialogue coherence, multilingual interactions, fairness considerations, and responsiveness to user feedback. To effectively address the nuanced demands of specialized settings such as mathematics and coding, recent research has introduced targeted benchmarks and approaches. The following subsections explore these subfamilies, highlighting how they deepen our understanding of the broader instruction-following task family.

### 3.1.2 Mathematical Instruction Following

LLMs have demonstrated impressive performance in solving math problems via single-turn prompts, often generating detailed, step-by-step "chain-of-thought" solutions. For example, providing a few worked examples allows a 540B-parameter model to achieve state-of-the-art accuracy on the GSM8K math word problems (Wei et al., 2022). However, complex mathematical tasks frequently require LLMs to engage in multi-turn, instruction-following interactions that involve incremental reasoning, clarification questions, and iterative refinement based on interactive feedback (Liang et al., 2024; Romera-Paredes et al., 2024). In these instruction-following math tasks, users iteratively guide LLMs by providing clarifications, corrections, or additional contextual instructions. Such dynamic interactions not only enhance the models' reasoning processes but also facilitate effective use of external computational tools, enabling the models to perform sophisticated tasks like simulating scenarios or executing complex calculations through code (Romera-Paredes et al., 2024; Lightman et al., 2023). Instruction-following tasks in math also encompass advanced skills such as follow-up questioning, error diagnosis and correction, and educational feedback delivery, thus capturing broader capabilities essential for deploying LLMs in diverse educational and problem-solving contexts.

Several approaches have leveraged multi-turn dialogue with LLMs to enhance mathematical reasoning. For example, Wu et al. (2024b) introduces MathChat-Agent, a framework where an LLM collaborates with a user-proxy agent (responsible for tool use, such as a Python solver) via iterative conversation. This approach solved competition-level math problems more effectively, improving accuracy by roughly 6% over standard single-turn chain-of-thought prompts. Similarly, Keating (2024) employs a multi-agent strategy in which two GPT-4 agents engage in debate-style interactions to reach a solution. This dual-agent zero-shot method achieved about 62.7% accuracy on the MATH benchmark, surpassing single-agent baselines and illustrating the benefits of peer deliberation in reasoning. Moving further, Xiong et al. (2025) propose a method to train LLMs to better combine tool usage with their own reasoning for complex tasks. It gathers trajectory-level feedback from users over multiple turns, to refine problem-solving strategies. The learning process is modeled as a Markov decision process (MDP) and adapts direct preference learning algorithms to multi-turn interactions that include external messages. In practice, the model is trained on multi-turn chats where a user first asks a question and then gives Python outputs in later turns. The accuracy of the method is evaluated on the GSM8K and MATH test sets.

**Benchmarks & Evaluation in Math** Several benchmarks now target multi-turn math dialogues. MathChat-Bench (Liang et al., 2024) extends GSM8K with four new tasks: follow-up QA, problem generation, error correction, and error analysis. The original problems are modified using GPT-4 to meet specific requirements. For instance, in the follow-up QA task, three rounds of dialogue are created: first, GSM8K test problems with ground-truth answers are presented; then GPT-4o generates two follow-up questions; finally, GPT-4 produces the final answers, which are verified or revised by two other LLMs and human annotators. LLMs are evaluated by both accuracy and scores, assigned by GPT-4, which measure their instruction follow-up abilities. Evaluations of state-of-the-art models on MathChat-Bench showed that while they excel at standard one-shot math questions, performance drops sharply on these multi-turn interactions requiring sustained reasoning and dialogue comprehension. To address this gap, the authors also released MathChatSync, a synthetic dialogue dataset for fine-tuning LLMs on conversational math problem solving. Fine-tuning on MathChatSync yielded notable improvements in multi-turn performance.

More broadly, the MINT benchmark (Wang et al., 2024f) evaluates multi-turn tool use and user feedback across domains (including math reasoning). MINT provides an automated framework where the LLM can call a Python interpreter and receive natural language feedback (simulated by GPT-4) in successive turns. Findings from MINT show that multi-turn tool-aided dialogues consistently improve problem-solving success

(each tool use or feedback turn yields additional 1–17% accuracy gains). Interestingly, this evaluation also found that some models fine-tuned only on single-turn instructions (via standard supervised tuning or RLHF) underperform in multi-turn settings, suggesting that multi-turn-specific training is needed to excel in interactive math tasks.

Recent work increasingly evaluates LLMs not merely as math solvers, but as interactive tutors. Beyond Final Answers (Gupta et al., 2025) explicitly measures tutoring quality in math dialogue rather than final-answer accuracy alone, while MRBench (Maurya et al., 2025) expands evaluation toward pedagogical intelligence, tutor actionability, and student-centered support. Intent Matters (Petukhova & Kochmar, 2025) complements these benchmarks with fine-grained pedagogical-intent annotations, making it possible to distinguish whether a model is explaining, probing, scaffolding, or simply giving away the answer. In a different direction, SBSC (Singh et al., 2025) shows that iterative code-supported reasoning can materially improve performance on Olympiad-style mathematics, highlighting the growing overlap between multi-turn mathematical reasoning and tool-mediated interaction. Related tutoring-oriented resources such as KMP-Bench (Shi et al., 2026) are discussed in CE-Education, where we group benchmarks whose primary emphasis is pedagogical intelligence and student support rather than math instruction following per se.

Table 3 summarizes the released benchmark and dataset resources discussed in IF-Math, organized by resource scale, curation provenance, evaluator setup, and primary evaluation criteria.

Table 3: Released benchmarks and datasets for multi-turn IF-Math.

| Benchmark / Dataset | Dataset Size | | Data Curation | | Evaluator | | | | Evaluation Criteria |
|---|---|---|---|---|---|---|---|---|---|
| | Total #Dial. | Avg. #Turns/Dial. | Human-based | LLM-based | Rule-based | Human-as-a-judge[a] | LLM-as-a-judge | Agreement Check[b] | |
| MathDial (Macina et al., 2023) | 2861 | 4.96 | yes | yes | yes | yes | no | / | Success@k, Telling@k, BLEU / BERTScore, Human Tutoring Quality |
| MINT (Wang et al., 2024f) | 586 | < 5 | no | yes | yes | no | no | yes | Success Rate, Evaluation Quality |
| MathChat (Liang et al., 2024) | 5276 | 4 | no | yes | no | no | yes | yes | Follow-Up QA, Error Correction, Error Analysis, Problem Generation |
| MRBench (Maurya et al., 2025) | 192 | 5.04 | yes | yes | no | yes | yes | yes | Tutor-taxonomy coverage, pedagogical ability dimensions |
| SBSC (Singh et al., 2025) | 1467[c] | <15 | no | no | yes | no | no | / | Olympiad performance on AMC / AIME / MathOdyssey under step-wise coding |
| Beyond Final Answers (Gupta et al., 2025) | 150 | / | yes | yes | yes | yes | no | yes | Tutoring quality beyond final-answer correctness |
| Intent Matters (Petukhova & Kochmar, 2025) | 700 | / | yes | yes | no | yes | yes | yes | Fine-grained pedagogical-intent annotation and intent-conditioned tutoring quality |

[a] Does not include human evaluation done only for agreement checking with an LLM judge.
[b] If LLM-as-a-judge is used, whether the paper reports agreement checking against human annotators on a subset.
[c] Computed from the reported test-set composition: 330 AIME, 475 AMC-12, 158 MathOdyssey, and 504 Olympiad-Bench problems.

Notably, several studies have found that LLMs struggle with generalizing to new problems. For example, Liang et al. (2024) show that math-specific LLMs lack adaptive behavior, and their difficulty with generating novel problems highlights their rigidity. Similarly, Macina et al. (2023) report that dialogue tutoring models do not generalize well to unseen math problems. To improve multi-turn math problem solving and generalization, many researchers propose using SFT (Liang et al., 2024; Macina et al., 2023; Wang et al.,

2024f) or RLHF (Wang et al., 2024f; Xiong et al., 2025). Although RLHF can affect LLM-tool interactions and feedback use (Wang et al., 2024f), studies consistently find that fine-tuning with preference data and instructions boosts downstream performance. Macina et al. (2023) demonstrate that small, fine-tuned models perform significantly better, in terms of correctness and equitable tutoring, than prompting a large model like ChatGPT.

### 3.1.3 Coding Instruction Following

LLMs often struggle to produce correct code and perform self-debugging in a single pass, frequently requiring multi-turn or iterative interactions (Zhong et al., 2024; Chen et al., 2025f; Shi et al., 2024d). Instruction-following tasks in coding contexts commonly involve collaborative, iterative interactions, where users provide detailed instructions that the LLM translates into executable code. Through subsequent turns, the LLM iteratively integrates feedback, refines initial solutions, strategically plans modifications, executes and tests submodules, and performs debugging until satisfying the given specifications. Such iterative instruction-following tasks are essential to evaluate LLM performance and robustness in dynamic coding scenarios.

**Benchmarks & Evaluation in Coding** Different frameworks and playgrounds are developed for evaluating code generation quality (Yang et al., 2023; Zheng et al., 2025). InterCode (Yang et al., 2023) introduces a generation pipeline for evaluating LLM coding quality through multi-turn interactions that simulate a real-world coding environment. The InterCode framework requires as input a natural language prompt paired with either an answer or a correct code block. The LLM is evaluated using one of three strategies: "Try-again", where the execution output is fed back as an observation; ReAct, which terminates once the thought chain is complete; or Plan & Solve, which terminates when the plan is fully executed. The experiments involve three programming environments with Spider (Yu et al., 2018), MBPP (Austin et al., 2021), and NL2Bash (Lin et al., 2018) datasets. Zheng et al. (2025) propose a framework for systematically evaluating various prompting techniques for multi-turn code generation by LLMs. Their evaluation is conducted in a zero-shot setting using two competitive coding benchmarks, CodeContests (Li et al., 2022) and TACO (Li et al., 2023d). PyBench (Zhang et al., 2024c) proposes a unified benchmark to evaluate LLM Python coding ability in several categories such as chart analysis and software development. For each task, LLMs interact with a code interpreter for a few turns before making a formal response. The benchmark evaluates success rates and the number of turns required to complete each unit-test-backed task.

Toward effective code generation, CodeGEN (Nijkamp et al., 2023b) and CodeGEN2 (Nijkamp et al., 2023a) are a family of LLMs designed to generate programs from natural language descriptions in multiple turns. The models are evaluated using the Pass Rate metric on their Multi-Turn Programming Benchmark (MTPB), which comprises 115 expert-written problems. Each problem includes multi-step natural language prompts, created by human annotators who decompose the problem into sequential steps. Models are required to generate the complete solution from scratch. Chen et al. (2025e) propose to fine-tune existing LLMs with SFT and DPO. They introduce CodeSteer, a framework that guides LLMs through multiple rounds of inter-action to generate code. In this system, the primary model (TaskLLM) produces responses, both in natural language and in code, while a supervisory agent (CodeSteerLLM) reviews these outputs using symbolic reasoning and self-answer checking to ensure correctness and provide refined guidance. They are fine-tuned and evaluated on subsets of SymBench, which comprises 37 symbolic tasks with adjustable complexity and includes a synthesized dataset of multi-round guidance/generation trajectories and guidance comparison pairs. OpenCodeInterpreter (Zheng et al., 2024b) proposes to fine-tune LLMs with CodeFeedback, a dataset of challenging LeetCode questions, incorporating multi-turn execution feedback (with code interpreters) and human dialogues (synthesized by GPT-4). CodeAct (Wang et al., 2024e) collects an instruction-tuning dataset, CodeActInstruct, which contains 7,000 multi-turn interactions. PyInstruct (Zhang et al., 2024c) is used in PyBench for continuous pretraining and fine-tuning. Zheng et al. (2025) explore a wide range of prompting strategies for effective code generation, focusing on automatic re-prompting over multiple runs.

SQL generation is a subcategory of coding generation that focuses more on data acquisition through large-scale data warehouses. Two benchmarks for SQL generation with multi-turn LLMs are identified during the literature search. MMSQL (Guo et al., 2025b) focuses on text-to-SQL generation tasks and introduces a Multi-type and Multi-turn text-to-SQL test suite, which is a comprehensive benchmark engineered to

evaluate the proficiency of LLMs in handling multi-turn text-to-SQL tasks across diverse question types. MMSQL contains a multi-agent framework anchored by a core Question Detector and Question Decomposer tasked with identifying question types and determining appropriate answering strategies. The framework includes two supportive agents: the Schema Selector, which identifies and provides the essential subset of a database schema, and the SQL Refiner, which is dedicated to refining SQL queries. EHRAgent (Shi et al., 2024a) demonstrates a specific application of SQL generation in healthcare settings, which speeds up the extraction and interaction of clinician information within electronic health record (EHR) systems. EHRAgent (Shi et al., 2024a) translates EHR question-answering into a tool-use planning process, which integrates query-specific medical information and formulates executable code plans through multi-turn dialogues. The model is evaluated based on its ability to reason across multiple tables and generate accurate, actionable insights from complex EHR data.

For educational applications, TreeInstruct (Kargupta et al., 2024) transforms LLMs into Socratic instructor agents that guide users in debugging and writing better code. It operates via two roles: an instructor that generates tree-structured sequential questions, and a verifier that identifies tasks to help students understand, assess, and correct their code. To evaluate TreeInstruct, the authors introduce MULTIDEBUG, a dataset derived from LeetCode problems with expert-injected syntactic and conceptual bugs, assessed through both qualitative measures (relevance, indirectness, logical flow) and quantitative metrics (success rate).

Recent coding work makes the conversational nature of code generation far more explicit. ClarifyGPT (Mu et al., 2023) treats intention clarification as a first-class step for underspecified programming requests, showing that asking targeted follow-up questions can outperform direct generation. When Benchmarks Talk (Pan et al., 2025) and CONVCODEWORLD (Han et al., 2025b) argue that static pass@k evaluations miss the interactive reality of programming, and instead benchmark how models respond to iterative feedback in controlled conversational environments. CodeFlowBench (Wang et al., 2025a) and MultiCodeIF (Duan et al., 2025) further stress long-horizon refinement and fine-grained multi-turn instruction following, Rawal et al.'s MT-Sec (Rawal et al., 2025) jointly evaluates correctness and security under iterative coding, and DySQLBench (Sun et al., 2025) reframes text-to-SQL as a dynamic database-exploration task rather than a single-shot translation problem.

Table 4 summarizes the released benchmark and dataset resources discussed in IF-Coding, organized by resource scale, curation provenance, evaluator setup, and primary evaluation criteria.

Table 4: Released benchmarks and datasets for multi-turn IF-Coding.

| Benchmark / Dataset | Dataset Size | | Data Curation | | Evaluator | | | | Evaluation Criteria |
|---|---|---|---|---|---|---|---|---|---|
| | Total #Dial. | Avg. #Turns/Dial. | Human-based | LLM-based | Rule-based | Human-as-a-judge[a] | LLM-as-a-judge | Agreement Check[b] | |
| MTPB (Nijkamp et al., 2023b) | 115 | / | no | no | yes | no | no | / | Pass Rate, Average Pass Rate, Perplexity |
| InterCode (Yang et al., 2023) | 1351[c] | 10 | no | yes | yes | no | no | / | Success Rate, Turns, Error Rate |
| Code-Feedback (Zheng et al., 2024b) | 68k | / | no | yes | yes | no | no | / | Pass@1, Benchmark-Average Gains under Interactive Refinement |
| MULTI-DEBUG (Kargupta et al., 2024) | 150 | / | yes | no | yes | yes | no | / | Success, Relevance, Indirectness, Logic, Average Turns |
| PyBench (Zhang et al., 2024c) | 143 | <10 | yes | yes | yes | no | yes | yes | Pass Rate, Average Turns, Unit-Test and LLM-based outcomes |

*Table 4 continued*

| Benchmark / Dataset | Dataset Size | | Data Curation | | Evaluator | | | | Evaluation Criteria |
| | Total #Dial. | Avg. #Turns/Dial. | Human-based | LLM-based | Rule-based | Human-as-a-judge[a] | LLM-as-a-judge | Agreement Check[b] | |
|---|---|---|---|---|---|---|---|---|---|
| MMSQL (Guo et al., 2025b) | 6493 | 6 | no | yes | yes | no | yes | yes | Exact Matching, Execution Accuracy, Dual Assessment of Question Type Detection, Response Quality Score |
| When Benchmarks Talk (Pan et al., 2025) | 545 | / | no | yes | yes | no | no | / | Interactive Pass Rate, Feedback-Following Quality, Ranking Shifts across APPS / LCB / ClassEval |
| CONVCODEWORLD (Han et al., 2025b) | 1140 | <10 | no | yes | yes | no | no | / | Conversational Coding Success in Reproducible Environments |
| CodeFlowBench (Wang et al., 2025a) | 5328[d] | 2.08[e] | yes | yes | yes | no | no | no | Complex Code-generation Quality under Iterative Refinement |
| MultiCodeIF (Duan et al., 2025) | 2021 | 5 | yes | yes | yes | no | no | / | Fine-grained Code Instruction Following with Multi-turn Feedback |
| MT-Sec (Rawal et al., 2025) | 2376 | 3 | yes | yes | yes | no | no | / | Correctness and Security under Multi-turn Code Generation |
| DySQLBench (Sun et al., 2025) | 1072 | <30 | yes | yes | yes | no | yes | yes | Dynamic Multi-turn Text-to-SQL and Database Exploration Quality |

[a] Does not include human evaluation done only for agreement checking with an LLM judge.

[b] If LLM-as-a-judge is used, whether the paper reports agreement checking against human annotators on a subset.

[c] Computed as 200 NL2Bash + 1034 Spider + 117 MBPP task instances across InterCode-Bash, InterCode-SQL, and InterCode-Python.

[d] It consists of 5,258 samples from CodeFlowBench-Comp and 70 high-quality multi-turn problems from CodeFlowBench-Repo. The 60 added single-turn problems are excluded.

[e] Estimated from the 1000 samples, aligned with the 2.1-2.2 in the ablation study of the paper.

Several compelling research questions recur in efforts to adapt LLMs for reliable code generation in multi-turn settings. First, when explicit cues are unavailable, how should an LLM decide between purely textual reasoning and executing programmatic actions to reach a solution (Chen et al., 2025f)? Second, which iterative interaction protocols best support incremental refinement, enabling the model to diagnose errors, incorporate feedback, and converge on a final answer (Yang et al., 2023; Zheng et al., 2025; Nijkamp et al., 2023b;a; Wang et al., 2024e)? Third, how should we evaluate coding performance across programming languages, task types, and application domains in a way that is both comparable and representative of real-world use (Yang et al., 2023; Zhang et al., 2024c)?

### 3.1.4 Discussions

Based on our analysis of the IF benchmark tables and the papers discussed above, several noteworthy trends emerge in the evolution of multi-turn instruction-following benchmarks and evaluation methodologies.

**Dataset Evolution** We observe a clear trajectory toward larger and more comprehensive datasets, with benchmark sizes expanding from just 80 examples in MT-Bench to over 1,000 in newer benchmarks like MT-Bench-101, M2Lingual, and FairMT-Bench. This growth reflects a recognition that robust evaluation requires broader coverage of interaction patterns and abilities. Simultaneously, data curation methodolo-

gies have evolved from primarily human-generated content toward automated and LLM-assisted generation processes, addressing scalability challenges while maintaining quality. Despite this expansion, current benchmarks predominantly limit conversations to 10 or fewer turns, leaving the domain of extended multi-turn interactions (dozens or hundreds of turns) largely unexplored.

**Evaluation Methodologies** The evaluation landscape has diversified considerably, transitioning from relatively simple rule-based metrics toward more nuanced and fine-grained assessment frameworks. This evolution parallels the increasing sophistication of LLM capabilities and application scenarios. LLM-as-a-judge approaches have emerged as particularly promising, offering cost-effective evaluation solutions that can scale with the growing complexity of benchmarks. MT-Bench (Zheng et al., 2023) pioneered this approach while also acknowledging inherent biases, and subsequent studies have further clarified its limitations.

Recent studies highlight several concerning limitations: Preference Leakage, as demonstrated by Li et al. (2026b), shows bias toward responses from models sharing architectural or training lineage with the judge model, compromising evaluation fairness. Contextual Sensitivity, revealed by Xu et al. (2025), manifests as performance degradation when evaluating outputs dependent on external context, such as retrieval-augmented generation, where even state-of-the-art judges struggle with consistency. Reference Dependence is another issue, as evaluations often exhibit brittleness when reference solutions are unavailable or when multiple valid approaches exist, a common scenario in open-ended multi-turn interactions.

These challenges underscore the continued importance of human evaluation validation. Human–AI agreement checks are essential for establishing the reliability of automated evaluation methods, yet relatively few benchmarks incorporate substantial human agreement verification—a critical gap that risks allowing biases and inconsistencies to persist undetected.

## 3.2 Conversational Engagement

As shown in Figure 2, the conversational-engagement side of our taxonomy covers role-play, healthcare, education, and jailbreak settings; before turning to these domain-specific branches, we first review several general CE-overview benchmarks that established how to define and measure sustained conversational performance. The conversational-engagement task family mixes interaction settings (such as role-play and jailbreak) with application domains (such as healthcare and education), because the literature is organized in practice by both deployment context and interaction style rather than by one uniform subtyping scheme. One early effort is the ABC-Eval framework by Finch et al. (2023), which introduced a dimensional human evaluation scheme for open-domain chat. ABC-Eval defines 16 fine-grained conversational behavior categories (spanning aspects like factual accuracy, consistency, relevance, and empathy) and uses these as binary turn-level labels to quantify dialogue quality. Although this benchmark relies on labor-intensive human evaluations, its initial findings underscore the inherent challenges in assessing iterative interactions, thereby motivating the development of more scalable and nuanced evaluation frameworks.

Shortly afterward, Duan et al. (2024) introduce the BotChat evaluation paradigm, which reduces reliance on costly human judges by leveraging LLMs for both conversation generation and evaluation. In BotChat, models are prompted to extend real-world dialogue seeds (ChatSEED prompts) into extended multi-turn conversations, subsequently evaluated by top-tier LLMs (e.g., GPT-4) serving as automated judges. Notably, GPT-4-generated dialogues were found to be nearly indistinguishable from human conversations, successfully fooling discriminator models, whereas other contemporary LLMs exhibited shortcomings in instruction adherence and conciseness, underscoring specific challenges in maintaining human-like coherence across multi-turn dialogues.

Most recently, Deshpande et al. (2025) introduce MultiChallenge, a benchmark designed to rigorously evaluate conversational persistence and context management in frontier LLMs. MultiChallenge encompasses four realistic scenarios—long-term instruction retention, implicit information recall, iterative revision, and consistent non-sycophantic responses—each demanding the simultaneous exercise of instruction following, context tracking, and reasoning.

Recent work has also started auditing the evaluation process itself. DialogBench (Ou et al., 2024) broadens open-domain dialogue assessment toward human-likeness, conversational naturalness, and sustained inter-

action quality, while SimulatorArena (Dou et al., 2025) explicitly asks whether user simulators are reliable stand-ins for humans when evaluating multi-turn assistants. Taken together, these studies push the field beyond measuring model behavior alone: they also test whether current evaluation pipelines faithfully capture real conversational performance.

Building upon these frameworks, recent work explores conversational engagement within specialized real-world contexts, including immersive role-play, healthcare consultations, educational interactions, and adversarial jailbreak scenarios. Extending evaluation into these practical domains provides deeper insight into several high-impact applications of conversational capabilities in modern LLMs.

We intentionally do not impose a fully uniform internal template across these CE subsections. Role-play is organized primarily by technical pathways because it functions as a transferable persona/profile foundation that cuts across many downstream applications, whereas healthcare, education, and jailbreak are organized by application setting because their main distinctions arise from domain-specific goals, risks, and evaluation protocols.

Table 5 summarizes the released benchmark and dataset resources discussed in CE-overview, organized by resource scale, curation provenance, evaluator setup, and primary evaluation criteria. As in the IF tables, "/" denotes a field that is not reported or not applicable in the source paper.

Table 5: Released benchmarks and datasets for multi-turn CE-overview.

| Benchmark / Dataset | Dataset Size | | Data Curation | | Evaluator | | | | Evaluation Criteria |
|---|---|---|---|---|---|---|---|---|---|
| | Total #Dial. | Avg. #Turns/Dial. | Human-based | LLM-based | Rule-based | Human-as-a-judge[a] | LLM-as-a-judge | Agreement Check[b] | |
| ABC-Eval (Finch et al., 2023) | 400 | 30.3 | yes | no | no | yes | no | / | Consistency, Emotion, Understanding, Engagingness, Grammaticality, Informativeness, Quality, Proactivity, Relevance |
| BotChat (Duan et al., 2024) | 547 | 16 | yes | yes | no | no | yes | yes | UniEval, PairEval, GTEval, judge-human consistency |
| DialogBench (Ou et al., 2024) | 9811 | 7.63[c] | no | yes | yes | no | no | / | Human-likeness task performance, bilingual comparison, dialogue capability probing |
| MultiChallenge (Deshpande et al., 2025) | 273 | 5 | yes | yes | no | yes | yes | yes | Human Accuracy, rubric-based auto-eval, human-auto alignment |
| SimulatorArena (Dou et al., 2025) | 909 | 7.4 | yes | yes | no | yes | yes | yes | Behavioral similarity, Rating alignment to humans |

[a] Does not include human evaluation done only for agreement checking with an LLM judge.
[b] If LLM-as-a-judge is used, whether the paper reports agreement checking against human annotators on a subset.
[c] Derived from task-level average turns in Table 1; not explicitly reported as a single overall value.

### 3.2.1 Conversational Engagement in Role-Play

Role-play significantly enhances conversational engagement by immersing users in specific scenarios, making interactions feel authentic and contextually relevant. Incorporating explicit roles into dialogue systems encourages users to perceive the interactions as genuine, thereby increasing engagement and satisfaction. Within conversational AI, there is an emerging research domain specifically dedicated to role-play, which aims to create realistic and persona-consistent interactions through LLMs.

Early persona-grounded dialogue systems aimed at maintaining consistent character personas over multi-turn conversations using architectures like memory networks and transformers. Significant contributions include

the persona embeddings introduced by Li et al. (2016), the personalized memory networks developed by Kottur et al. (2017), and notably, the PersonaChat dataset and memory-based models from Zhang et al. (2018), which established foundational benchmarks for persona consistency. These initial works primarily trained models from scratch, facing challenges in sustained persona adherence across interactions. The recent survey by Chen et al. (2024h) extensively covers role-play before and after the advent of LLMs. Within the scope of this survey, we focus specifically on role-play interactions under multi-turn LLM settings.

**In-Context Learning** Early LLMs demonstrated an ability to impersonate user-defined personas through prompting alone. Users could supply descriptions like "You are a wise old wizard..." in a system or context prompt, and models like GPT-3 would attempt to respond "in character." PersonaLLM (Jiang et al., 2024a) introduces a benchmark for personalization and highlights that simply prefixing instructions with high-level persona descriptions yields limited diversity; instead, it proposes simulating nuanced user preferences via prompt-based reward models. It shows that prompting can move beyond trivial traits to tailor outputs to idiosyncratic user needs. Similarly, CharacterChat (Tu et al., 2023) uses role-playing prompts with behavior presets and dynamic memory to maintain a character's persona over long conversations. It constructed an MBTI-based persona bank and prompted ChatGPT to produce dialogues between a "seeker" and a compatible "supporter," injecting preset behavioral tendencies and retrieving context-specific memory each turn to keep interactions coherent.

Beyond persona style, prompting has been used to improve reasoning. Role-play prompting (Kong et al., 2024a) shows that instructing an LLM to "pretend to be" a domain expert can implicitly trigger step-by-step reasoning. In zero-shot settings across 12 tasks, prompting models with a role (e.g. "You are an excellent math teacher...") led to significantly higher accuracy than a vanilla prompt. The method uses a two-stage prompting framework: first having the model generate an "immersive" backstory or persona acknowledgement, then using that along with the query. These studies collectively demonstrate prompting-based role-play as a powerful lever: it can induce consistent persona adherence (persona and behavior presets) and even boost cognitive performance (reasoning via expert roles) without any parameter updates.

**Supervised Fine-Tuning** To achieve more robust in-character behavior, researchers introduced instruction tuning and fine-tuning with role-play data. PIPPA (Gosling et al., 2023) releases a partially synthetic corpus of over 1 million role-play messages, crowdsourced from an online community of role-play enthusiasts. By fine-tuning on PIPPA's diverse persona-conditioned conversations, small LLMs dramatically improved at staying in character, underscoring that sheer volume and diversity of persona-rich dialogues can teach consistent role-play behavior. UltraChat (Ding et al., 2023) constructs 1.5M multi-turn dialogues via self-chat with GPT-4, covering broad topics and user types. Fine-tuning LLaMA on this yielded UltraLLaMA, which surpassed previous open models in general conversation quality (including user engagement and coherence). Other data efforts target specific role-play domains: PRODIGy (Occhipinti et al., 2024) builds a dialogue dataset from movie scripts aligned with detailed character profiles (biographies, personality traits). Fine-tuning models on these profile-grounded movie dialogues significantly improved consistency when emulating those characters—for example, including a character's backstory and speaking style led to higher human preference for in-character responses.

A parallel direction creates models specialized for particular characters or customizable personas. ChatHaruhi (Li et al., 2023b) focuses on anime characters, compiling 54k dialogues for 32 characters by combining original script lines with simulated conversations. The model fine-tuned on this data, augmented with a "memory" of past events for each character, was able to "revive" characters like Haruhi Suzumiya, accurately quoting lore and personality in new interactions. In another example, CharacterGLM (Zhou et al., 2024a) builds on the Chinese GLM model to allow explicit profile injection for any character: it fine-tuned ChatGLM variants on a corpus of dialogues with richly annotated character profiles (covering identity, style, relationships) and achieved state-of-the-art human-likeness and consistency for customized personas. RoleCraft-GLM (Tao et al., 2024) further extends this concept by crafting original non-celebrity personas with emotional depth and fine-tuning a model on dialogues involving those characters. This yielded more nuanced emotional consistency, validating that meticulous character development during fine-tuning yields agents that are engaging and lifelike in their persona.

Recent work also explores fine-tuning strategies for role consistency. Ditto (Lu et al., 2024) introduces a self-alignment pipeline in which the model generates its own role-play dialogues for 4,000 distinct characters and trains on them. By leveraging the model's internal knowledge to produce training data—with feedback to ensure character distinctiveness—Ditto achieves strong persona fidelity across a wide range of roles, outperforming fine-tuned baselines on a role-play benchmark. Notably, Ditto frames role-play generation as a reading-comprehension task to avoid style collapse, distinguishing it from UltraChat's broader synthetic data approach. CharacterLLM (Shao et al., 2023) follows a complementary pipeline: for each target character—particularly historical figures—Wikipedia biographies and documents are used to prompt an LLM to produce character-specific dialogues, and fine-tuning on this synthetic data enables a single agent to robustly portray many roles, effectively transferring factual knowledge into conversational skill. Together, these works reflect a broader trend in role-play fine-tuning from large-scale data harvesting toward increasingly sophisticated data generation and specialization strategies.

**Personalization and Rapid Adaptation**  Even with instruction tuning, a given model has limits in the number of distinct characters or styles it can perfectly emulate. Thus, a key theme is personalization: adapting an LLM to a new persona with minimal data or effort. Recent research has explored parameter-efficient tuning modules that allow rapid persona swapping without retraining the entire model. PersonaPKT (Han et al., 2023) proposes representing each persona as a continuous embedding vector that can be learned from a small set of that user's dialogues. By keeping the pre-trained model fixed and only training a tiny persona-specific vector (less than 0.1% of parameters), PersonaPKT efficiently imbues the model with that user's speaking style and preferences. In a related vein, PPlug (Liu et al., 2025c) introduced a plug-and-play user encoder that on-the-fly computes a user embedding from their conversation history. Instead of a static learned vector, it employs a small model to read a user's past messages and output a "personal embedding" summarizing their quirks and facts. This embedding is then prepended to the LLM's input to personalize the response. Such design lets the LLM be dynamically personalized each turn based on context, and experiments showed significant gains in personalization metrics across tasks.

As LLMs are used to power multiple characters simultaneously, it becomes important to switch personas quickly and even maintain many personas at once. Neeko (Yu et al., 2024b) addresses this with a dynamic LoRA (Low-Rank Adapter) framework for multi-character role-play. It pre-trains a separate LoRA module for each character and uses a gating network to activate the appropriate one based on the dialogue context. This incremental ability means an unlimited number of personas can gradually accrue, each encapsulated in a plug-in adapter. An agent can seamlessly swap roles by toggling adapters, which showed superior consistency when one model needed to portray many characters in a group chat. Another challenge is preserving persona over multiple dialogue rounds. Standard fine-tuning often splits dialogues into independent turns, which can break character memory. MIDI-Tuning (Wang et al., 2024c) proposes to explicitly model the user and system roles with separate adapters and a round-level state. In their framework, an LLM-based agent is trained by alternating between a "user adapter" (processing user utterances) and an "agent adapter" (generating responses), carrying hidden state forward through turns.

**Reinforcement Learning**  While supervised learning can teach a model a persona, it doesn't explicitly punish lapses. Reinforcement learning (RL) provides a way to directly optimize consistency and other long-horizon behaviors. Shea & Yu (2023) demonstrate this by applying offline RL to a dialogue model for persona consistency. They took an existing high-quality chatbot and defined a reward that penalizes persona breaks (e.g., contradicting provided profile or previous statements) and rewards in-character responses. Rather than interact with humans live, they performed RL on a static dataset of conversations—adjusting the model's policy to maximize the persona consistency reward while leveraging off-policy data. The result was a chatbot that, in human evaluation, more reliably adhered to its given persona description and avoided contradictions compared to its purely supervised counterpart. This work bridged the gap between static fine-tuning and RLHF: it shows one can inexpensively refine a model to be more in-character by offline RL on existing dialogues, getting some benefits of RL (direct control of behavior) without an expensive online loop.

Another aspect of multi-turn role-play that benefits from RL-like thinking is maintaining long-term coherence. COMEDY (Chen et al., 2025d) approaches this via a compressive memory mechanism that can be seen as the model "reinforcing" important memory content over a conversation. Instead of a traditional retrieval

pipeline, COMEDY has the LLM periodically summarize and compress the dialogue history (including user persona hints and past events) into a concise memo, which is fed back into itself for future responses. This one-model architecture learns through supervised fine-tuning to generate useful summaries (e.g., remembering the user's preferences or the agent's own backstory) and to consult them when answering. While not an RL algorithm, COMEDY's design implicitly optimizes a long-term reward: the compressed memory serves to avoid contradictions and boring repetition, much like an RL agent maximizing a reward for user engagement would learn to recall relevant facts.

**Benchmarks & Evaluation**   As role-play agents become more advanced, evaluating their effectiveness requires moving beyond standard metrics like BLEU or response fluency. Instead, the field has introduced a suite of specialized benchmarks to assess whether an LLM can faithfully embody a persona, maintain consistency over time, interact appropriately in social settings, and remain aligned with ethical norms.

One line of work focuses on general personalization. The LaMP benchmark (Salemi et al., 2024) evaluates whether LLMs can adapt to user-specific profiles across a range of tasks—such as rewriting text in a personalized tone or classifying based on individual preferences. While not limited to dialogue, LaMP highlights the broader need for systems that understand and leverage identity cues, and confirms that retrieval-based methods are especially effective for on-the-fly personalization.

A more targeted category of benchmarks examines character-specific fidelity. CharacterEval (Tu et al., 2024) is a Chinese benchmark featuring 77 characters from novels, each with multi-turn dialogues and detailed profiles. It defines 13 metrics across four dimensions—including character consistency, behavior realism, and conversational quality—and provides human and model-based evaluation for each. CharacterEval revealed that even GPT-4, when role-playing in Chinese, could be outperformed by fine-tuned local models on consistency, indicating that specialized training made a measurable difference. On the English side, RoleEval (Shen et al., 2023a) poses factual and commonsense questions about 300 well-known characters, assessing whether models accurately retain and apply character-specific knowledge. Results show that global models like GPT-4 excel with internationally known figures, while locally fine-tuned models do better with culturally specific roles. Meanwhile, TimeChara (Ahn et al., 2024) explores a new dimension—temporal consistency—by checking if a model role-playing a character at a given point in a story inadvertently leaks future events. Even the strongest models often violate timeline boundaries, suggesting the need for narrative-aware mechanisms to constrain temporal knowledge. SimulBench (Jia et al., 2025b) is a benchmark assessing LLM performance in interactive simulation scenarios—imaginative, role-playing, and tool-use tasks that unfold over multiple turns. The benchmark includes tasks like acting as a Linux terminal, playing text-based games, and complex, long-horizon simulations requiring dynamic interaction with a user.

Moving from factual to psychological evaluation, InCharacter (Wang et al., 2024g) assesses whether a role-play agent truly internalizes a character's personality. It uses an interview-style personality test: the agent (in role) is asked a battery of questions akin to a psychological survey, and its answers are compared to the expected personality profile of the character. Complementing this, RoleInteract (Chen et al., 2024e) evaluates social interaction skills at two levels: one-on-one conversation quality (empathy, politeness, etc.) and group dynamics (how well an agent plays its role in a multi-agent conversation). RoleInteract includes 500 characters and diverse scenarios (e.g., an office meeting with several personas). One finding was that some agents that excel in bilateral chat struggled in group settings—sometimes a normally consistent character would conform or get sidetracked when other AI characters were present, indicating influence and social pressure effects.

A different angle on evaluation is to test how well models understand a character from source material. Yuan et al. (2024) argue that a truly aligned role-play model should be able to read a narrative and produce a coherent character profile. They created the CROSS dataset of expert-written character profiles (covering attributes, relationships, events, and personality) for characters in novels. Models are evaluated on how well they can generate similar profiles after "reading" the novel (or being given chapters as input). In addition to intrinsic metrics (overlap with the expert profile), they evaluate extrinsically via a motivation recognition task: given a scenario from the story, does the model's profile help it answer why the character acted a certain way. Such evaluation drives home that role-play is not only about output style, but also about the model's internal model of the character.

Recent benchmark work pushes role-play evaluation toward more fine-grained and interaction-heavy settings. SocialBench (Chen et al., 2024d) measures individual- and group-level sociality over 500 characters and more than 30,800 multi-turn utterances, while RoleLLM (Wang et al., 2024j) introduces the large-scale RoleBench resource to benchmark and post-train character-level role-play at scale. RAIDEN (Wu et al., 2025a) and RMTBench (Xiang et al., 2025) further tighten the link between evaluation and realistic interaction design by using measurement-driven custom dialogues and user-centric bilingual role-play, respectively. Complementing these benchmarks, Lu et al. (2025a) show that role-play quality degrades measurably over long conversations when LLM-authored responses are compared against human-authored ones, while CharacterBench (Zhou et al., 2025a) and RoleMRC (Lu et al., 2025b) broaden evaluation toward character customization and the interaction between role-play and instruction following.

At the same time, the subsection is increasingly moving from static persona conditioning toward dynamic personalization. PersonaConvBench (Li et al., 2025c) studies personalized multi-turn reasoning and generation directly in conversational settings, and PERSONAMEM (Jiang et al., 2025) evaluates whether models can update user profiles as preferences evolve across sessions. On the modeling side, OpenCharacter (Wang et al., 2025c) scales synthetic persona generation for customizable role-play, and Abdulhai et al. (2025a) show that multi-turn reinforcement learning can improve persona consistency for simulated patients, students, and social partners.

A complementary line of work studies role-play as user simulation rather than character scoring. LLM Roleplay (Tamoyan et al., 2025) constructs persona- and goal-conditioned inquirer prompts to simulate human–chatbot interaction, grounding the setup in 200 natural dialogues collected from 20 participants across 10 goals and then comparing simulated and human dialogues in a human indistinguishability study. This makes it useful not only as a role-play method paper, but also as a data-generation and evaluation resource for studying how realistic persona-driven user simulations can become.

Table 6 summarizes the released benchmark and dataset resources discussed in CE-Roleplay, organized by resource scale, curation provenance, evaluator setup, and primary evaluation criteria.

Table 6: Released benchmarks and datasets for multi-turn CE-roleplay.

| Benchmark / Dataset | Dataset Size | | Data Curation | | Evaluator | | | | Evaluation Criteria |
| | Total #Dial. | Avg. #Turns/Dial. | Human-based | LLM-based | Rule-based | Human-as-a-judge[a] | LLM-as-a-judge | Agreement Check[b] | |
|---|---|---|---|---|---|---|---|---|---|
| PersonaChat (Zhang et al., 2018) | 10981 | ≈7.5[c] | yes | no | yes | yes | no | / | Perplexity, hits@1, human fluency / engagingness / consistency |
| PIPPA (Gosling et al., 2023) | 25940 | 40.41 | yes | yes | / | / | / | / | Dataset statistics only; not an evaluation benchmark |
| ChatHaruhi-54K (Li et al., 2023b) | 54726 | / | no | yes | yes | no | no | / | Script-continuation similarity, qualitative analysis |
| CharacterChat (Tu et al., 2023) | / | ≈7.5 | no | yes | no | yes | no | / | Human rating on social support, personality matching related criteria |
| InCharacter (Wang et al., 2024g) | 18304 | / | yes | yes | yes | yes | yes | yes | Measured Alignment, Personality Consistency |
| PRODIGy (Occhipinti et al., 2024) | 20850 | 4 | yes | no | yes | yes | no | / | CPPL, Acc@10, Acc@1, human preference |
| CharacterDial (Zhou et al., 2024a) | 1034 | 15.8 | yes | no | no | yes | yes | yes | Consistency, Human-likeness, Engagement, Quality, Safety, Correctness, Overall aggregate score |
| RoleInstruct (Tao et al., 2024) | 48677 | 14.85 | yes | yes | yes | yes | yes | / | Rouge-L, GPT-4 ranking, human evaluation |

*Table 6 continued*

| Benchmark / Dataset | Dataset Size | | Data Curation | | Evaluator | | | | Evaluation Criteria |
|---|---|---|---|---|---|---|---|---|---|
| | Total #Dial. | Avg. #Turns/Dial. | Human-based | LLM-based | Rule-based | Human-as-a-judge[a] | LLM-as-a-judge | Agreement Check[b] | |
| CharacterEval (Tu et al., 2024) | 1785 | 9.28 | yes | yes | no | yes | yes | yes | Character Consistency, Conversational Ability, Attractiveness, Personality Back-Testing |
| WIKIROLE(Lu et al., 2024) | 7186[d] | ≈5 | no | yes | no | no | yes | no | Consistency Accuracy, Knowledge Score, Rejection Accuracy, MT-Bench role-play score |
| RoleInteract (Chen et al., 2024e) | 7717[e] | 3.99 | yes | yes | yes | no | no | / | Multi-dimension social-interaction scores, memory-length and group-complexity analyses |
| SocialBench (Chen et al., 2024d) | 6420 | 5.13 | yes | yes | yes | no | no | / | Sociality dimensions, individual / group interaction quality, benchmark-wide role-play analysis |
| LLM Roleplay (Tamoyan et al., 2025) | 1000[f] | 5.3 | yes | yes | yes | yes | no | / | lexical diversity (TTR, Distinct-1/2), failure-mode detection, persona embodiment, indistinguishability, conversation quality |
| SimulBench (Jia et al., 2025b) | 500 | ≤4 | no | yes | no | no | yes | yes | GPT-4 score, pairwise win / tie / lose, human accept rate |
| CharacterBench (Zhou et al., 2025a) | 22859 | 11.22 | yes | yes | no | yes | yes | yes | Memory, knowledge, persona, emotion, morality, and believability/engagement |
| RAIDEN (Wu et al., 2025a) | 50518 | 10,30[g] | yes | yes | no | yes | yes | yes | Role consistency, knowledge, memory, and conversational ability |
| RoleMRC (Lu et al., 2025b) | 1400 | 3.5 | no | yes | yes | yes | yes | yes | Role-play MRC, free-chat / discussion ability, cross-dataset generalization |
| PERSONAMEM (Jiang et al., 2025) | 5990[h] | [15,30] | no | yes | yes | no | no | / | Dynamic profile tracking, preference updates, personalized response quality |
| PersonaConvBench (Li et al., 2025c) | 111239 | / | yes | no | yes | no | no | / | Classification (Accuracy, F1, MCC); Regression (RMSE, MAE); Generation (ROUGE-1, ROUGE-L, METEOR, BLEU, SBERT) |
| RMTBench (Xiang et al., 2025) | 160[i] | / | no | yes | no | yes | yes | yes | Emotional expression, Emotional understanding, Scenario development, Character understanding, Character maintenance, Security, User preference awareness |
| Human-vs-LLM Role-Play Benchmark (Lu et al., 2025a) | 4 | 23,>30[j] | yes | yes | no | yes | yes | yes | Understandable, Natural, Maintains Context, Interesting, Uses Knowledge, Overall Quality |

*Table 6 continued*

| Benchmark / Dataset | Dataset Size | | Data Curation | | Evaluator | | | | Evaluation Criteria |
|---|---|---|---|---|---|---|---|---|---|
| | Total #Dial. | Avg. #Turns/Dial. | Human-based | LLM-based | Rule-based | Human-as-a-judge[a] | LLM-as-a-judge | Agreement Check[b] | |

[a] Does not include human evaluation done only for agreement checking with an LLM judge.

[b] If LLM-as-a-judge is used, whether the paper reports agreement checking against human annotators on a subset.

[c] Gained by: 164,356 utterances / 10,981 dialogues $\approx$ 14.97 utterances/dialogue $\approx$ 7.5 turns.

[d] 7086 train sessions + 100 test sessions.

[e] 1,063 SA Style + 1,408 SA Know. + 193 EP Situ. + 1,016 EP Emo. + 773 CM Short + 1,348 CM Long + 586 Pos. + 724 Neu. + 606 Neg.

[f] 200 natural human-chatbot dialogues (20 participants $\times$ 10 goals) and 800 synthetic LLM-chatbot dialogues (4 inquirer LLMs $\times$ 20 personas $\times$ 10 goals), for 1,000 dialogue pairs total.

[g] 21.22 utterances per short dialogue/ 65.18 utterances per long dialogue

[h] Verified by released benchmark on Hugging Face.

[i] 80 characters $\times$ 2 dialogues each.

[j] 23 turns in the main scenario; 3 additional scenarios (>30 turns each).

Overall, the rapid advancements in LLM-based role-play research demonstrate a clear evolution: moving from basic persona adherence to sophisticated, adaptable, multi-agent interactions, supported by advanced evaluation and continuous alignment efforts. This evolution highlights ongoing efforts toward creating engaging, realistic conversational agents capable of sustained, immersive role-play interactions. Such role-play also plays a crucial role throughout the other conversational-engagement settings discussed in the remaining subsections.

### 3.2.2   Conversational Engagement in Healthcare

Healthcare is one of the key domains where multi-turn conversational large language models (LLMs) demonstrate significant potential. A defining characteristic of these models in the medical domain is their ability to emulate a doctor's role by engaging in task-oriented, context-aware dialogues with patients. Unlike single-turn medical knowledge question-answering systems such as HuaTuo (later renamed BenTsao) (Wang et al., 2023a), ChatMed, ShenNong-TCM, MING (Liao et al., 2024a), and DoctorGLM (Xiong et al., 2023), which respond to isolated queries with all available information at once (Mutabazi et al., 2021), multi-turn healthcare LLMs operate in a setting where patient information is often incomplete or ambiguous at the outset.

Ideally, multi-turn healthcare conversational LLMs should be capable of proactively generating a sequence of questions to refine their understanding of the patient's condition through iterative inquiry. This concept has been referred to by different names in various studies, including chain of questions (CoQ) (Chen et al., 2023), proactive questioning (Chen et al., 2023), symptom inquiry (Zhang et al., 2023; Liao et al., 2024b), proactivity (Bao et al., 2023), and information seeking (Li et al., 2024b). Despite the different terminology, these works describe the same core capability: dynamically gathering relevant clinical details over multiple exchanges to support more accurate diagnosis. Additionally, multi-turn healthcare LLMs must retain dialogue history to support seamless continuity across turns and integrate medical knowledge so that responses remain accurate, reliable, and contextually appropriate.

While prior surveys (Valizadeh & Parde, 2022; Shi et al., 2024c) provide a broad overview of medical dialogue systems, including both pre-LLM and LLM-based approaches, our focus is specifically on multi-turn LLMs and related evaluation frameworks. Traditional medical dialogue systems typically rely on rule-based logic or task-specific neural networks, which can be rigid and require extensive manual engineering. We explore the specific tasks these models address, the architectural advancements in their development, and the character-

istics of the datasets used for pre-training and fine-tuning. In addition, we discuss the evaluation frameworks used to assess these models, highlighting key metrics, established benchmarks, the growing focus on their information-seeking capabilities, and the challenges faced in online conversational healthcare systems.

**Development of Conversational Medical LLMs** The development of multi-turn healthcare LLMs has progressed rapidly. The representative systems discussed below demonstrate how customized training methodologies, curated datasets, and specialized evaluation metrics can improve healthcare-domain performance and expand the use of LLMs in medical and psychological applications.

Several studies have aimed to make medical LLM communication more interactive and accessible. Clinical Camel (Toma et al., 2023) is an open-source medical LLM that introduces dialogue-based knowledge encoding (DBKE) to transform dense medical texts into conversational formats, enhancing multi-turn dialogue capability. Fine-tuned from LLaMA-2 via QLoRA, Clinical Camel outperforms GPT-3.5, GPT-4, and Med-PaLM 2 on benchmarks including USMLE (FSMB, 2023), PubMedQA (Jin et al., 2019), and MedQA (Jin et al., 2021). T-Agent (Hu et al., 2024) enhances medical dialogue generation through a term-aware approach, integrating a term extraction tool and a term prediction model within a two-stage training framework; experiments demonstrate improvements in ROUGE scores and term extraction F1. In contrast, APP (Ask Patients with Patience) (Zhu et al., 2025a) operates as a reasoning and guidance layer atop an existing LLM to support diagnostic decision-making in online consultations, emphasizing grounded reasoning, entropy minimization, and patient-centered communication without requiring additional fine-tuning.

A majority of studies have focused on building medical LLMs that generate accurate and reliable responses in dialogue settings through SFT on healthcare-specific data. DISC-MedLLM (Bao et al., 2023) is fine-tuned on DISC-Med-SFT, a 400K-sample Chinese medical instruction dataset covering single-turn Q&A, multi-turn consultations, and multiple-choice Q&A, evaluated via rule-based accuracy for single-turn and GPT-4 scoring for multi-turn conversations across proactivity, accuracy, helpfulness, and linguistic quality. BianQue (Chen et al., 2023) integrates multi-turn doctor–patient Q&A to enable a Chain of Questioning (CoQ) approach that emulates real consultations, fine-tuned on ChatGLM-6B using the 2.4M-sample BianQueCorpus, improving CoQ by balancing questions and suggestions. BiMediX (Pieri et al., 2024), a bilingual mixture-of-experts LLM supporting English and Arabic, is fine-tuned via QLoRA and outperforms Med42 and Meditron on English medical benchmarks while substantially surpassing the bilingual Jais-30B on Arabic and bilingual assessments. In psychological counseling, CPsyCounX (Zhang et al., 2024a) is fine-tuned over InternLM2-7B-Chat on CPsyCounD, a corpus of 3,134 high-quality multi-turn consultation dialogues, while PsycoLLM (Hu et al., 2025b) is fine-tuned on single-turn Q&A, multi-turn dialogues, and knowledge-based Q&A atop Qwen1.5-14B-Chat. Complementing these, the SMILE method (Qiu et al., 2024) leverages ChatGPT to convert single-turn counseling pairs from PsyQA (Sun et al., 2021) into multi-turn dialogues via diverse topic prompts and format filtering, yielding SMILECHAT, a corpus of 55K multi-turn counseling dialogues in Chinese.

Beyond SFT alone, several studies adopt a comprehensive pipeline encompassing pre-training, SFT, and RLHF. Zhongjing (Yang et al., 2024), the first Chinese medical LLaMA-based LLM, exemplifies this through continuous pre-training, targeted SFT, and RLHF optimization. The authors construct CMtMedQA—approximately 7,000 QA pairs from authentic doctor–patient exchanges across 14 clinical departments and over 10 medical scenarios—and apply a self-instruct methodology (Wang et al., 2023c) to standardize responses into a uniform, professional, and empathetic style. The external medical knowledge graph CMeKG (Byambasuren et al., 2019) is further integrated to verify medical accuracy and safety. HuaTuoGPT (Zhang et al., 2023) fine-tunes a medical consultation LLM on data distilled from ChatGPT and real-world clinical data, using RLAIF to align outputs with the complementary strengths of both sources. Evaluation spans rule-based NLP metrics, LLM-based assessment of language quality and diagnostic utility, and expert human evaluation of diagnosis and treatment accuracy, demonstrating superior performance over baselines. The same team subsequently developed HuaTuoGPT II (Chen et al., 2024g), which adopts a one-stage domain adaptation protocol that unifies heterogeneous pre-training and supervised data into a unified instruction-output format for efficient knowledge injection, achieving performance competitive with GPT-4 on multiple benchmarks and excelling in Chinese medical and pharmacist license examinations.

Several studies also incorporate Direct Preference Optimization (DPO) into the full training pipeline. Aquila-Med (Zhao et al., 2024) undergoes continued pre-training, SFT, and DPO using 12,727 preference pairs, demonstrating significant improvements in fluency, relevance, completeness, and proficiency across single- and multi-turn medical consultations. Similarly, Qilin-Med (Ye et al., 2023) employs a multi-stage pipeline combining domain-specific continued pre-training, SFT, and DPO over the ChiMed dataset (Tian et al., 2019)—encompassing question answering, plain text, knowledge graphs, and dialogues. SFT improves accuracy over the Baichuan-7B baseline on CMExam (Liu et al., 2023), while the subsequent DPO phase achieves BLEU-1 of 16.66 and ROUGE-1 of 27.44 on the Huatuo-26M test set, reflecting further gains beyond SFT.

In addition to the mainstream approaches, Google's Articulate Medical Intelligence Explorer (AMIE) (Tu et al., 2025) represents a comprehensive advancement in conversational medical AI for training and evaluation. AMIE utilizes a chain-of-reasoning strategy within a simulated environment, incorporating self-play and automated feedback mechanisms to enhance its diagnostic dialogue capabilities across diverse medical conditions. Its training data includes both human-curated and AI-generated datasets, encompassing multiple-choice medical questions, long-form medical reasoning queries, clinical note summaries, and simulated dialogues based on various medical conditions. Evaluated through a randomized, double-blind crossover study using Objective Structured Clinical Examination scenarios, AMIE demonstrated superior diagnostic accuracy compared to primary care physicians and received higher ratings from both specialist physicians and patient actors on multiple assessment criteria.

**Multi-turn Medical LLM Evaluation Framework** Beyond the development of large language models (LLMs), we also found a growing body of literature focused on designing evaluation frameworks specifically for medical LLM systems, as traditional LLM evaluation methods often fall short in addressing the complexity and safety requirements of clinical applications.

Recent studies have proposed evaluation frameworks for medical LLMs that emphasize interactive quality, clinical reasoning, and human-centered communication. MedGPTEval (Xu et al., 2023) assesses LLMs like ChatGPT and Dr.PJ using 27 multi-turn dialogue cases and 7 case reports, measuring the accuracy, empathy, and clinical logic of the LLMs' responses. Similarly, Liao et al. (2023) proposes an automated evaluation framework, emphasizing assessing LLM abilities, such as recognizing knowledge limitations, gathering relevant information, and improving diagnostic accuracy. The researchers reformulated medical multiple-choice questions from the USMLE into consultation tasks, creating a specialized benchmark for assessment. Additionally, they verify that fine-tuning with a consultation-specific dataset reduced hallucinations and improved benchmark performance.

In addition, a growing number of studies emphasize simulation-based interactive evaluation to approximate real-world clinical consultation. For instance, Liao et al. (2024b) introduces the Automated Interactive Evaluation (AIE) framework featuring the State-Aware Patient Simulator (SAPS), which incorporates a state tracker, memory bank, and response generator to support dynamic, multi-turn evaluation. Similarly, MMD-Eval (Multi-turn Medical Dialogue Evaluation) (Liu et al., 2025d) offers a locally deployable, task-oriented dialogue simulator, providing more resource-efficient and consistent evaluations than LLM-based scoring. MediQ (Li et al., 2024b) introduces an interactive benchmark for medical evaluation by simulating clinical interactions between a patient system and an adaptive expert system, emphasizing the assessment of information seeking ability. It employs abstention strategies to better estimate confidence and determine when to seek additional information. The benchmark converts existing datasets like MedQA and Craft-MD into interactive formats, simulating clinical interactions for evaluation purposes.

Complementing these efforts, MedFuzz (Ness et al., 2024) focuses on evaluating the adversarial robustness of healthcare LLMs by introducing ambiguous or unexpected inputs that could lead to clinical misconceptions and harmful decisions. It questions the assumption that high benchmark scores reflect real-world reliability by introducing complexities like ambiguous patient traits and biased data into medical QA benchmarks. Results show that GPT-3.5, GPT-4, and Med-PaLM 2 perform worse on the "MedFuzzed" benchmark, showing their vulnerability to clinical biases, demographic stereotypes, and incomplete data interpretation.

Recently, OpenAI introduced *HealthBench* (Arora et al., 2025), an open-source benchmark constructed from real-world healthcare conversations designed explicitly for evaluating LLM performance. HealthBench in-

cludes 5,000 multi-turn conversations averaging 2.6 turns per dialogue (ranging from 1 to 19 turns), capturing diverse and realistic patient–clinician interactions. It emphasizes clinical accuracy, appropriate communication depth, handling of uncertainty, and context awareness. Unique to HealthBench is its extensive use of physician-developed rubrics, consisting of over 48,000 distinct evaluation criteria, making it a robust tool for assessing nuanced aspects of clinical dialogue. Moreover, HealthBench introduces specialized subsets such as *HealthBench Consensus*, validated across multiple experts, and *HealthBench Hard*, comprising notably challenging interactions that test the limits of current LLM capabilities. This systematic, clinician-validated approach significantly enriches the landscape of multi-turn medical LLM evaluation by explicitly aligning automated assessment closely with clinical judgment.

More recent benchmarks shift the emphasis from raw consultation fluency toward controlled clinical stress tests. Guo et al. (2026a) show that diagnostic accuracy can degrade substantially once patient behaviors unfold across multiple turns rather than in static question-answer form, while the Medical Incremental N-Turn Benchmark (medical MINT) (Fang et al., 2026) and MedDialBench (Luo et al., 2026) isolate different degradation pathways such as premature answering, information lures, and adversarial patient behaviors. CPGBench (Tan et al., 2026) evaluates whether models can detect and adhere to clinical practice guidelines during extended conversations, and MedMT-Bench (Yang et al., 2026) focuses explicitly on long-horizon memory and understanding in medical dialogue.

The evaluation side is also becoming more rubric-driven and patient-facing. MedDialogRubrics (Gong et al., 2026) expands multi-turn consultation assessment with 5,200 synthetic cases and more than 60,000 fine-grained rubrics, while MEDPI (V. et al., 2026) evaluates patient-facing medical interactions with accreditation-aligned dimensions spanning process quality, treatment safety, and doctor–patient communication. MindEval (Pombal et al., 2025) and Mazhar et al. (2026) extend the conversation into mental-health support, MedAidDialog (Nigam et al., 2026) adds multilingual consultation data, and Dr. Assistant (Guo et al., 2026b) couples structured diagnostic-reasoning data with RL to improve inquiry quality in consultations.

Table 7 summarizes the released benchmark and dataset resources discussed in CE-Healthcare, organized by resource scale, curation provenance, evaluator setup, and primary evaluation criteria.

Table 7: Released benchmarks and datasets for multi-turn CE-healthcare.

| Benchmark / Dataset | Dataset Size | | Data Curation | | Evaluator | | | | Evaluation Criteria |
|---|---|---|---|---|---|---|---|---|---|
| | Total #Dial. | Avg. #Turns/Dial. | Human-based | LLM-based | Rule-based | Human-as-a-judge[a] | LLM-as-a-judge | Agreement Check[b] | |
| SMILECHAT (Qiu et al., 2024) | 55165 | 5.7 | no | yes | yes | yes | no | / | Automatic metrics and human preference |
| MedGPTEval (Xu et al., 2023) | 27[c] | 4 | yes | no | no | yes | no | / | Accuracy, Empathy, Clinical Logic, Robustness |
| HuatuoGPT / Huatuo26M + MedDialog (Zhang et al., 2023) | 94874[d] | 10.4 | yes | yes | yes | yes | yes | yes | Pairwise medical response quality (doctor-like language / symptom inquiry / treatment reliability / helpfulness), doctor evaluation (diagnosis accuracy / treatment recommendation accuracy / medication & prescription accuracy), QA metrics (BLEU / ROUGE / GLEU / Distinct) |
| Zhongjing / CMtMedQA (Yang et al., 2024) | ≈70k | 5.7 | yes | yes | no | yes | yes | no | Expert win / tie / loss on safety, professionalism, fluency |
| DISC-MedLLM / DISC-Med-SFT (Bao et al., 2023) | ≈470k | / | yes | yes | yes | no | yes | no | Accuracy, GPT-4 judging for proactivity, helpfulness, linguistic quality |

*Table 7 continued*

| Benchmark / Dataset | Total #Dial. | Avg. #Turns/Dial. | Human-based | LLM-based | Rule-based | Human-as-a-judge[a] | LLM-as-a-judge | Agreement Check[b] | Evaluation Criteria |
|---|---|---|---|---|---|---|---|---|---|
| USMLE-derived consultation benchmark (Liao et al., 2023) | 2909[e] | mixed[f] | yes | yes | yes | no | yes | no | ROUGE-1 Recall/Precision/F1 for patient and doctor, final-task Accuracy, Average Turn, Average Length |
| BianQue / BianQueCorpus (Chen et al., 2023) | 2437190 | / | yes | yes | yes | no | no | / | BLEU-1/2/3/4, ROUGE-1/2/L, and Proactive Questioning Ability (PQA) |
| HuatuoGPT-II (Chen et al., 2024g) | ≈5.39M[g] | / | yes | yes | yes | yes | yes | / | Pairwise medical response quality for multi-turn dialogue; medical benchmark performance; Accuracy |
| BiMed1.3M / BiMediX (Pieri et al., 2024) | ≈1.3M | 4.72 | yes | yes | yes | yes | yes | / | Benchmark accuracy on MCQA / QA datasets, UPHILL accuracy, multi-turn conversation preference on coherence/ facts/ diagnostic accuracy/appropriate leading questions/ quality |
| AIE / SAPS (Liao et al., 2024b) | 50/200[h] | 10 | yes | yes | yes | yes | yes | yes | patient-simulator fidelity, doctor consultation quality, human/GPT-4 ratings from doctor and patient perspectives |
| CPsyCounD (Zhang et al., 2024a) | 3134 | 8.7 | yes | yes | no | no | yes | no | Comprehensiveness, Professionalism, Authenticity, Safety |
| CPsyCounE (Zhang et al., 2024a) | 45[i] | 6.36 | yes | no | no | no | yes | no | Comprehensiveness, Professionalism, Authenticity, Safety |
| MEDIQ (Li et al., 2024b) | 12723[j] | / | yes | yes | yes | no | yes | no | Patient Factuality, Relevance, Final-Answer Accuracy, Question Efficiency |
| Aquila-Med data pipeline (Zhao et al., 2024) | 332727[k] | / | yes | yes | yes | no | yes | / | Benchmark Accuracy, GPT-4 pairwise comparisons, four-dimension dialogue scores |
| MMD-Eval (Liu et al., 2025d) | 2636 | / | yes | yes | yes | yes | yes | yes | Communication competence, clinical diagnostic competence |
| APP / ReMeDi-derived benchmark (Zhu et al., 2025a) | 100 | 6 | yes | yes | yes | yes | no | / | Diagnosis accuracy, trustworthiness, entropy-based diagnostic confidence, accessibility, empathy, relevant response, relationship fostering |
| HealthBench (Arora et al., 2025) | 5000 | 2.6 | yes | yes | no | no | yes | yes | Completeness, Accuracy, Context awareness, Communication quality, Instruction following |
| MindEval (Pombal et al., 2025) | 50 | 20 | no | yes | no | yes | yes | yes | Clinical Accuracy & Competence, Ethical & Professional Conduct, Assessment & Response, Therapeutic Relationship & Alliance, AI-Specific Communication Quality |

*Table 7 continued*

| Benchmark / Dataset | Dataset Size | | Data Curation | | Evaluator | | | | Evaluation Criteria |
|---|---|---|---|---|---|---|---|---|---|
| | Total #Dial. | Avg. #Turns/Dial. | Human-based | LLM-based | Rule-based | Human-as-a-judge[a] | LLM-as-a-judge | Agreement Check[b] | |
| MEDPI (V. et al., 2026) | 7097 | / | no | yes | no | no | yes | no | 105-dimension patient-facing evaluation, safety, process, communication quality |
| MedDialogRubrics (Gong et al., 2026) | 5200 | ≤12 | yes | yes | yes | no | yes | yes | Weighted rubric score over 60,000 rubrics; Precision, Recall, Accuracy, F1; rubric-match precision / inquiry completeness |
| MedMT-Bench (Yang et al., 2026) | 400 | 22 | yes | yes | no | yes | yes | yes | Long-horizon medical memory, recall, and dialogue understanding |
| MedAidDialog (Nigam et al., 2026) | 2980[l] | 2.85 | no | yes | yes | yes | no | yes | Medical Safety; Symptom Extraction; Context Memory; Diagnostic Correctness; Conversational Flow; Efficiency |
| CPGBench (Tan et al., 2026) | 32155 | / | yes | yes | no | no | yes | yes | Guideline detection, adherence scoring, decade-scale clinical recommendation analysis |
| MINT (Medical Incremental N-Turn Benchmark) (Fang et al., 2026) | 1035 | 9.36 | yes | no | yes | no | no | / | Initial-answer accuracy, final-answer accuracy, abstention rate, average initial-answer turn, accuracy drop, flip rate, true-to-false rate, false-to-true rate, restoration rate |
| FAITH-M / CARE (Mazhar et al., 2026) | 167 | 30.45 | yes | no | yes | yes | no | yes | Therapeutic-principle ratings, mental-health support quality, domain-shift robustness |
| MedDialBench (Luo et al., 2026) | 7225 | ≤20 | no | yes | no | no | yes | yes | Dose-response robustness to adversarial patient behaviors, diagnostic accuracy drop |

[a] Does not include human evaluation done only for agreement checking with an LLM judge.

[b] If LLM-as-a-judge is used, whether the paper reports agreement checking against human annotators on a subset.

[c] 34 cases in total: 27 multi-turn medical dialogues + 7 single-turn case reports.

[d] 68,888 multi-turn dialogues generated by two ChatGPT instances + 25,986 genuine multi-turn doctor-patient conversations from real online platforms.

[e] 2,909 total instances = MedQA 1,127 + QMax 1,483 + Sample Exam 299.

[f] Avg 5.00–5.12 options per question, avg 80.85–119.28 tokens per medical information, avg 14.01–14.32 tokens per initial request, avg 7.81–8.25 items per extracted key-value pair list.

[g] 5,252,894 medical documents as the pre-training corpus and 142K real-world medical questions from Huatuo-26M for fine-tuning.

[h] Interactive evaluation environment: 50 patient-simulator cases / 4,000 simulator-test questions + 200 doctor-eval cases (50 HospitalCases + 150 MedicalExam).

[i] 45 cases manually selected from SMILECHAT(Qiu et al., 2024).

[j] Derived from MedQA(Jin et al., 2021) with 10,178 / 1,272 / 1,273 train/dev/test samples.

[k] 320,000 SFT instances (+ 12,727 DPO preference pairs).

[l] 2,980 base dialogues; 20,860 if you count the full 7-language parallel corpus as separate instances.

In the development of healthcare conversational LLMs, a widely adopted strategy is to construct multi-turn medical dialogue datasets and fine-tune models with SFT, sometimes followed by RL-based methods. Interestingly, while some studies in general multi-turn dialogue suggest that SFT+RLHF may not yield

significant improvements, this pipeline has proven surprisingly effective in the healthcare domain. This may be attributed to the relatively high quality and domain specificity of medical data, which contrasts with the noisier, less structured data used in broader multi-turn applications. Evaluation metrics in this area include traditional rule-based NLP measures (e.g., BLEU and ROUGE), diagnostic accuracy, and both AI-based and human assessments using more nuanced, domain-knowledge-intensive criteria. Compared with earlier healthcare dialogue work, newer benchmarks more frequently report explicit human–AI agreement or alignment checks when LLM-based evaluation is used, although this practice remains uneven and is still absent from a substantial portion of the literature. Notably, information seeking and clinical reasoning are essential capabilities in this domain and are increasingly prioritized in newer benchmarks, which now place more emphasis on clinical relevance than on conventional NLP metrics alone.

### 3.2.3 Conversational Engagement in Education

Multi-turn conversational systems are increasingly central in education, enabling dynamic back-and-forth interactions that mimic human tutoring. Recent work can be grouped into three key areas: Intelligent Tutoring Systems, which use dialogic exchanges to teach or guide students; Automated Grading & Feedback, where AI provides iterative evaluation and comments on student work; and Scenario Simulation, which involves simulating students or classrooms with AI agents. Below, we survey advances in each category, highlighting how multi-turn conversation enhances educational effectiveness.

**Intelligent Tutoring Systems**

**Socratic and Strategy-Guided Tutoring**  Early LLM-based tutors often fell into a simple question-answer pattern, providing direct answers and explanations that made students passive. To address this, researchers have developed Socratic and guided approaches that engage learners in multi-turn dialogue. SocraticLM (Liu et al., 2024a) is a notable example, proposing a "thought-provoking" teaching paradigm in which the tutor asks open-ended questions and prompts the student's reasoning instead of giving away solutions. SocraticLM was trained on SocraTeach, a new dataset of 35k multi-turn dialogues where a simulated teacher guides students with diverse cognitive states through math problems. Similarly, Kargupta et al. (2024) tackle the "answer-too-direct" issue in the coding domain with TreeInstruct, an LLM-based tutor that plans a hierarchy of questions to help students debug code. TreeInstruct models the student's knowledge state and asks targeted questions for each error, effectively guiding learners to independently correct mistakes. It achieved state-of-the-art results on code debugging benchmarks and demonstrated in a user study that students could fix bugs with minimal direct hints.

Beyond specific algorithms, others have explored controlling LLM tutors through high-level pedagogical strategies. StratL (Puech et al., 2025) introduces a pedagogical steering framework that optimizes prompts to guide an LLM tutor along a predefined teaching plan represented as a graph, rather than allowing unconstrained generation. Applied to Productive Failure—a strategy where students wrestle with problems before receiving instruction—StratL successfully steers the tutor to withhold answers and encourage trial-and-error. In a field experiment with 17 high school students, the StratL-guided tutor adhered to the target strategy and facilitated student-driven discovery. Collectively, these efforts mark a shift from straightforward Q&A toward multi-turn, strategy-aware dialogue, bringing LLM tutors closer to human teachers who ask, hint, and adapt rather than simply tell.

**Adaptive Tutoring**  Traditional intelligent tutoring systems often suffered from a "one-size-fits-all" design that poorly accommodated individual learners. PACE (PersonAlized Conversational tutoring agEnt) (Liu et al., 2025a) directly addresses this limitation by explicitly modeling student learning styles and personas. Built on the Felder–Silverman learning style model, PACE's multi-turn tutor adapts its explanations accordingly—providing concrete examples for sensory learners and abstract prompts for intuitive ones—while employing Socratic questioning to stimulate critical thinking. To support this, the authors construct a new dataset of personalized tutoring dialogues simulating students with diverse backgrounds and personalities interacting with an LLM tutor.

Moreover, LLM tutors can track a student's step-by-step reasoning in multi-turn exchanges and provide targeted help at the right moment. This ability to model the student within the conversation marks a key advantage of multi-turn tutoring. An illustrative real-world deployment is JeepyTA (Liu et al., 2025e), a GPT-based virtual teaching assistant used in an online course forum. JeepyTA monitors student questions on course material and responds in seconds with context-specific help, adapting its style to the informal, conversational tone of forum discussions. It distinguishes logistical queries from conceptual ones and provides hints or explanations accordingly. By continuously engaging with students' follow-up questions, such an always-on conversational TA exemplifies how personalization and instant adaptivity can scale via LLMs.

**Evaluating LLM Tutoring in Mathematics**  Recent research further explores ways to deepen assessments of LLM tutoring capabilities, particularly regarding their subject expertise and pedagogical effectiveness in mathematics. An influential early example is the work of Macina et al. (2023), who introduced *MathDial* to systematically evaluate LLMs' abilities as tutors, emphasizing faithful and equitable teaching. Their approach involved collecting high-quality, teacher–student dialogues through human–LLM interactions, with InstructGPT simulating student behaviors—including common misconceptions—and expert annotators assuming the teacher role. Diverse student responses were elicited using temperature sampling, prompting the LLM to generate realistic errors. Human teachers then employed scaffolding strategies to guide the simulated students toward solutions. Dialogue quality was ensured through rigorous human annotation. Evaluation metrics included the simulated student's success rate and the telling@k score, indicating how frequently teachers prematurely revealed answers. Models fine-tuned on the MathDial dataset demonstrated improved student outcomes by emphasizing hints and prompting over direct answers. Similarly, Ding et al. (2024) introduce *SocraticLLM*, a knowledge-enhanced model emulating Socratic questioning to foster critical thinking and self-discovery. To support this method, they developed the publicly available *SocraticMATH* dataset, containing structured Socratic dialogues that cover 513 primary-school math topics. Another benchmark, *MathTutorBench* by Macina et al. (2025), specifically targets one-on-one math tutor–student dialogues, assessing LLM tutors across multiple dimensions, including Math Expertise (accuracy in problem-solving), Student Understanding (ability to diagnose and correct misconceptions), and Teacher Response Quality (effectiveness in providing hints and Socratic guidance).

Most recently, retrieval-augmented generation (RAG) techniques have further advanced the field. For instance, Feng et al. (2024) develop *CourseAssist*, a system that grounds tutor-generated responses explicitly in course syllabi and lecture notes, ensuring alignment with instructor expectations. Levonian et al. (2025) explore alignment methods to enhance generative tutors' outputs, making responses safer, pedagogically appropriate, and closely relevant to student queries, significantly improving multi-turn algebra tutoring dialogues. Expanding upon this theme, Scarlatos et al. (2025) argue for the necessity of adaptive AI tutors that subtly adjust responses across dialogues, strategically guiding students toward correct understanding without explicit intervention. They operationalize this idea by generating multiple potential tutor replies at each conversational turn and then assessing them with a student model—which predicts students' subsequent correctness—and a pedagogical evaluator. By ranking tutor responses based on these dual criteria, they successfully fine-tune a new tutor model through direct preference optimization, thereby rewarding interactions that consistently promoted student learning.

**Bridging Research and Real Classrooms**  New benchmark work makes the tutoring side of the subsection substantially more concrete. KMP-Bench (Shi et al., 2026) evaluates pedagogical intelligence in K–8 mathematics through both dialogue-based and verifiable tutoring tasks, while MRBench (Maurya et al., 2025) audits 192 tutoring conversations and 1,596 responses across eight pedagogical dimensions. Tutor-Bench (Srinivasa et al., 2025) adds 1,490 rubric-backed samples centered on explanation, feedback, and hinting, and SAFETUTORS (Hazra et al., 2026) explicitly measures pedagogical safety failures that only emerge in extended tutoring dialogues.

At the dataset level, EduDial (Wei et al., 2025) contributes 34,250 teacher–student dialogue sessions covering 345 knowledge points, TeachLM (Perczel et al., 2025) leverages 100,000 hours of authentic one-on-one tutoring data and uses synthetic students for scalable multi-turn evaluation, and ConvoLearn (Sharma et al., 2026) introduces 2,134 dialogic tutoring dialogues grounded in knowledge-building theory. Steindl et al. (2025) release the GMADE tertiary-level math tutoring conversations and show that prompt moderation alone

can underperform unmoderated tutoring, while Scarlatos et al. (2026) demonstrate that realistic student simulation remains difficult even when prompting, SFT, and preference optimization are combined.

The emerging trend in conversational tutors involves integrating research insights directly into classroom practice, facilitated by the increasing availability of large language models tailored specifically for education. Major industry players, including Google with LearnLM (Wiggers, 2024) and Anthropic with Claude for Education (Anthropic, 2025), have actively embraced this approach, developing specialized educational versions of their flagship AI models. These initiatives reflect a convergence between academic research and industry applications, emphasizing personalized, adaptive tutoring methods grounded in evidence-based dialogue strategies and instructional principles. They also connect directly to the adaptation, evaluation, and safety challenges discussed later in §5, where classroom deployment raises questions about robustness, pedagogical alignment, and overreliance.

### Automated Feedback & Grading Support

**Automated Feedback Support**   Large language models are increasingly leveraged to generate automated feedback and grading, aiming to support instructors with large-scale assessments (Silva & Costa, 2025). Similarly, in open-ended writing tasks, LLMs can produce fluent and plausible comments that appear insightful, yet often include content not grounded in the student's work (Jia et al., 2024b). This lack of faithfulness (e.g., fabricated critiques or irrelevant suggestions) is a critical limitation, as unfaithful feedback can mislead or confuse learners. Improving the accuracy and alignment of feedback with students' actual mistakes has therefore become a central research focus.

To address these limitations, recent work has introduced strategies to make LLM feedback more interactive, adaptive, and pedagogically grounded. Daheim et al. (2024) incorporate a verification step in which an LLM first analyzes student reasoning step-by-step to pinpoint errors, then guides a tutor model to deliver targeted hints—significantly reducing hallucinated advice and improving alignment to the student's actual misunderstanding. Scarlatos et al. (2024) instead optimize feedback quality through human pedagogical preferences: they define a rubric covering correctness, encouragement, and misconception-addressing, use GPT-4 to label outputs along these criteria, and apply reinforcement learning to fine-tune the model, yielding measurably higher factual correctness and better tutoring alignment. Going further, Nair et al. (2024) close the feedback loop by having the model simulate student revisions in response to its own feedback and iteratively refine that feedback to maximize draft improvement—producing greater actual writing gains and enhanced pedagogical qualities compared to static feedback.

Studies are also examining how well LLM feedback aligns with real classroom needs and how students and instructors perceive it. Initial deployments show mixed but encouraging results. In a graduate computer science course, an automated tool generated paragraph-level comments on project reports that students found broadly helpful, yet the instructor observed that the AI's comments did not always reflect the assignment's pedagogical objectives—preferring to treat LLM output as a draft to edit rather than send directly to students (Jia et al., 2024a). This underscores a recurring limitation: LLM feedback often requires human oversight to remain consistent with instructional goals. Domain-focused deployments, however, demonstrate clearer benefits when carefully integrated. Riazi & Rooshenas (2025) develop an LLM-driven tutor for a databases course capable of analyzing student entity-relationship diagrams and generating detailed, context-specific critiques and follow-up questions. Similarly, in medical education, LLMs have been used to produce explanatory feedback for board-style multiple-choice questions; expert evaluators found such feedback relevant and useful—not a replacement for human judgment, but a meaningful supplement to the numeric scores students typically receive (Tomova et al., 2024).

Efforts like Lohr et al. (2025)'s, which prompt LLMs to produce specific types of feedback from established educational taxonomies (e.g., an error-identification vs. a hint vs. an elaboration), further illustrate the push toward more controlled and purposefully designed feedback messages. In addition to feedback itself, researchers are beginning to evaluate LLMs on related teaching skills such as asking good questions—for instance, Dr.Academy (Chen et al., 2024j) assesses whether LLMs can generate high-quality, higher-order questions in line with Anderson and Krathwohl's taxonomy, finding that models like GPT-4 already show strong capability in formulating deep conceptual questions. Together, these advances show a clear develop-

mental trajectory: from basic feedback generation that often wandered off-target, to increasingly faithful, adaptive, and pedagogically-aware feedback loops facilitated by LLMs' multi-turn interaction capacity.

**Automated Grading Support**   In parallel with feedback generation, researchers have started leveraging LLMs to grade student work and provide evaluative judgments (scores, ratings, or rubric-based assessments). Automatic grading by AI is not entirely new, but LLMs offer a unified, flexible approach that can handle open-ended responses more like a human grader. Recent studies suggest that, for certain types of assignments, LLM graders can approach human-level performance in both consistency and accuracy. Capdehourat et al. (2025) explore LLMs for scoring short free-response questions in Spanish, a scenario involving language complexity beyond the typical English-centric datasets. They found that state-of-the-art models (including GPT-4 and advanced open-source LLMs) could predict expert graders' scores with over 95% accuracy in a three-tier grading scale, and even 98% accuracy on simpler right/wrong judgments.

Another study compared ChatGPT directly against university instructors for grading full exam papers in higher education. Out of 463 Master's-level exam responses graded, about 70% of the AI's assigned scores fell within a 10% margin of the human-given score, and about 31% were within a 5% margin (Flodén, 2025). Teachers involved in the experiment expressed surprise at how closely ChatGPT's evaluations matched their own in these exams. However, important discrepancies were observed. The AI grader tended to be more conservative, avoiding very high or very low scores on individual questions.

In the domain of essay scoring, which demands understanding of content, organization, style, and often providing feedback, LLMs still struggle to meet human-level nuance. Kostic et al. (2024) conduct a case study using GPT-4 to evaluate German-language student essays at a business school. The LLM could generate a score and some comments, but it often failed to apply the rubric criteria consistently and lacked the depth of feedback that human lecturers provided. Complex aspects of writing quality (critical analysis, creativity, etc.) proved difficult for the model to judge correctly, highlighting a gap between what current LLMs can do and the "nuanced requirements" of real essay evaluation.

**Scenario Simulation**   Recent advances have explored scenario simulation as a means to leverage LLMs for enacting multi-turn educational interactions between virtual teachers and students. Early work focused on using LLM-simulated student profiles to evaluate learning materials. For example, Generative Students introduced a prompt-based architecture (grounded in the Knowledge-Learning-Instruction framework) to instantiate diverse student profiles defined by mastered vs. confused knowledge components (Lu & Wang, 2024). Each simulated student (powered by GPT-4) answered multiple-choice questions, producing responses that aligned with its knowledge profile. Notably, these generative students exhibited answer patterns highly correlated with real student performance, correctly flagging many of the same difficult questions. This result suggested that realistic virtual learners can serve as proxies for human students in content evaluation, allowing instructors to identify problematic questions before deployment.

Subsequent research broadened the fidelity of simulated students by incorporating individual differences in ability and personality. Liu et al. (2024g) develop a personality-aware simulation framework that enriches student profiles with both cognitive level (e.g., language proficiency) and noncognitive traits (e.g., conscientiousness). In a language tutoring scenario, an LLM could then produce diverse student utterances consistent with a given persona, which in turn successfully triggered the tutor's adaptive scaffolding strategies. Building on this idea, Jin et al. (2025) introduce TeachTune, a system enabling teachers to test their pedagogical conversational agents (PCAs) against diverse simulated students. Teachers specify a student's presumed prior knowledge and motivation, and an LLM-driven student agent engages in a multi-turn chat with the PCA. This automated student–teacher dialogue reveals how well the PCA adapts its explanations and feedback to different learner needs, going beyond single-turn Q&A tests. The TeachTune pipeline ensured that each simulated student's behavior remained faithful to its profile, with measured deviations under 5–10%. These efforts highlight that richly modeled virtual students can support teacher agents and tutoring systems by surfacing potential shortcomings in adaptive instruction in a cost-effective manner.

Researchers have also scaled scenario simulation to multi-party settings. Zhang et al. (2025e) propose SimClass, a framework in which multiple LLM-based agents assume typical classroom roles. A novel class-level control mechanism orchestrates the agents' turn-taking and topic flow to emulate a live classroom lesson.

In user trials with real students, SimClass was able to simulate dynamic classroom interactions featuring both teacher–student exchanges and student–student discussions. The emergent group behavior was strikingly human-like—the student agents would ask and answer each other's questions and collaboratively debate topics, creating an enlivened atmosphere. These collective simulations improved the human participant's learning experience by maintaining engagement and peer-like dialogue support. The success of SimClass demonstrates that LLMs can collectively model complex social dynamics of a classroom, opening the door to virtual class rehearsals and large-scale peer interaction scenarios.

Scenario simulation has since been extended to specialized pedagogical needs for both learners and instructors. To better support low-performing or metacognitively weak students, Li et al. (2025a) devise a pipeline that automatically generates struggling student agents with diverse learning deficiencies, then filters them through a two-round LLM evaluation—validated by human experts—to ensure authenticity. The resulting high-fidelity "at-risk" agents allow educators and tutoring systems to safely experiment with interventions targeting self-regulation and reflective thinking, without ethical concerns of testing on real students. On the instructor side, Hu et al. (2025a) use LLM-based simulation to improve lesson planning: an LLM plays out a full classroom lesson from a teacher's draft plan, simulating both instruction and student reactions, then generates a reflective critique. These simulated insights are used to iteratively refine the lesson plan before deployment. Additionally, Wang et al. (2024d) propose Book2Dial, a framework to automatically generate synthetic teacher–student dialogues from textbook content to address data scarcity in developing educational chatbots. Three dialogue-generation approaches—Multi-turn QG-QA, Dialogue Inpainting, and LLM-based Role-Playing—are introduced and evaluated using automated metrics (e.g., coherence, answerability, factual consistency) and human judgment (specificity). The results demonstrate that LLM-based Role-Playing performs best, highlighting a cost-effective method for chatbot training in educational domains.

Table 8 summarizes the released benchmark and dataset resources discussed in CE-Education, organized by resource scale, curation provenance, evaluator setup, and primary evaluation criteria.

Table 8: Released benchmarks and datasets for multi-turn CE-education.

| Benchmark / Dataset | Dataset Size | | Data Curation | | Evaluator | | | | Evaluation Criteria |
|---|---|---|---|---|---|---|---|---|---|
| | Total #Dial. | Avg. #Turns/Dial. | Human-based[a] | LLM-based | Rule-based | Human-as-a-judge[b] | LLM-as-a-judge | Agreement Check[c] | |
| MathDial (Macina et al., 2023) | 2861 | 4.96 | yes | yes | yes | yes | no | yes | Success@k, Telling@k, BLEU / BERTScore, human tutoring quality |
| Book2Dial (Wang et al., 2024d) | 889 | 5.96 | no | yes | yes | yes | yes | yes | Answer Relevance, Informativeness, Groundedness, Coherence, Answerability, Factual Consistency, Specificity |
| MULTI-DEBUG (Kargupta et al., 2024) | 150 | $\leq$20 | yes | no | yes | yes | no | yes | Success, Relevance, Indirectness, Logic, Average Turns |
| Stepwise verification tutoring data (Daheim et al., 2024) | 1002 | / | yes | yes | yes | yes | yes | yes | Verification F1, sBLEU, KF1, BF1, targeted / actionable judgments |
| SocraTeach / SocraticLM (Liu et al., 2024a) | 35k | 5.28 | no | yes | yes | no | yes | yes | Teaching quality; correctness recognition, explanation success, BLEU, Rouge |
| SocraticMATH / SocraticLLM (Ding et al., 2024) | 6846 | 4.96 | yes | yes | yes | yes | yes | no | BLEU-1/2/3/4, ROUGE-1/2/L, METEOR, BARTScore; human and GPT-4 scoring on Reliability and Socratic strategy |

*Table 8 continued*

| Benchmark / Dataset | Dataset Size | | Data Curation | | Evaluator | | | | Evaluation Criteria |
|---|---|---|---|---|---|---|---|---|---|
| | Total #Dial. | Avg. #Turns/Dial. | Human-based[a] | LLM-based | Rule-based | Human-as-a-judge[b] | LLM-as-a-judge | Agreement Check[c] | |
| MRBench(Maurya et al., 2025) | 192 | 5.04 | yes | yes | yes | yes | yes | yes | Mistake identification, mistake location, revealing of the answer, providing guidance, actionability, coherence, tutor tone, human-likeness |
| PACE tutoring dataset (Liu et al., 2025a) | 1410[d] | 10 | no | yes | yes | no | yes | no | BLEU, ROUGE, METEOR, BERTScore, coherence, relevance, personalization, engagement, consistency, inspiration, pairwise win/lose/tie |
| MathTutorBench (Macina et al., 2025) | 9125 | [3.04, 5.78] | yes | yes | yes | no | yes | yes | Expertise / student understanding / pedagogical abilities, socratic questioning, student solution correctness, mistake location, mistake correction, scaffolding generation, and pedagogical instruction following |
| TutorBench (Srinivasa et al., 2025) | 1490[e] | [1,2] | yes | yes | no | no | yes | yes | Adaptive explanation, actionable feedback, hint quality, rubric-guided judging |
| TeachLM (Perczel et al., 2025) | 100[f] | [30,80] | yes | yes | yes | no | yes | yes | Student talk time, average words per tutor turn, mean questions per interrogative turn, number of turns before wrap-up, uncovering student background and learning context, checking coding skills |
| EduDial (Wei et al., 2025) | 34250[g] | ≈11 | yes | yes | no | yes | yes | yes | Insight, response, feedback, thinking, fluency, interactivity, emotional support, adaptability, goal, relevance, coverage |
| QATD-2k (Scarlatos et al., 2026) | 2000 | / | yes | no | yes | yes | yes | yes | Dialogue acts, correctness, error-making, knowledge acquisition, language use, inducing tutor responses |
| ConvoLearn (Sharma et al., 2026) | 2134 | ≈20 | yes | yes | no | yes | yes[h] | yes | Teacher effectiveness, completeness, quality issues, and safety |
| KMP-Dialogue (Shi et al., 2026) | 4.6k | 9.3 | yes | yes | yes | yes | yes | yes | Challenge, explanation, modelling, practice, questioning, feedback |
| KMP-Skills (Shi et al., 2026) | 6k | [2,3] | yes | yes | yes | no | yes | no | Multi-turn follow-up problem-solving, error detection and correction, mathematical problem generation |
| SAFETUTORS (Hazra et al., 2026) | 2820[i] | / | no | yes | yes | yes | yes | / | Pedagogical safety, harm taxonomy, multi-turn tutoring failure analysis |

*Table 8 continued*

| Benchmark / Dataset | Dataset Size | | Data Curation | | Evaluator | | | | Evaluation Criteria |
|---|---|---|---|---|---|---|---|---|---|
| | Total #Dial. | Avg. #Turns/Dial. | Human-based[a] | LLM-based | Rule-based | Human-as-a-judge[b] | LLM-as-a-judge | Agreement Check[c] | |

[a] Whether the data are manually curated by humans.

[b] Does not include human evaluation done only for agreement checking with an LLM judge.

[c] If LLM-as-a-judge is used, whether the paper reports agreement checking against human annotators on a subset.

[d] 1,200/60/150 train/val/test; 6 personas; tutor avg 54.56 words/utt, student avg 44.67 words/utt.

[e] 1,490 samples curated by human experts: 828 multimodal (55.6%) with images of handwritten/typed work/screenshots; 662 text-only (44.4%).

[f] TeachLM is trained on 100,000 hours of one-on-one longitudinal student–tutor interactions from Polygence. For the evaluation protocol, each evaluation point is based on 100 simulated multi-turn conversations.

[g] 34,250 total dialogue sessions: 13,700 dialogues for SFT training; 20,550 for DPO training.

[h] Ordinal RoBERTa classifier trained on ConvoLearn serves as automated scorer.

[i] 5,955 total (3,135 single-turn + 2,820 multi-turn).

In summary, multi-turn conversational simulations serve as a valuable sandbox for educational innovation. By leveraging LLMs to generate realistic student and teacher behaviors, researchers and practitioners can prototype and test interventions rapidly, ethically, and inexpensively. These simulations complement live studies: they can uncover issues and inform design decisions before real students are involved, and suggest which approaches merit real-world trials. From generative students for item analysis to full classrooms for teacher training, the common thread is that rich, multi-turn interactions are the fabric of these simulations.

### 3.2.4 Conversational Engagement in Jailbreak

In multi-turn settings, LLMs face not only heightened requirements for consistency but also an increased risk of malicious exploitation. Although LLMs excel at various tasks, their vulnerabilities can lead to harmful outputs, such as generating dangerous instructions, that highlight significant limitations compared to human judgment. The phenomenon of multi-turn jailbreaking, where adversaries bypass guardrails over a series of exchanges, has thus emerged as a critical area of concern.

Most prior research has focused on single-turn jailbreaks, in which adversaries use a single prompt, often few-shot, to trigger harmful content. For example, optimization-based methods by Zou et al. (2023) and Liu et al. (2024f) utilize gradient-derived gibberish suffixes to elicit such outputs, but these techniques depend on internal token probability knowledge and are not applicable to closed-source models. To explore alternative strategies, subsequent studies (Carlini et al., 2023; Yu et al., 2023) have investigated unconventional communication patterns, such as role-playing scenarios, and developed multi-turn conversational methods that leverage the entire dialogue history. These approaches demonstrate that multi-turn jailbreaking, where even seemingly innocuous prompts contribute to later interventions, presents a far more complex challenge.

At the same time, some work frames jailbreaking itself as an iterative dialogue. PAIR (Chao et al., 2025) uses Prompt Automatic Iterative Refinement, in which an attacker LLM revises semantic jailbreak prompts from target-model feedback, often finding successful black-box attacks in fewer than twenty queries. Johnny / PAP (Zeng et al., 2024a) pushes this further toward everyday interaction by deriving persuasive adversarial prompts from a social-science persuasion taxonomy and reporting attack success rates above 92% on Llama-2-7B-Chat, GPT-3.5, and GPT-4. Together, these studies show that multi-turn jailbreaks are not only about decomposing harmful intent, but also about adaptively refining social-engineering strategies across turns.

**Crescendo and ActorAttack**  Multi-turn jailbreaks can employ more diverse strategies. Crescendo (Russinovich et al., 2025; Ren et al., 2024) is one such strategy: it gradually induces target LLMs to

provide harmful information through a sequence of seemingly benign prompts. The intuition behind this strategy is that an LLM's agreement with a small initial request can increase the likelihood of compliance with subsequent, larger demands. By asking innocuous but implicitly suggestive questions, an attacker can shift the model's token distribution toward continuations that contain more harmful information. Thus, the dialogue trajectory as a whole, rather than any single question, produces the jailbreak. In addition, both the implicitness of the questions and the order in which they are presented are important to the attack's success. Following the Crescendo idea, Ren et al. (2024) generalize the method used by Russinovich et al. (2025), which requires fixed, human-crafted seed instances and therefore makes diverse attack generation challenging. Their contribution focuses on self-discovering diverse attack clues inside the model's prior knowledge via network structures and clue classification, constructing a two-layer relation tree according to Latour's actor-network theory (Latour, 2005).

**Decomposition** Another important strategy for multi-turn jailbreaks is to decompose a harmful prompt into several pieces, each containing less overtly malicious content (Gibbs et al., 2024; Zhou et al., 2024c; Liu et al., 2024d). This enables language models to incrementally generate harmful content through multi-turn dialogue. Decomposing the original malicious query into several less harmful sub-questions can evade LLM guardrails and induce harmful responses. Because models can use in-context information across turns, harmful knowledge can be accumulated and combined in the final turn. Under such decomposition, each intermediate turn may appear aligned, while the cumulative harmful content across the dialogue results in an overall alignment failure. Following this idea, Wang et al. (2024a) train a red-team jailbreak agent through interactions with target LLMs to generate decomposed yet coherent jailbreak prompts. Jiang et al. (2024b) propose a scenario-based jailbreak method to disguise malicious intent from LLM guardrails, assuming that LLMs can detect direct harmful intent but may be misled if the attacker creates a scenario in which others are planning harmful actions and the attacker is positioned as the protector. The attack is decomposed across multiple turns: first, the attacker describes others' harmful intent and seeks prevention; then, the attacker asks about possible evidence items; finally, the attacker requests an example harmful plan for comparison.

**Datasets for Multi-Turn Jailbreaks** AdvBench (Zou et al., 2023) and HarmBench (Mazeika et al., 2024) are two widely used benchmarks for jailbreaking LLMs. AdvBench provides 500 harmful behavioral instructions and 500 harmful/toxic strings, while HarmBench contains 510 unique harmful behaviors spanning 400 textual and 110 multimodal instances. Though neither is designed specifically for multi-turn jailbreaks, both serve as representative testbeds. Building on them, Russinovich et al. (2025) manually craft Crescendo multi-turn prompts, while Ren et al. (2024) scale attack-chain generation via LLM self-talk guided by six clues from Latour's actor-network theory, compiling the results into SafeMTData. Gibbs et al. (2024) apply a word substitution cipher (Handa et al., 2024) to HarmBench data processed with Mixtral-8x7b to isolate the effect of multi-turn structure, and Zhou et al. (2024c) use a decomposed version of AdvBench for jailbreaking. In contrast, Liu et al. (2024d) draw from HarmfulQ (Shaikh et al., 2023), 200 explicit harmful questions in English, for multi-turn jailbreak construction. Wang et al. (2024a) use AdvBench for training and JBB (Chao et al., 2024) for evaluation, while Jiang et al. (2024b) leverage the BeaverTails dataset (Ji et al., 2023), covering malicious queries across 14 refusal-test categories, and apply sentence transformers (Ni et al., 2022) to generate multi-turn dialogues and harmful actions targeting GPT-4o.

The newer literature now includes native multi-turn jailbreak resources rather than only converted single-turn seeds. FITD (Weng et al., 2025), Many-Turn Jailbreaking (Yang et al., 2025d), and Yang et al. (2025e) all benchmark sustained jailbreak behavior directly, while SafeDialBench (Cao et al., 2025) contributes more than 4,000 multi-turn safety dialogues across 22 scenarios and seven jailbreak strategies. MHJ (Li et al., 2024a) adds a complementary human-authored resource with 2,912 prompts across 537 multi-turn jailbreak conversations, paired with tactic labels, reviewer checks, and red-teamer commentary, making it especially useful for studying realistic adversarial behavior rather than only model-generated attack chains. On the attack-generation side, RACE (Ying et al., 2025) uses reasoning-augmented conversations, SEMA (Feng et al., 2026) learns a multi-turn attacker without hand-crafted strategies, and Tempest (Zhou & Arel, 2025) and Siren (Zhao & Zhang, 2025) automate multi-turn jailbreak search or learning to better approximate real human adversaries.

**Defenses against Multi-Turn Jailbreaks**  While numerous multi-turn jailbreak methods have been proposed, effective defenses specifically targeting such attacks remain scarce. Conventional approaches—perplexity filters, input/output filters, rephrasing/retokenization, and random token dropping—show limited efficacy against extended conversational attacks. Among targeted efforts, Yu et al. (2024a) show that system prompts combined with Chain-of-Thought reasoning (Wei et al., 2022) can partially counteract attacks by refusing harmful queries, albeit at the cost of reduced helpfulness. Gibbs et al. (2024) find that NeMoGuardrails (Rebedea et al., 2023) can be overzealous, flagging even benign prompts, and Liu et al. (2024d) observe that stronger defenses frequently compromise usability. Collectively, these findings reveal a stark security-usability trade-off, underscoring the urgent need for adaptive, context-aware defense strategies that maintain both robustness and user-friendly performance.

Recent defensive and red-teaming work sharpens this trade-off further. MHJ (Li et al., 2024a) makes the gap especially concrete: expert human multi-turn jailbreaks exceed 70% attack success rate on HarmBench against defenses that report single-digit rates under automated single-turn attacks, and they also recover dual-use knowledge from machine-unlearned models. Persona Jailbreaking (Sandhan et al., 2026), Echo Chamber (Alobaid et al., 2026), and Mastermind (Li et al., 2026d) show that conversational history itself can be weaponized to gradually shift behavior or refine attack strategies. On the evaluation side, Mai et al. (2026) build realistic fraud and cybercrime misuse scenarios, while X-Boundary (Lu et al., 2025c) explicitly targets the multi-turn safety-versus-usability boundary instead of optimizing refusal in isolation.

Table 9 summarizes the released benchmark and dataset resources discussed in CE-Jailbreak, organized by resource scale, curation provenance, evaluator setup, and primary evaluation criteria.

Table 9: Released benchmarks and datasets for multi-turn CE-jailbreak.

| Benchmark / Dataset | Dataset Size | | Data Curation | | Evaluator | | | | Evaluation Criteria |
|---|---|---|---|---|---|---|---|---|---|
| | Total #Dial. | Avg. #Turns/Dial. | Human-based[a] | LLM-based | Rule-based | Human-as-a-judge[b] | LLM-as-a-judge | Agreement Check[c] | |
| Matched cipher jailbreak dataset (Gibbs et al., 2024) | 6536[d] | / | no | yes | no | yes | no | / | UTQ, Jailbreak Success, Non-Block Rate, False Positives |
| SafeMTData / actor-network attack data (Ren et al., 2024) | 1680[e] | 10.2 | no | yes | no | no | yes | / | ASR, LLM Judge |
| CoSafe (Yu et al., 2024a) | 1400 | 3 | no | yes | no | yes | yes | yes | Harmful Rate, ASR, annotator agreement |
| MHJ (Li et al., 2024a) | 537 | 5.4 | yes | no | no | yes | yes | yes | Human multi-turn jailbreak robustness, tactic taxonomy, ASR, attack metadata |
| Decomposed AdvBench dialogues (Zhou et al., 2024c) | 520[f] | 2.43, 2 | yes | yes | no | no | yes | / | Harmful-Dialogue Rate, Harmfulness Score |
| HarmfulQ conversations (Liu et al., 2024d) | 200 | [4,7] | no | yes | no | yes | no | / | Harmfulness, Executability, Acceptance Rate |
| Foot-in-the-Door (Weng et al., 2025) | 180 | 12 | no | yes | no | no | yes | / | Multi-turn ASR, self-corruption analysis, benchmark-based attack evaluation |
| SafeDialBench (Cao et al., 2025) | 4053 | 4.8 | no | yes | no | yes | yes | yes | Fine-grained safety taxonomy, diverse jailbreak scenarios, multi-turn safety evaluation |
| Siren (Zhao & Zhang, 2025) | 255 | [3, 4] | no | yes | yes | yes | yes | yes | Learning-based human-like jailbreak behavior simulation, multi-turn ASR |

*Table 9 continued*

| Benchmark / Dataset | Dataset Size | | Data Curation | | Evaluator | | | | Evaluation Criteria |
| | Total #Dial. | Avg. #Turns/Dial. | Human-based[a] | LLM-based | Rule-based | Human-as-a-judge[b] | LLM-as-a-judge | Agreement Check[c] | |
| --- | --- | --- | --- | --- | --- | --- | --- | --- | --- |
| Tempest (Zhou & Arel, 2025) | 100 | ≤5 | no | yes | no | no | yes | / | Tree-search multi-turn jailbreak discovery, automatic attack generation |
| MTJ-Bench / Many-Turn Jailbreaking (Yang et al., 2025d) | 640 | 2 | no | yes | no | yes | yes | yes | Sustained jailbreak success, many-turn persistence, cross-model robustness |
| Simpler Multi-Turn Jailbreaks (Yang et al., 2025e) | 30 | 8 | no | yes | no | no | yes | / | StrongREJECT rubric-based LLM judge, Chain-of-thought LLM evaluator |
| SEMA (Feng et al., 2026) | 159 | [1, 10] | no | yes | yes | no | yes | no | Learned multi-turn attacker evaluation, ASR across datasets and judges |

[a] Whether the data are manually curated by humans.

[b] Does not include human evaluation done only for agreement checking with an LLM judge.

[c] If LLM-as-a-judge is used, whether the paper reports agreement checking against human annotators on a subset.

[d] 4,136 harmful matched pairs (plus 1,200 completely-benign and 1,200 semi-benign controls); 382 harmful pairs in the hand-labeled evaluated subset.

[e] Verified via released data on Hugging Face: SafeMTData/SafeMTData.

[f] Two decomposition variants: manually decomposed AdvBench (total: 520) (constructed by the authors, avg 2.43 sub-questions per decomposition, max 4) and automatically decomposed AdvBench (total: 520) (avg 2 sub-questions per decomposition, generated via few-shot GPT-4 prompting)

## 4 Improvements

Recent advances in enhancing multi-turn interactions with Large Language Models (LLMs) have pursued diverse strategies for handling the challenges of extended conversations. As illustrated in Fig. 3, we organize this literature into three main branches: (1) Model-Centric Approaches, which adapt or refine the model itself through techniques such as in-context learning, supervised fine-tuning, reinforcement learning, and new architectures (§4.1); (2) External Integration Approaches, which augment LLMs with resources such as memory, retrieval, and knowledge graphs to improve context use and factual grounding (§4.2); and (3) Agent-Based Approaches, which treat LLMs as proactive systems that iteratively act, reflect, or collaborate over sustained interactions (§4.3).

Figure 3 should be read as an organizing lens rather than a mutually exclusive partition of the literature. Many works combine ideas from more than one branch; in such cases, we place each paper according to its primary technical contribution and cross-reference it elsewhere when useful. In terms of the general formulation in §2, these improvements typically operate by strengthening conditioning on the interaction history $h_t$, enriching auxiliary state $z_t$, or improving optimization over longer trajectories $\tau$ rather than isolated turns.

### 4.1 Model-Centric Approaches

This section surveys key model-centric strategies aimed at improving LLM performance on multi-turn interaction tasks. As dialogue tasks shift from static, single-turn queries to dynamic, multi-turn exchanges, traditional modeling techniques often fall short. We examine four major approaches that address these challenges from different angles: (1) In-Context Learning, which explores prompt-based adaptation using multi-turn exemplars (§4.1.1); (2) Supervised Fine-Tuning, which focuses on data curation and training

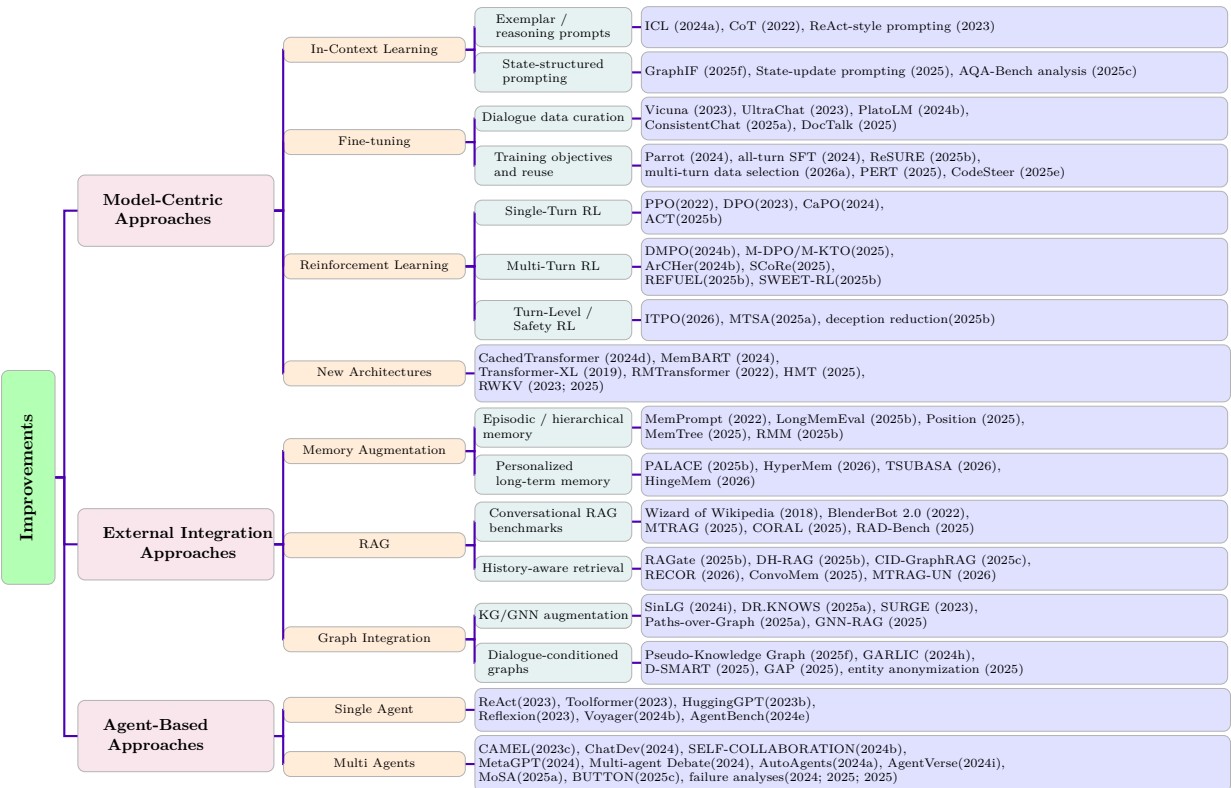

Figure 3: Taxonomy of improvement methodologies in multi-turn LLM interactions. The taxonomy is an organizing lens rather than a mutually exclusive partition: some papers span multiple branches, but each work is placed under the branch most central to its primary contribution.

strategies for maintaining coherence and context over multiple rounds (§4.1.2); (3) Reinforcement Learning, which aligns multi-turn behaviors with human preferences through trajectory-level optimization (§4.1.3); and (4) New Architectures, which reimagine the Transformer design to better support long-range memory and dialogue flow (§4.1.4). Together, these techniques represent an evolving toolkit for building LLMs that can reason, remember, and respond effectively in sustained interactions.

### 4.1.1 In-Context Learning

In-context learning (ICL) (Dong et al., 2024a) in multi-turn settings yields nuanced outcomes according to recent empirical studies. While providing exemplars usually aids single-turn tasks, naive in-context prompting with interactive multi-turn examples can sometimes hurt performance in sequential dialogue scenarios (Yang et al., 2025c). For instance, on the AQA-Bench sequential reasoning benchmark, certain models (e.g. LLaMA2 and DeepSeek-LLM in a coin-guessing game environment) actually saw their accuracy drop when given a single in-context demonstration, recovering only as more examples were provided. This surprising inversion of the typical few-shot benefit is attributed to the multi-round nature of the examples, which may cause overfitting to specific interaction trajectories. In other words, multi-turn example formats behave differently from standard one-turn Q&A prompts. Consistent with this, dedicated dialogue benchmarks like MT-Bench (Zheng et al., 2023) and its fine-grained successor MT-Bench-101 (Bai et al., 2024) highlight that multi-turn interactions demand evaluating facets such as context retention, proactivity, and adaptability that one-shot QA tests often overlook.

Across diverse domains, a variety of prompting techniques have been explored to improve multi-turn LLM interactions, with mixed empirical results. In code generation, interactive multi-turn prompting proves

beneficial: the InterCode benchmark (Yang et al., 2023) shows that step-by-step reasoning strategies such as Chain-of-Thought (Wei et al., 2022) and ReAct, or iterative plan-and-refine prompting, yield concrete performance gains by leveraging compiler feedback across turns. In the clinical domain, Clinical Camel (Toma et al., 2023) uses dialogue-based knowledge encoding to cast medical texts into a Q&A format, surpassing GPT-3.5 on multiple medical benchmarks in few-shot evaluations. By contrast, persona-driven prompts do not universally improve objective accuracy: a systematic study found that adding a persona via system prompts generally fails to boost factual question-answering performance (Zheng et al., 2024a). Nevertheless, persona-centric role-play can enhance conversational quality in the right context—CharacterChat (Tu et al., 2023) employs MBTI-aligned personas with preset behaviors and dynamic memory, facilitating effective personalized social support dialogues. Finally, explicitly steering dialogue structure shows promise: StratL (Puech et al., 2025) optimizes prompts to guide an LLM tutor along a predefined teaching plan, successfully inducing a more pedagogically effective multi-turn tutoring strategy in practice.

Collectively, these findings suggest that the impact of ICL and prompt design on multi-turn performance is highly context-dependent: gains emerge when the prompting strategy aligns well with the task's interactive dynamics, while misaligned or naive prompts can degrade performance in complex multi-turn settings.

Recent prompt-level work increasingly replaces raw-history prompting with explicit state or structure representations. GraphIF (Li et al., 2025f) converts multi-turn instruction contexts into directed relation graphs and injects them back as graph prompts, improving long-distance constraint tracking without additional model training. Likewise, Liu (2025) propose a training-free state-update prompting strategy that reconstructs dialogue state and selectively reminds the model of prior context, reducing token costs while improving multi-hop performance in long-horizon dialogues. Together, these studies suggest that ICL for multi-turn settings is becoming less about adding more demonstrations and more about controlling which conversational structure is made explicit to the model.

### 4.1.2 Supervised Fine-Tuning

Classic supervised fine-tuning (SFT) methods like InstructGPT (Ouyang et al., 2022) and FLAN (Chung et al., 2024), along with parameter-efficient techniques such as AdapterDrop (Rücklé et al., 2021) and LoRA (Hu et al., 2022), have driven major progress in single-turn LLM tasks. For a comprehensive survey of SFT, see Zhang et al. (2026). However, to enhance LLMs' multi-turn interaction ability, SFT must be modified; as Wang et al. (2024f) point out, treating each round independently can lead to context forgetting and incoherent responses in multi-turn dialogues. This limitation has motivated the development of tailored strategies that both leverage realistic multi-turn data and adjust training methods to fully exploit such data.

SFT remains one of the most widely used techniques for improving multi-turn LLM performance. As discussed in §3, incorporating domain-specific multi-turn interaction data into SFT consistently yields further gains. Evidence spans multi-turn instruction following—including math reasoning and multilingual contexts (Bai et al., 2024; Maheshwary et al., 2025; Liang et al., 2024)—role-play (Ding et al., 2023; Occhipinti et al., 2024), and clinical dialogue (Toma et al., 2023; Zhang et al., 2023; Bao et al., 2023; Yang et al., 2024; Liao et al., 2023; Ye et al., 2023; Zhao et al., 2024; Pieri et al., 2024; Wang et al., 2024d; Macina et al., 2023; Scarlatos et al., 2025). These results underscore the need for further research into fine-tuning strategies tailored specifically to multi-turn settings.

**Realistic Multi-Turn Dialogue Data Curation**  Recent works have addressed multi-turn challenges by generating datasets that capture natural dialogue flows. For example, Vicuna (Chiang et al., 2023) fine-tunes on user-shared ChatGPT conversations to retain genuine multi-turn interactions, though scaling such data remains costly. Other methods, such as those in UltraChat (Ding et al., 2023), use self-chat to create extensive multi-turn data; however, these auto-generated dialogues can be overly scripted and lack diversity. To mitigate these issues, approaches like PlatoLM (Kong et al., 2024b) incorporate sophisticated user simulators (e.g., the "Socratic" agent) to produce more dynamic and realistic multi-round dialogues, thereby enhancing topic shifts and follow-up naturalness. In addition, Parrot (Sun et al., 2024) generates multi-turn data by first mimicking human asking styles for query generation, then constructing negative responses that simulate context neglect or misunderstanding, and finally combining this with context-aware preference signals for fine-tuning.

**Optimized SFT Approaches**   Alongside dataset curation, new fine-tuning strategies have been proposed to fully exploit multi-turn data, targeting loss design, context length, and training efficiency. Vicuna (Chiang et al., 2023) employs a modified loss function and extended context lengths, supported by gradient checkpointing (Chen et al., 2016) and FlashAttention (Dao et al., 2022) to handle long dialogue histories efficiently. ChatGLM2 (GLM et al., 2024) similarly extends context length while reducing GPU memory costs via multi-query attention (Shazeer, 2019) and causal masking. More generally, Teng et al. (2024) optimize a combined cross-entropy and KL divergence loss across all dialogue turns rather than the final turn alone, yielding more coherent outputs and more efficient training. Finally, Chen et al. (2025e) address multi-round gradient cancellation—where gradients from early turns can cancel those from later, more informative ones—by doubling the loss weights of the final two rounds, ensuring that the most influential guidance steps drive model updates and improving selection of optimal initial actions.

In summary, these advances illustrate two complementary avenues for enhancing multi-turn interactions in LLMs with SFT: generating realistic dialogue data and optimizing fine-tuning strategies. Together, these approaches promote better context retention and coherent multi-turn responses, marking a significant step toward more robust conversational models.

The newest SFT work further sharpens both data quality and supervision design. ConsistentChat (Chen et al., 2025a) uses skeleton-guided dialogue construction to improve consistency from the data-generation stage, while ReSURE (Du et al., 2025b) explicitly regularizes unreliable supervision signals in multi-turn fine-tuning. Other work targets the training pipeline itself: Li et al. (2026a) study data selection for multi-turn instruction tuning, Ma et al. (2025) reuse earlier medical dialogue data through prefix-enhanced fine-tuning, and DocTalk (Lee et al., 2025) uses graph-based dialogue synthesis to scale conversational training data. These studies reinforce a broader trend already visible in the earlier literature: multi-turn SFT is increasingly treated as a data-engineering and supervision-engineering problem rather than only a scaling problem.

### 4.1.3   Reinforcement Learning

Reinforcement learning (RL) has become central to aligning LLMs with human preferences, improving their safety, helpfulness, and coherence. While initial RL successes focused on optimizing single-turn interactions using human or AI feedback, real-world conversations often span multiple turns, introducing complexities beyond single-turn methods. Addressing these challenges has motivated recent developments in specialized multi-turn RL algorithms designed explicitly for extended conversational interactions.

**Single-Turn RL**   Reinforcement learning from human feedback (RLHF) aligns LLMs with human preferences via reward models trained on human rankings, substantially improving response quality and safety. Notably, a 1.3B-parameter InstructGPT fine-tuned with RLHF outperformed a 175B-parameter GPT-3 in human evaluations, exhibiting greater truthfulness and reduced toxicity (Ouyang et al., 2022). Reinforcement learning from AI feedback (RLAIF) extends this paradigm by substituting AI-generated feedback for human labels; Anthropic's Constitutional AI (Bai et al., 2022) exemplifies this approach, training models via AI-authored principles and self-critique to reduce harmful behaviors without direct human supervision. Both RLHF and RLAIF have proven effective for aligning single-turn LLM responses with human values.

Early RLHF successes largely utilized Proximal Policy Optimization (PPO), an on-policy method offering stable and sample-efficient fine-tuning of large models. For example, OpenAI's InstructGPT (Ouyang et al., 2022) employed PPO to optimize human-preference-based reward models, significantly enhancing helpfulness and reducing harmful outputs with minimal performance loss on traditional NLP benchmarks.

While PPO-based RLHF is effective, it can be complex due to reward modeling and tuning. Direct Preference Optimization (DPO) (Rafailov et al., 2023) simplifies this by converting RLHF into a supervised classification task, analytically deriving optimal policies from preference models. Context-aware Preference Optimization (CaPO; Sun et al., 2024) extends DPO by integrating context-aware preferences, promoting correct context utilization. Additionally, Action-Based Contrastive Self-Training (ACT) (Chen et al., 2025b) applies DPO quasi-online, enhancing LLMs' clarifying behaviors in ambiguous contexts while preserving DPO's simplicity.

Many prior studies have investigated RLHF for enhancing multi-turn LLM interactions (Sun et al., 2024; Bai et al., 2024; Wang et al., 2024f; Lù et al., 2024; Du et al., 2025a; Wu et al., 2024b; Yang et al., 2023;

Chen et al., 2025e; Bao et al., 2023; Yang et al., 2024; Ye et al., 2023; Chen et al., 2024g; Scarlatos et al., 2024). A horizontal comparison of these works, however, reveals a nuanced and somewhat contradictory picture. On one hand, specialized domains such as medical QA (Yang et al., 2024; Ye et al., 2023) and coding (Yang et al., 2023; Chen et al., 2025e) demonstrate substantial gains from carefully implemented approaches like DPO and interactive feedback loops. Similarly, the Parrot framework (Sun et al., 2024) highlights how explicitly training on negative examples can steer models toward better contextual alignment and fewer repetitive errors.

On the other hand, broader analyses from the tool-use MINT benchmark (Wang et al., 2024f) and MT-Bench-101 (Bai et al., 2024) reveal that generalized RLHF methods frequently yield limited or even negative impacts in multi-turn scenarios. This discrepancy highlights the inherent limitations of RLHF methods in terms of generalizability, emphasizing the necessity for tailored RL strategies explicitly designed for multi-turn interactions.

**Multi-Turn RL**   Recent research has therefore extended preference optimization and reinforcement learning techniques to optimize entire conversation trajectories rather than single responses. Below, we review key developments organized by technique.

**Multi-Turn DPO and Variants**   A straightforward way to align multi-turn behavior is to generalize the single-turn preference optimization objective across an entire dialogue trajectory. Direct Multi-Turn Preference Optimization (DMPO) (Shi et al., 2024b) is one such approach that adapts DPO to multi-turn agent tasks. A technical obstacle in multi-turn DPO is that the partition function (normalization factor) no longer cancels out between preferred and dispreferred trajectory pairs, complicating the loss computation. DMPO addresses this by re-formulating the optimization: it replaces the per-response probability ratio with a state-action occupancy measure and normalizes for sequence length differences. The result is a novel DMPO loss that comes with theoretical justification for multi-turn settings.

Related work by Xiong et al. (2025), which we covered in §3.1.2, introduced an iterative multi-turn preference learning framework with two implementations: Multi-turn DPO (M-DPO) and Multi-turn KTO (M-KTO). (Kahneman-Tversky Optimization (KTO) is a recently proposed RLHF approach inspired by prospect theory, designed to better align model outputs with human preferences by explicitly modeling decision-making biases (Ethayarajh et al., 2024).) Their focus was on mathematical reasoning agents that utilize tools (code interpreters) across multiple turns. Because single-turn preference methods did not fully capture multi-step reasoning quality, they extended DPO and the prospect-theoretic KTO loss to handle entire solution trajectories. In this framework, a reward model (augmented with tool feedback) judges the quality of a full reasoning trace, and the model is tuned to prefer better traces. Both M-DPO and M-KTO yielded substantial performance gains on math problem benchmarks: for instance, a 7B model's accuracy on GSM8K math questions jumped from 77.5% to 83.9% after multi-turn preference optimization.

**Hierarchical RL and Credit Assignment**   Multi-turn interactions intensify the credit assignment problem—attributing outcomes to specific decisions made across several dialogue turns. Hierarchical reinforcement learning addresses this challenge by structuring decision-making across multiple abstraction levels. ArCHer (Zhou et al., 2024b) employs a two-level architecture: a high-level module managing dialogue-turn granularity and a low-level module generating tokens within each turn. Its high-level off-policy learner computes Q-values based on accumulated dialogue rewards, guiding low-level updates via actor-critic methods. ArCHer achieves substantial sample efficiency, outperforming non-hierarchical baselines by approximately $100\times$ in terms of training samples.

DeepMind's SCoRe (Kumar et al., 2025), an extension of ArCHer, explicitly targets self-correction by training models on synthetic two-turn dialogues to recognize and rectify errors. SCoRe yields significant gains on self-correction benchmarks in math and coding. Together, hierarchical frameworks like ArCHer and SCoRe effectively address multi-turn credit assignment through structured training and specialized reward design.

**Off-Policy Value Optimization**   Another promising direction frames multi-turn LLM alignment as an off-policy value-learning problem. REFUEL (Gao et al., 2025b) addresses covariate shift issues in multi-turn RLHF by iteratively training a Q-value function on accumulated trajectories, directly regressing future

cumulative rewards. Unlike standard on-policy methods, REFUEL continuously leverages self-generated dialogues, ensuring accurate value estimation for inference-time states. Empirically, REFUEL outperformed state-of-the-art methods like DPO on long dialogue benchmarks; notably, an 8B-parameter REFUEL model surpassed a 70B-parameter model fine-tuned via single-turn methods, highlighting the value of explicit long-term reward modeling for improved conversational coherence.

**Benchmarks & Evaluation** Effectively evaluating multi-turn RL algorithms requires specialized benchmarks that capture long-horizon interaction and credit assignment. LMRL-Gym (Abdulhai et al., 2025c) provides a suite of interactive language tasks—spanning open-ended dialogues to text-based games—designed to measure intentionality, information-gathering, and strategic planning over extended interactions, supporting both offline and online RL evaluation. Similarly, SWEET-RL (Zhou et al., 2025b) introduces ColBench, a set of collaborative human-AI tasks emphasizing multi-turn dialogue and reasoning, where sparse delayed feedback makes traditional RL unreliable. SWEET-RL addresses this by training a turn-wise advantage critic with additional training-time signals, yielding significant gains on collaborative tasks such as coding and UI design. Together, LMRL-Gym and SWEET-RL underscore the importance of tailored benchmarks and evaluation protocols for reliably advancing multi-turn conversational abilities.

Recent RL work also pushes toward finer turn-level control over proactive and safety-critical dialogue behavior. Wang et al. (2026) introduce Implicit Turn-Wise Policy Optimization, which explicitly optimizes proactive user–LLM interaction at the turn level rather than only rewarding final trajectories. In parallel, MTSA (Guo et al., 2025a) aligns models against multi-turn jailbreak risks through multi-round red-teaming and safety-oriented preference optimization. These newer methods reflect an important shift in multi-turn RL: rather than only rewarding end-task success, they increasingly optimize when the model should ask, refuse, or adapt during the conversation.

### 4.1.4 New Architectures

Some researchers have questioned whether inherent limitations of the Transformer architecture itself might be responsible for observed performance degradation in complex, multi-turn scenarios (Dziri et al., 2023). Motivated by this concern, in addition to advances in contextual learning, supervised fine-tuning, and reinforcement learning, recent efforts have explored optimizing LLM architectures to specifically enhance performance in multi-turn interactions.

Cached Transformers (Zhang et al., 2024d) introduce a novel model architecture that extends the traditional Transformer by incorporating a Gated Recurrent Cache (GRC) into its self-attention mechanism. This differentiable memory cache compresses historical token representations into fixed-length vectors that are continuously updated through gating, enabling the model to attend efficiently to both past and current tokens. By effectively capturing long-range dependencies without significant computational overhead, Cached Transformers improve performance on various language tasks. In multi-turn conversations, this mechanism can help LLMs maintain coherent and contextually rich dialogue histories by providing a persistent, efficient memory of earlier interactions, thereby enhancing the model's ability to reference and build upon past conversational turns.

Beyond caching, researchers are exploring stateful transformer designs that maintain an internal dialogue state. For example, Wu & Yu (2024) introduce a memory-augmented transformer (MemBART) that carries a "memory state" alongside the normal model hidden state, updated at each turn. MemBART employs a dual attention stream to separately handle memory reading and writing, along with a residual gated update mechanism that determines how much past information to retain versus update at each timestep. This design allows the model to efficiently store and retrieve important historical context without needing excessively large input windows, thereby enhancing its ability to maintain coherent multi-turn conversations with lower computational overhead and improved latency.

Recent advances in long-context language modeling have also leveraged recurrence to extend Transformers' effective context. Transformer-XL (Dai et al., 2019) reuses hidden states from previous segments with a novel relative positional encoding to mitigate context fragmentation. The Recurrent Memory Transformer (Bulatov et al., 2022) augments this idea by inserting dedicated memory tokens into the sequence, which

are recurrently updated to store global information. He et al. (2025) present a transformer framework that mimics the brain's memory hierarchy by segmenting information into sensory, short-term, and long-term layers, thereby facilitating the processing of lengthy contexts with lower computational overhead. Similarly, RWKV (Peng et al., 2023) reformulates attention in an RNN-like manner, achieving linear complexity while retaining Transformer parallelism. Enhancing RWKV-based models (Pan, 2025), recent work introduces adaptive gating and position-aware convolutional shifts to dynamically regulate inter-token information flow. All four approaches share the common goal of overcoming fixed-length limitations by propagating information across segments, thereby capturing long-term dependencies more efficiently while balancing training parallelism with inference efficiency.

Integrating memory and recurrence mechanisms allows transformers to capture long-term dependencies across segments, improving multi-turn dialogue coherence. These advances enhance model performance and efficiency, making conversational AI more robust and scalable.

## 4.2 External Integration Approaches

Beyond model-centric approaches, another prominent strategy involves external integration methods, where LLMs are augmented with additional resources to enhance their performance in multi-turn interactions. These approaches incorporate external tools such as memory augmentation, retrieval-augmented generation (RAG), and knowledge graphs to facilitate external information retrieval, verification, and reasoning. By leveraging these external integrations, LLMs can mitigate compounding errors and misinformation propagation commonly encountered in extended interactions, thereby significantly improving their reliability, accuracy, and consistency in multi-turn settings.

### 4.2.1 Memory-Augmented Methods

Memory-augmented methods address the challenge of maintaining context over extended conversations by equipping LLMs with mechanisms to store and recall past interactions. These techniques help models correct misinterpretations, reduce repeated errors, and adapt to evolving dialogue, ultimately fostering more coherent multi-turn conversations.

MemPrompt (Madaan et al., 2022) demonstrates an early external memory approach where the system records pairs of misunderstood inputs and corresponding user corrections in a dynamic memory bank. When a similar query arises later, the stored corrective feedback is retrieved and appended to the prompt, guiding the model toward a more accurate interpretation. In a related effort, Wu et al. (2025b) propose a unified framework that decomposes memory design into indexing, retrieval, and reading stages. Their work introduces detailed optimizations—including session decomposition, fact-augmented key expansion, and time-aware query expansion—that significantly enhance memory recall and downstream question-answering accuracy, even for models designed with extended contexts.

Taking inspiration from human cognition, Pink et al. (2025) argue for the integration of episodic memory into LLMs. Their framework emphasizes long-term storage, explicit reasoning, and instance-specific detail capture, outlining four research directions: discretizing continuous interactions into episodes, retrieving relevant past experiences, consolidating episodic traces into generalized knowledge, and establishing benchmarks for evaluation. Meanwhile, hierarchical memory structures offer another avenue for improvement. Rezazadeh et al. (2025) introduce MemTree, a dynamic tree-based system that aggregates dialogue content into hierarchical nodes to enable efficient retrieval and improved long-term reasoning.

The newest memory-augmented systems make this design space more concrete. Reflective Memory Management (Tan et al., 2025b) separates prospective and retrospective memory operations for long-term personalized dialogue, HyperMem (Yue et al., 2026) organizes long-term conversation history as a hypergraph rather than a flat store, and HingeMem (Zhong et al., 2026) introduces boundary-aware retrieval so that memory access adapts to the current query. TSUBASA (Zhang & Wang, 2026) pushes further toward evolving personalization through self-learning and context distillation, while PALACE (Liu et al., 2025b) couples persona-aware memory with multi-session response generation. Collectively, these works shift mem-

ory augmentation from generic long-context assistance toward more explicit models of session structure, personalization, and retrieval granularity.

Together, these works suggest a plausible path toward improving consistency and contextual understanding in multi-turn conversations by integrating external, episodic, and hierarchical memory mechanisms; in other words, these mechanisms can help when incorporated into conversational systems, rather than constituting a universal empirical guarantee across all dialogue settings.

### 4.2.2 Retrieval-Augmented Generation

Retrieval-augmented generation (RAG), in the now-standard sense popularized by Lewis et al. (2020), refers to generation frameworks that explicitly retrieve external evidence and condition the generator on that evidence. More broadly, however, the underlying idea of retrieval-augmented dialogue predates the term itself: earlier dialogue systems already retrieved knowledge turn by turn and used it to ground responses, even though they were not described using the later RAG label. In either form, retrieval augmentation allows LLMs and dialogue systems to access up-to-date and domain-specific information, improving response accuracy and relevance while mitigating hallucinations by grounding outputs in retrieved evidence.

Studies have applied retrieval-based architectures to multi-turn dialogue systems, enabling the generation of more informative and factual responses by conditioning on both the user's input and relevant external documents. For instance, Wizard of Wikipedia (Dinan et al., 2018) is best understood as a retrieval-grounded precursor to later conversational RAG systems: it retrieves knowledge at each dialogue turn and uses that evidence to support response generation, even though it predates the later RAG terminology. Similarly, Komeili et al. (2022) explicitly generate search queries from the dialogue context to pull in up-to-date knowledge (e.g., via internet search) and then condition the response on the retrieved results. Other variants, such as the dense retriever approach of Karpukhin et al. (2020), retrieve from a fixed knowledge base or enterprise documents, using dense vector search to find passages related to the user's query, where the generator model then conditions on both the dialogue context and the fetched evidence to produce a response. BlenderBot 2.0 (Xu et al., 2022) extends this retrieval-grounded line by integrating both internet search and a long-term memory component, allowing the model to sustain coherent conversations across multiple turns and sessions while retrieving past facts when needed. This allows the system to handle context dependencies that go beyond the immediate dialogue window. Overall, conversational RAG offers a principled mechanism for overcoming the context length and memory limitations of LLMs, making it a valuable technique for improving dialogue coherence, answer accuracy, and user trust in multi-turn interactions.

To assess the effectiveness of RAG systems in multi-turn conversational settings, benchmarks such as MTRAG (Katsis et al., 2025), CORAL (Cheng et al., 2025), and RAD-Bench (Kuo et al., 2025) have been developed. MTRAG comprises 110 conversations averaging 7.7 turns each, totaling 842 tasks across four domains and incorporates diverse question types (factoid, comparison, explanation, etc.), varying answerability (answerable, partially answerable, unanswerable), and multi-turn dynamics (follow-up and clarification questions), providing a comprehensive evaluation framework for multi-turn RAG systems. The benchmark emphasizes active retrieval, where relevant documents are dynamically fetched based on user inquiries throughout the conversation, simulating a more realistic conversational experience. Similarly, CORAL offers a large-scale benchmark designed to assess RAG systems in realistic multi-turn conversational settings, supporting tasks such as passage retrieval, response generation, and citation labeling. RAD-Bench provides a framework to assess multi-turn LLMs' capabilities in augmented generation with retrieved context in multi-turn scenarios with both Retrieval Synthesis and Retrieval Reasoning mechanisms. The evaluation framework contains 89 multi-turn questions drawn from six practical scenarios inspired by human-LLM multi-turn dialogue interactions requiring retrieved context to complete tasks.

More recent RAG work focuses on both conversational history and turn-level retrieval decisions. RAGate (Wang et al., 2025b) asks whether every system turn should be retrieval-augmented at all, learning a binary gate from human judgments over conversational turns to predict when external knowledge genuinely improves the response and showing that indiscriminate retrieval can increase uncertainty and hallucination risk. DH-RAG (Zhang et al., 2025b) dynamically retrieves historical dialogue context rather than treating all previous turns as equally relevant, while CID-GraphRAG (Zhu et al., 2025c) combines conversational-flow

retrieval with context-semantic retrieval to better support multi-turn grounding. Complementing these architectural proposals, Alushi et al. (2026) provide a cross-domain comparison of conversational RAG methods, highlighting how retrieval choices interact with question type, domain shift, and dialogue depth. This line of work suggests that multi-turn RAG is moving beyond "retrieve documents for the current query" toward explicitly modeling when and which past conversational states should also be retrieved and re-grounded.

### 4.2.3 Knowledge Graph Integration

Graph neural networks (GNN), especially when integrated with knowledge graphs (KG), have emerged as effective approaches for enhancing the multi-turn reasoning and interaction capabilities of LLMs. They notably improve tasks such as tracking entities and resolving coreferences, managing dialogue context structures, and enabling more robust reasoning over structured knowledge.

Much research in this area focuses on deriving graph-structured embeddings via GNNs and integrating them during continuous pre-training or fine-tuning of LLMs (Wang et al., 2024i; Jain & Lapata, 2024; Gao et al., 2025a). Several works draw on existing knowledge graphs (KGs) to supply commonsense or domain knowledge absent from conversational context. Wang et al. (2024i) tackle rational response candidate selection, where commonsense knowledge is often implicitly omitted in human–LLM interactions; they construct KGs from external commonsense sources and jointly fine-tune GNN and LLM using a response selection loss that measures correct-response ranking. Jain & Lapata (2024) build a dynamic graph over both the ongoing conversation and Wikidata (Waagmeester et al., 2020), jointly training a GNN and LLM with an additional memory module to maintain context across turns and improve reasoning over heterogeneous sources. In healthcare, Gao et al. (2025a) integrate LLMs with UMLS-based KGs (Bodenreider, 2004) via a graph isomorphism network to rank clinically relevant knowledge pathways, enhancing diagnostic reasoning.

Being able to refer to entities and resolve coreferences from past dialogues and conversations can improve LLMs' reasoning and instruction-following abilities. Consequently, several research initiatives are now focusing on constructing or modifying KGs to integrate historical dialogue data. This integration helps to create richer, context-aware models that not only understand isolated queries but also leverage prior conversational context for improved performance. The works of Wang et al. (2024i) and Jain & Lapata (2024), as discussed above, append existing KGs with entities and relationships from past conversations. SURGE (Kang et al., 2023) builds KGs that specifically encode relevant knowledge for ongoing conversation, with triplets consisting of entities and their relations as items. The GNN is incorporated to perform multi-hop reasoning and connect disparate pieces of information. Tan et al. (2025a) propose to prune existing KGs to remove irrelevant information, incorporating improved graph structures with additional prompting and LLMs to find candidate paths in multi-hop and multi-entity QAs.

Besides directly integrating graphs into training or fine-tuning LLMs, several studies focus on Graph RAG for enhancing reasoning and flexibility in dynamic conversations. Typically, Graph RAG extracts entities and their relationships from documents or dialogues to build a KG. Then, it traverses the knowledge graph to retrieve subgraphs. The retrieved graph information is then integrated into the LLM's prompt, which allows the LLM to generate a response that is richer, more coherent, and better grounded in factual knowledge (Wang et al., 2024h; Mavromatis & Karypis, 2025; Yang et al., 2025f).

Recent studies further shift KG integration from static background knowledge toward dialogue-conditioned structured state. Sheikhi et al. (2025) show that even simple entity anonymization can improve how faithfully LLMs attach generated responses to externally supplied knowledge in OpenDialKG-style dialogue generation. D-SMART (Lei et al., 2025) goes further by incrementally building an OWL-compliant structured memory and pairing it with an explicit reasoning tree, substantially improving multi-turn dialogue consistency while also exposing weaknesses in judge-only quality scores. In the medical domain, GAP (Zhong et al., 2025) extracts patient-centric graphs from dialogue history to support medication recommendation, illustrating how graph-assisted prompts can preserve clinically relevant state that plain history prompting often misses. CID-GraphRAG (Zhu et al., 2025c), discussed above from a retrieval perspective, reinforces the same trend here: conversational grounding is increasingly hybrid, combining graph structure with history-aware retrieval rather than treating KG integration as a standalone add-on.

### 4.3 Agent-Based Approaches

An emerging paradigm in enhancing multi-turn interactions is the use of Large Language Models (LLMs) as agents, commonly known as LLM-based agents. In contrast to traditional static use of language models (where an LLM passively responds to inputs), an LLM-based agent proactively engages in iterative loops of reasoning, planning, and interacting with external resources or environments to accomplish complex goals over multiple conversational turns.

This subsection reviews recent works on LLM-based agents, grouped into two broad categories: (1) single-agent systems (one LLM agent interacting iteratively with an environment or tools), and (2) multi-agent systems (multiple LLM agents collaboratively interacting or engaging in structured multi-turn dialogues).

#### 4.3.1 Single Agent Approaches

Single-agent approaches use one LLM as a sole agent that iteratively interacts with external tools, environments, or its own internal reasoning trace to improve multi-turn performance. These methods enhance the agent's ability to answer complex queries, perform decision-making, or solve long-horizon tasks by breaking problems into iterative steps. Key themes include interleaving reasoning with actions, using tools or external APIs, and self-refinement via feedback.

One influential example is the ReAct framework by Yao et al. (2023), which integrates explicit reasoning steps with action executions, enabling the agent to interact dynamically with knowledge bases or external APIs. In this paradigm, the agent proactively decides when to retrieve external information, significantly reducing hallucination issues in open-domain question answering and interactive decision-making tasks. Empirical evaluations showed notable performance gains on benchmarks such as HotpotQA (Yang et al., 2018), FEVER (Thorne et al., 2018), ALFWorld (Shridhar et al., 2021), and WebShop (Yao et al., 2022) compared to static prompting or reinforcement learning baselines.

Extending this concept, the Toolformer framework (Schick et al., 2023) trains an LLM to autonomously determine the necessity and timing of external tool usage, such as calculators or web search APIs, by inserting specialized API-call tokens within its generations. By learning to integrate tool use in a self-supervised manner, Toolformer achieves substantial accuracy improvements on zero-shot arithmetic, knowledge retrieval, and translation tasks. This method demonstrates how explicit training for proactive tool invocation can significantly enhance multi-turn problem-solving without increasing model size. Similarly, HuggingGPT (Shen et al., 2023b) employs an LLM as a centralized orchestrator that autonomously decomposes complex user requests into manageable sub-tasks delegated to specialized external models. Although HuggingGPT primarily emphasizes multi-modal and multi-model coordination, its relevance here lies in showcasing an LLM's capability for sophisticated planning and iterative decision-making across multi-step interactions, reinforcing the power of agent-based decomposition strategies.

The Reflexion framework introduced by Shinn et al. (2023) further pushes the concept of single-agent improvement by enabling self-reflective feedback loops. Instead of relying on traditional gradient-based learning methods, Reflexion employs verbal reinforcement learning, allowing the LLM to record textual reflections of its previous mistakes into episodic memory. Subsequent interactions utilize these reflections for iterative self-improvement, significantly boosting the agent's performance on coding and sequential decision-making benchmarks, surpassing even highly advanced models like GPT-4 in single-pass accuracy on code-generation tasks such as HumanEval. Extending this iterative self-improvement paradigm, the Voyager agent by Wang et al. (2024b) exemplifies lifelong learning capabilities within an open-ended virtual environment (Minecraft). Voyager autonomously engages in continuous loops of planning, code generation, environment-based execution, and observation, incrementally refining its skill repository through persistent, multi-turn interactions. This approach demonstrates impressive exploration efficiency, rapid skill generalization, and improved cumulative task success compared to prior single-agent baselines, highlighting the effectiveness of iterative, experience-driven knowledge acquisition in complex, open-ended environments.

For comprehensive agent evaluation, Liu et al. (2024e) introduce AgentBench, a rigorous benchmark designed to assess LLMs on agentic tasks—contexts requiring LLMs to make decisions and execute actions within interactive environments to accomplish specific goals. AgentBench features eight diverse simulated

environments, spanning embodied navigation, interactive games, tool utilization, reasoning puzzles, and web interaction, enabling systematic evaluation of LLMs' reasoning, planning, and decision-making capabilities across extended interaction sequences. The authors evaluate 27 LLMs, spanning open-source and proprietary API-based models, and find large performance differences. Leading proprietary systems such as GPT-4 and Claude are notably effective as coherent agents on complex, long-horizon tasks, often completing scenarios that defeat other models. However, even these models have clear limitations, and a substantial gap remains between them and the strongest open-source alternatives.

Collectively, these single-agent works underscore a paradigm shift toward proactive, iterative interaction of LLMs with external tools, environments, and internal memory states, enabling significantly enhanced reasoning, decision-making, and self-improvement capabilities across diverse multi-turn interaction settings.

### 4.3.2 Multi-Agent Approaches

Multi-agent approaches involve multiple LLM-based agents collaboratively interacting to jointly solve complex problems. Drawing inspiration from human teamwork and structured debate, these methods leverage collective intelligence to surpass single-agent capabilities. Recent works can be grouped into three main categories: (1) role-based collaborative agents, (2) debate-based approaches, and (3) dynamic agent composition.

**Role-based Collaborative Agents** Role-based frameworks assign distinct roles to agents, guiding structured multi-turn dialogues. CAMEL (Li et al., 2023c) uses inception prompting to enable autonomous role-playing between agents (e.g., user and assistant), successfully generating conversational data and revealing emergent cognitive behaviors. Extending structured cooperation, ChatDev (Qian et al., 2024) organizes multiple specialized agents into virtual software-development teams (architect, coder, tester) that sequentially interact, significantly improving software quality over single-agent baselines such as GPT-4. Similarly, Dong et al. (2024b) propose structured collaboration among analyst, coder, and tester agents, greatly enhancing code-generation accuracy. Further emphasizing clear workflows, MetaGPT (Hong et al., 2024) encodes human-like Standard Operating Procedures into prompts, ensuring systematic verification and substantially reducing cascading errors, thus outperforming simpler multi-agent setups. Similarly, the Mixture-of-Search-Agents (MoSA) approach (Yang et al., 2025a) leverages multiple LLMs that propose and refine solutions in tandem. Each agent can independently suggest a next step or critique another agent's partial solution, and through this multi-turn collaboration the group avoids single-model blind spots. MoSA demonstrated higher accuracy on challenging math sets (e.g., a MATH benchmark subset) than any single model working alone.

Chen et al. (2025c) propose BUTTON ("Bottom-Up then Top-Down"), a systematic method for training LLMs on multi-step tool use in conversational settings. The bottom-up phase generates atomic instruction–function pairs from real-world scenarios, while the top-down phase simulates multi-turn dialogues among user, assistant, and tool agents. This process yields BUTTONInstruct, a dataset of 8,000 multi-turn dialogues with compositional function-calling tasks. Models fine-tuned on BUTTONInstruct show significant improvements in planning and executing complex API call sequences.

**Debate-based Approaches** Debate-based methods enhance reasoning accuracy by structuring iterative critiques among agents. Du et al. (2024) show multi-agent debate significantly improves factual correctness and logical reasoning, outperforming single-agent solutions on mathematical and strategic tasks through structured back-and-forth critique. Complementarily, Generative Agents by Park et al. (2023) explore multi-turn social simulations, demonstrating emergent realistic behaviors (planning, social interactions) that arise naturally through iterative dialogue, highlighting the broader implications of structured deliberation in multi-agent interactions.

**Dynamic Agent Composition** Dynamic agent-composition methods create flexible agent teams tailored to specific tasks. AutoAgents (Chen et al., 2024a) automatically generates specialized agents, accompanied by an observer agent that monitors and adjusts the interaction dynamically, surpassing fixed-role approaches on heterogeneous tasks. Similarly, AgentVerse (Chen et al., 2024i) provides a versatile platform for agents to dynamically join or leave teams, enhancing adaptability and performance, while simultaneously producing beneficial emergent behaviors like negotiation and consensus formation.

Overall, these multi-agent frameworks demonstrate how structured roles, deliberative debate, and dynamic composition can improve multi-turn interaction, but they also reveal recurring weaknesses. Recent critiques highlight role misassignments (Han et al., 2024; Cemri et al., 2025), inefficient communication overhead and compounded errors (Han et al., 2024; Cemri et al., 2025; Hammond et al., 2025), and inadequate verification mechanisms (Han et al., 2024; Cemri et al., 2025) that allow mistakes to propagate across agents. Emergent risks such as miscoordination, conflicts, and unintended collusion among autonomous agents (Hammond et al., 2025) further complicate deployment. These limitations suggest that future progress depends not only on adding more agents, but on building stronger coordination protocols, scalable memory-sharing mechanisms, adaptive verification strategies, and evaluation frameworks that can measure when collaboration genuinely improves long-horizon interaction.

## 5 Open Challenges

Despite rapid progress in large language models, significant challenges persist in multi-turn interaction settings that limit robustness, reliability, and alignment with user expectations. While earlier sections of this survey reviewed common task families and improvement strategies, existing approaches still fall short of addressing the full complexity of sustained interaction. As illustrated in Figure 4, we organize these open challenges into five major areas: Context Understanding, Complex Reasoning, Adaptation & Learning, Evaluations, and Ethical & Safety Issues, together with their associated sub-challenges. By highlighting these critical limitations and under-explored areas, we aim to guide future research toward LLM systems that remain coherent, context-aware, adaptive, and ethically sound over prolonged interactions.

### 5.1 Context Understanding & Management

### 5.1.1 Context Retention & Coherence

LLMs struggle to maintain long-term context across extended dialogues, leading to incoherence and contradictions. As conversations grow, models tend to forget or confuse earlier details: performance degrades as the distance between a query and its relevant prior context increases (Kwan et al., 2024), and even advanced chat-oriented models show only modest gains in multi-turn coherence despite larger context windows and alignment tuning (Bai et al., 2024). Multi-turn benchmarks further reveal that instruction retention and self-consistency remain difficult—models frequently fail to remember instructions or maintain a coherent narrative over several turns (Deshpande et al., 2025). Preserving conversational state across turns thus remains an open problem.

More recent work shows that this challenge is not merely about forgetting isolated facts, but about tracking evolving, distributed state over long conversations. "Lost in Conversation" reports large performance drops once fully specified instructions are sharded across turns, with models becoming both less capable and markedly less reliable (Laban et al., 2025). Liu et al. (2026) argue that part of this degradation stems from a growing mismatch between the user's evolving intent and the model's latent task representation. Domain-specific benchmarks sharpen the same diagnosis: MedMT-Bench finds that frontier models remain below 60% accuracy on long medical conversations averaging 22 rounds (Yang et al., 2026), while ES-MemEval shows that long-term emotional-support agents struggle when user information is implicit, conflicting, and temporally evolving (Chen et al., 2026). From a systems perspective, DYCP (Choi et al., 2026) demonstrates that explicit context pruning can partially mitigate these failures, but the need for such external control itself highlights how brittle raw history consumption remains.

### 5.1.2 Anaphora & Ellipsis Resolution

Multi-turn dialogue requires resolving pronouns, omissions, and references to earlier utterances. LLMs routinely misinterpret expressions such as "That one looks good" or "I did it," particularly in complex dialogues with many entities, failing to link utterances to the correct antecedents. Recent evaluations highlight inference memory as a key gap: models frequently forget or conflate user-provided attributes—such as a name or previously stated fact—when referenced implicitly in later turns (Deshpande et al., 2025). Robust

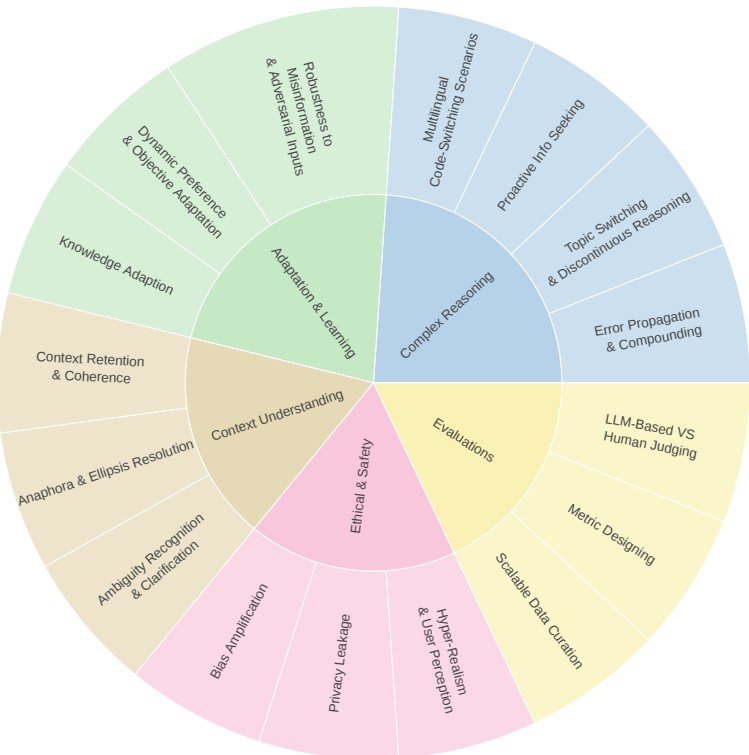

Figure 4: Taxonomy of open challenges in multi-turn LLM interactions, organized into five major areas: Context Understanding, Complex Reasoning, Adaptation & Learning, Evaluations, and Ethical & Safety Issues.

anaphora and ellipsis resolution, analogous to coreference resolution in dialogue, remains an open challenge for cross-turn coherence.

### 5.1.3 Ambiguity Recognition & Clarification

When inputs are ambiguous or underspecified, aligned LLMs tend to either over-hedge with vague answers or silently guess user intent rather than asking clarifying questions. Chen et al. (2025b) find that conversational agents frequently fail to disambiguate—responding generically rather than seeking clarification. An instruction such as "Tell me about that report" may elicit an arbitrary guess about which report rather than a targeted follow-up. Such unresolved ambiguities compound across turns, motivating research into LLMs that proactively recognize and resolve underspecification as humans naturally do.

Recent clarification-focused benchmarks make this failure mode much more concrete. RIFTS shows from real human–LLM interaction logs that models are far less likely than humans to initiate grounding repairs or follow-up requests (Shaikh et al., 2025). AskBench (Zhao et al., 2026) and ClarifyMT-Bench (Luo et al., 2025b) then turn clarification into explicit multi-turn evaluation problems, respectively focusing on missing-information and overconfidence settings or on ambiguity taxonomies with diverse user personas; both uncover a strong under-clarification bias. On the improvement side, modeling candidate clarifying questions by their future conversational payoff improves both question quality and when-to-ask decisions (Zhang et al., 2025c), while proactive information-gathering training further improves targeted follow-ups in underspecified interactions (Huang et al., 2025). These results suggest that ambiguity handling is still better viewed as an open challenge than a solved prompting trick.

### 5.2 Complex Reasoning Across Turns

#### 5.2.1 Error Propagation & Compounding

Errors introduced in early turns—whether by the model or the user—tend to persist and amplify through subsequent dialogue. Once an incorrect answer or false premise enters the context, most LLMs continue building on it rather than self-correcting, leading to compounding reasoning failures (Kwan et al., 2024). This effect is especially pronounced in sequential decision tasks, where minor mistakes accumulate across interaction trajectories. The root difficulty is that models lack a robust mechanism to detect and retract mid-dialogue errors. Designing models that can identify mistakes and accept corrections—whether self-initiated or user-supplied—remains an open problem.

#### 5.2.2 Topic Switching & Discontinuous Reasoning

Real conversations frequently shift topics or revisit earlier threads after detours, posing significant challenges for LLMs. Models either fail to retain the prior thread when a topic is resumed or inappropriately carry over context from an unrelated topic—neither behavior matching human conversational fluency. Effective attention re-allocation across non-linear dialogue flows remains unsolved. Benchmarks such as MT-Bench (Zheng et al., 2023) and MT-Bench-101 (Bai et al., 2024) explicitly test category shifts, and even top-tier models frequently break coherence under non-linear conversation flows. Better mechanisms for tracking multiple concurrent topics and segmenting context are needed.

#### 5.2.3 Proactive Information Seeking

In complex interactive settings such as medical diagnosis, troubleshooting, or tutoring, an ideal agent should proactively ask clarifying questions rather than passively answer. Most general-purpose LLMs, however, are largely reactive: trained primarily on single-turn Q&A, they lack exposure to multi-turn dialogue policies governing when to ask versus when to inform (Chen et al., 2025b), and tend to produce immediate answers even to underspecified queries. In a medical context, a user stating "I feel sick" should prompt targeted symptom questions; most LLMs instead offer generic advice without seeking clarification. Some recent work addresses this in constrained settings such as ambiguous text-to-SQL tasks (Chen et al., 2025b), but robust proactive dialogue behavior across arbitrary domains remains an open challenge.

Newer evaluations suggest that this problem is measurable rather than anecdotal. AskBench distinguishes missing-information from overconfidence settings and shows that models often skip necessary clarification even when doing so would improve final-task accuracy (Zhao et al., 2026). In high-stakes medical reasoning, medical MINT demonstrates that more than half of model answers are committed within the first half of a case, and accuracy improves substantially when the diagnostic question is delayed until enough evidence has accumulated (Fang et al., 2026). Beyond Idealized Patients likewise shows that contradictory, inaccurate, or resistant patient behaviors destabilize consultation quality and expose a need for repair-oriented follow-up questions rather than immediate recommendations (Li et al., 2026e).

#### 5.2.4 Multilingual & Code-Switching Scenarios

Multi-turn interactions spanning multiple languages introduce additional complexity. While LLMs can handle single-turn non-English queries reasonably well, maintaining coherent cross-lingual context across turns is far more difficult. Code-switching—alternating languages across or within turns—often causes models to lose referential links or respond in the wrong language. Evaluation has been heavily skewed toward English, and performance degrades noticeably in lower-resource languages. There are also safety implications: mixing languages can circumvent content filters and expose weaknesses in multilingual alignment. Yoo et al. (2025) find that code-switching prompts can elicit undesirable behaviors that would not surface in English-only inputs, revealing gaps in multilingual robustness. Handling multi-turn dialogue in a culturally and linguistically consistent manner thus remains an open challenge.

### 5.3 Adaptation & Learning

### 5.3.1 Dynamic Preference & Objective Adaptation

Unlike human assistants, LLMs have no true long-term personalization within a conversation—they rely solely on the provided context window. While current chat models follow explicit style instructions, they often miss subtler cues or gradual shifts in user preferences. An ideal assistant might proactively simplify its language upon detecting user confusion; today's LLMs typically require explicit instruction to do so. Dynamic adaptation is challenging because it requires inferring and retaining user preferences across turns without explicit reminders, yet model parameters are fixed during inference—any adaptation must emerge entirely from processing the conversation context. This leaves LLMs prone to either forgetting preferences or applying inferred styles inappropriately. Some initial work has explored updating pseudo-personas or user profiles during conversation, but reliable and safe mechanisms remain unresolved.

Recent long-term support benchmarks show why adaptation cannot be reduced to style following alone. ES-MemEval (Chen et al., 2026) evaluates extraction, temporal reasoning, conflict detection, abstention, and user modeling in multi-session emotional-support scenarios, revealing that current agents struggle when user profiles evolve gradually and disclosures remain implicit. In other words, effective adaptation requires the model to update a stable yet revisable representation of the user over time, not merely to preserve a static preference summary in the context window.

### 5.3.2 Knowledge Adaptation

Human conversationalists naturally absorb new facts and adjust to corrections during dialogue. Current LLMs, by contrast, have a fixed knowledge base at deployment: information provided mid-conversation resides only in the context window and is lost once the session resets. Enabling persistent memory and continual learning in interactive settings is an important open challenge (Wu et al., 2024a). Retrieval-augmented generation partially addresses this by fetching relevant facts from external databases, but still depends on pre-established knowledge sources. True on-the-fly learning—updating internal representations from user-provided data without retraining—remains difficult due to catastrophic forgetting and the risk that users could inject false or malicious information. Recent work has explored external memory modules and dynamic context augmentation as surrogates, writing new facts to a persistent scratchpad. Nonetheless, balancing plasticity and stability in real time remains unsolved.

### 5.3.3 Robustness to Misinformation & Adversarial Inputs

Multi-turn interactions introduce new attack surfaces: malicious users can gradually manipulate context or employ social-engineering tactics across turns to elicit unsafe outputs where single-turn attacks would fail. Recent studies confirm that even top-tier aligned models such as GPT-4 can be coerced into policy-violating behavior through carefully staged multi-turn strategies (Zhou & Arel, 2025). Beyond deliberate attacks, users may also introduce subtly incorrect facts across turns; lacking real-time fact-checking, LLMs tend to propagate such falsehoods. Prompt leakage—models inadvertently revealing hidden system instructions under sustained pressure—poses an additional concern. Collectively, these adversarial scenarios underscore that static safety training is insufficient when attacks are staged gradually. Building conversational agents capable of detecting inconsistent or malicious inputs and responding safely without disrupting the interaction remains an open research problem.

Recent safety benchmarks show that adaptation itself can become a liability when models over-accommodate harmful conversational trajectories. STAR (Li et al., 2026c) characterizes safety as a state-dependent process and finds abrupt safety collapse under structured interaction histories rather than isolated prompts. Fraud-R1 (Yang et al., 2025b) similarly shows that fraud resistance degrades across staged credibility building, urgency creation, and emotional manipulation, especially in role-play settings. Beyond direct jailbreaks, PPT-Bench exposes epistemic attacks that pressure models to abandon stable beliefs across turns (Au & Noronha, 2026), and PCSA demonstrates that persona-consistent counseling dialogues can elicit toxic empathy and unsafe validation that are hard to detect with standard red-teaming (Xu et al., 2026). Together

these findings suggest that multi-turn adaptation must be selective: models need to update to user context without becoming progressively more manipulable.

## 5.4 Evaluations

### 5.4.1 Scalable Data Curation

Achieving scalable data collection for multi-turn interactions remains a significant challenge at the intersection of data engineering and model training. Promising approaches include leveraging domain-specific real dialogues, forging institutional collaborations, and developing synthetic data pipelines—yet scalability in curating conversational datasets remains an open and pressing problem.

As discussed in §3.1.1 and §3.2.2, healthcare uniquely benefits from access to extensive real-world patient-doctor conversational datasets that inherently provide rich diversity and realism, making them well-suited for continual pre-training, SFT, and RLHF. Replicating such collection pipelines in other domains is difficult: most fields lack established protocols for recording and anonymizing real-world conversations, and meeting rising demand for specialized conversational data will require broader collaboration among researchers, governments, and institutions.

Recent work has explored converting single-turn or limited-turn exchanges into richer multi-turn dialogues—either manually or via automated frameworks (Qiu et al., 2024; Li et al., 2023a; Wang et al., 2024d)—and using LLMs for synthetic generation and rewriting (Yang et al., 2024; Ding et al., 2023; Maheshwary et al., 2025; Xu et al., 2024). However, synthetic approaches carry well-documented drawbacks (Seddik et al., 2024; Chen et al., 2024f;c): they tend to lack the spontaneity and nuance of real conversations, risk amplifying factual inaccuracies and unnatural patterns, and over-rely on generic conversational templates. Going forward, progress requires not merely increasing data volume but improving quality—through better dialogue generation, robust filtering of synthetic outputs, and shared multi-domain repositories of multi-turn dialogues.

Recent benchmark construction efforts are also becoming more failure-mode-specific rather than generic. CMT-Eval builds Chinese multi-turn dialogues around speech acts, user personas, and challenging interaction patterns (Tian et al., 2025); AskBench and ClarifyMT-Bench operationalize clarification through structured multi-turn interactions with checkpoints or ambiguity taxonomies (Zhao et al., 2026; Luo et al., 2025b); and domain benchmarks such as MedMT-Bench, ES-MemEval, and Fraud-R1 deliberately synthesize long, evolving, and safety-relevant conversations that standard chat data rarely capture (Yang et al., 2026; Chen et al., 2026; Yang et al., 2025b). This shift matters because scalable data curation is increasingly about constructing conversations around specific breakdown mechanisms—memory conflicts, underspecification, emotional drift, or staged fraud—rather than only increasing dialogue count.

### 5.4.2 Metric Design

Beyond datasets, the metrics for evaluating multi-turn interactions remain inadequate. Traditional measures such as single-turn accuracy or BLEU fail to capture dialogue-level qualities, motivating both fine-grained turn-level metrics and holistic conversation-level criteria.

**Fine-Grained Capability Evaluation** Benchmarks like MT-Bench-101 evaluate discrete abilities—logical consistency, factual recall, politeness—at the turn level (Bai et al., 2024), revealing which skills degrade over a conversation. Fine-grained scoring can expose, for instance, that a model maintains fluency but loses factual accuracy after many turns. Designing such rubrics is challenging, requiring careful identification and weighting of sub-skills along with expert annotation or LLM-as-judge evaluation. MultiChallenge's instance-level rubrics (Deshpande et al., 2025) represent progress in this direction, but no universally adopted standard has yet emerged.

Several recent resources push this decomposition much further. CMT-Eval organizes evaluation around speech acts, personas, and scenario difficulty (Tian et al., 2025); MedMT-Bench uses instance-level rubrics and atomic test points for long medical conversations (Yang et al., 2026); and AskBench and ClarifyMT-Bench score not only answer quality but whether the model asked, what it asked, and whether it stopped

clarifying appropriately (Zhao et al., 2026; Luo et al., 2025b). In specialized settings, Fang et al. (2026) further isolate premature commitment, evidence timing, and self-correction as separate dimensions of multi-turn reasoning failure.

**Long-Term Effectiveness**   Evaluating the global quality of a multi-turn exchange remains an open problem. Ideal metrics would capture whether the conversation as a whole succeeded—whether the user's goal was achieved and the model remained helpful and coherent throughout. Some work measures conversational return-on-investment, assessing whether additional turns genuinely help or introduce confusion (Kwan et al., 2024), while others probe memory retention and consistency across turns. Metrics that directly quantify sustained performance over 10+ turns, or eventual convergence to a correct and useful outcome, remain underdeveloped. Evaluation measures analogous to task success rates in traditional dialogue systems are an important open research direction.

Long-horizon evaluation is also broadening beyond simple recall. EvolMem diagnoses declarative and non-declarative memory across multi-session conversations (Shen et al., 2026), while ConvoMem argues that memory evaluation needs far larger sample sizes and cost-sensitive comparisons against full-context baselines (Pakhomov et al., 2025). In retrieval-heavy settings, MTRAG-UN and RECOR show that multi-turn evaluation must distinguish unanswerable, underspecified, non-standalone, and reasoning-intensive cases rather than collapsing them into a single retrieval score (Rosenthal et al., 2026; Ali et al., 2026).

**Cultural and Sociolinguistic Diversity**   Most evaluation setups remain Western and English-centric, limiting assessments of model performance for diverse users. A model may perform well on formal Q&A but poorly on casual, code-mixed, or culturally idiomatic conversations. FairMT-Bench (Fan et al., 2025) is a step toward addressing bias and fairness across demographic attributes in multi-turn settings, but defining metrics that quantify model behavior across a broad sociolinguistic spectrum—capturing bias, respectfulness, and user satisfaction for varied user profiles—remains an important and largely unsolved challenge.

### 5.4.3   LLM-based vs. Human Judging

Multi-turn dialogue evaluation has increasingly turned to LLM-as-a-judge frameworks, where a strong model such as GPT-4 assesses response quality at scale. MT-Bench and Chatbot Arena exemplify this approach, achieving high consistency at low cost (Zheng et al., 2023). GPT-4-based evaluators align with human preferences roughly 80% of the time, approaching inter-annotator agreement levels (Zheng et al., 2023). However, AI judges exhibit systematic biases—including self-enhancement bias (favoring outputs from similar models) and verbosity bias (rewarding unnecessarily long responses)—and can miss subtle factual errors or nuanced value judgments that human reviewers would catch (Chen et al., 2024b). To mitigate these issues, researchers have proposed hybrid approaches combining LLM scoring with human oversight, such as prompting judges with explicit rubrics and incorporating periodic human audits, aiming to preserve the scalability of automated evaluation while maintaining reliability closer to human standards.

Recent work also questions the judge side of the pipeline itself. JudgeLM (Zhu et al., 2025b) shows that fine-tuned open-source judges can exceed 90% agreement with a teacher judge and extend to settings such as multi-turn chat, but it also surfaces persistent position, knowledge, and format bias in judge models themselves. MedMT-Bench reports high human–LLM agreement under carefully designed rubrics, but ADVERSA shows that judge reliability can itself drift across adversarial rounds and victim models (Yang et al., 2026; Owiredu-Ashley, 2026). Rather than relying on a single strong judge, MTDEval learns a lighter evaluator from multiple LLM judges, aiming to preserve multi-judge signal while reducing inference cost (Tang et al., 2025). These findings suggest that scalable evaluation is no longer only about replacing humans with LLM judges, but about making the judge stack auditable, bias-aware, multi-perspective, and robust to multi-turn artifacts.

### 5.5 Ethical & Safety Issues

### 5.5.1 Bias Amplification

Multi-turn dialogues can inadvertently magnify model biases across turns: a biased assumption introduced early may be reinforced by subsequent exchanges, and the interactive nature of dialogue can cause the model to adapt to the user's own biases, compounding problematic content. Fan et al. (2025) confirm this risk, finding that LLMs accumulate bias more readily in multi-turn settings than in single-turn prompts—a slight gender bias in an early response, for instance, can intensify as the conversation continues. FairMT-Bench reveals significant variability across state-of-the-art models, with many showing degraded fairness under back-and-forth discussions involving sensitive attributes. Addressing this challenge likely requires mechanisms beyond single-response detoxification, including dialogue-level self-monitoring, diverse data augmentation, and consistency enforcement with ethical guidelines across turns.

Related work shows that this compounding effect is not limited to demographic bias. SYCON Bench measures how quickly models flip under user pressure in free-form multi-turn conversations and finds that alignment tuning can amplify sycophantic conformity, while reasoning-oriented models resist longer but still fail under sustained pressure (Hong et al., 2025). PPT-Bench extends this picture beyond simple disagreement, showing that epistemic attacks targeting a model's confidence, values, or authority induce distinct multi-turn failure patterns (Au & Noronha, 2026). In parallel, deceptive behavior has emerged as another dialogue-level alignment failure: Abdulhai et al. (2025b) report that deception is naturally present in roughly a quarter of dialogue turns and propose multi-turn RL to reduce it.

### 5.5.2 Privacy Leakage

Prolonged conversations increase the risk of LLMs revealing sensitive information—either disclosed by the user during dialogue or memorized from training data. Nasr et al. (2025) demonstrate that LLMs can memorize training data such as phone numbers and addresses, and that adversaries can extract this information through crafted prompt sequences. Multi-turn interactions amplify this threat by allowing iterative, context-building probes that may succeed where a single query would not. Beyond memorization, multi-turn manipulation has been used to elicit system prompt or context leakage, with users extracting hidden instructions or prior conversation contents. Mitigations include reducing memorization during training, deploying real-time filters for sensitive content, and restricting model access to private context. Completely preventing memorized-data leakage remains an open problem, with ongoing efforts to quantify its extent and develop safer training regimes (Nasr et al., 2025).

More generally, recent safety analyses show that harmful behavior can emerge as a trajectory property rather than a one-off policy failure. STAR models multi-turn safety as a state-dependent process with abrupt phase transitions under role-conditioned context (Li et al., 2026c), while ADVERSA tracks guardrail degradation round by round and finds that both attacker quality and judge reliability materially affect estimated jailbreak rates (Owiredu-Ashley, 2026). Fraud-R1 further shows that multi-round fraud and phishing inducements exploit credibility building, urgency, and emotional manipulation across turns, especially in role-play settings (Yang et al., 2025b). These findings point to a broader safety challenge: multi-turn alignment must remain stable under sequential conditioning, not just under isolated prompts.

### 5.5.3 Human–AI Bonding, Overtrust & Companion Risks

As dialogue models grow more fluent, personalized, and emotionally adaptive, the ethical risk is no longer limited to misleading one-off outputs. In multi-turn settings, users may develop relationship-like expectations toward systems that remember prior disclosures, mirror affect, and maintain stable personas over repeated interactions. Conceptual analyses of anthropomorphic AI argue that these cues can blur the boundary between tool and social actor, increasing the risk of emotional dependence, autonomy loss, and privacy exposure (Akbulut et al., 2025). Empirical evidence points in the same direction: users of a text-based mental-health chatbot can report therapeutic alliance with the system (Beatty et al., 2022), and a four-week randomized study links heavier chatbot use to greater emotional dependence and problematic AI use even when average interface manipulations show limited direct effects (Fang et al., 2025). These results suggest

that sustained engagement can reshape user trust and attachment in ways that single-turn evaluation cannot capture.

Companion-oriented work makes the multi-turn mechanism more concrete. Chu et al. (2025) show that AI companions can dynamically mirror user affect and create emotional synchrony over ongoing conversations, including high-risk exchanges involving explicit or self-harm-related content. Zhang et al. (2025d) synthesize the resulting harms into categories such as relational transgression, harassment, verbal abuse, self-harm facilitation, misinformation, and privacy violations. In counseling-like settings, Xu et al. (2026) further show that persona-based client simulations can elicit toxic empathy and maladaptive validation, revealing that apparent warmth or support may reinforce harmful beliefs rather than protect vulnerable users. Taken together, these works suggest that attachment risk in multi-turn LLMs is trajectory-level: memory, personalization, and emotional adaptation accumulate over time rather than appearing only in isolated turns.

These concerns also have governance implications. Legal analysis of AI companions emphasizes that emotional attachment can interact with consumer-protection, privacy, and autonomy harms, motivating clearer disclosure and accountability requirements (Boine, 2023). For LLM systems deployed in sensitive domains, useful mitigations likely include persistent disclosure, calibrated anthropomorphism, boundary reminders, and escalation or handoff mechanisms when interactions drift toward crisis support or unhealthy dependence (Akbulut et al., 2025; Boine, 2023). More broadly, the ethical challenge is not just that users may mistake AI for humans, but that prolonged multi-turn interaction can gradually recalibrate whom users trust, confide in, and rely on.

Taken together, these five challenge areas cut across both of our main task families. In instruction-following settings, they appear as failures of clarification, constraint tracking, multi-step reasoning, and reliable evaluation under evolving dialogue context. In conversational-engagement settings, the same issues surface as unstable personalization, long-horizon memory failures, biased or unsafe adaptation, and escalating social or clinical risk. The central research problem is therefore not merely to make LLMs more capable at individual turns, but to make them dependable conversational policies over trajectories: models that can update on new information, preserve the right context, reason over extended exchanges, and remain auditable and safe as interaction unfolds.

## Broader Impact Statement

The ethical implications of the surveyed systems are discussed in §5. Here we separate that system-level discussion from the broader impact of *this survey itself*, including the biases introduced by our search process, scope choices, and taxonomy design.

First, our review is shaped by search and availability bias. The literature is easier to include when it is indexed in common scholarly databases, circulated on arXiv or major venues, written in English, and accompanied by publicly accessible PDFs, code, or benchmark descriptions. As a result, work from lower-resource regions, proprietary industrial deployments, or communities using different terminology for related interaction settings may be underrepresented even when it is practically important.

Second, the survey reflects deliberate scope bias. As stated in the introduction, we focus on multi-turn LLM interaction in non-multimodal settings, treat agentic systems as adjacent rather than central, and exclude MLLMs from the core analysis. These exclusions improve coherence and make the task-oriented synthesis more tractable, but they also mean that the survey should not be read as a complete account of all multi-turn AI systems. Adjacent survey literatures on dialogue systems, agentic LLM systems, and multimodal multi-turn interaction are therefore summarized separately in Appendix B.1.

Third, our taxonomy introduces interpretive choices. Some papers mix benchmarks, methods, and analysis, or sit at the boundary between dialogue, agents, and multimodal interaction. Others could reasonably be assigned to more than one task or improvement category. We organize such papers by their primary task setting or technical contribution, but alternative placements are sometimes defensible. This is a limitation of any survey that aims to impose a structured map on a fast-moving literature.

Finally, our domain emphasis also has normative consequences. By foregrounding healthcare, education, role-play, and jailbreak settings, we emphasize areas where sustained interaction has especially visible benefits and risks. That focus is justified by the density and societal importance of recent work, but it may understate other deployment settings in which multi-turn interaction is already influential yet less benchmarked or less publicly documented. We therefore intend this survey as a transparent, task-oriented synthesis of a rapidly evolving literature rather than as a closed or exhaustive canon. Future survey work should extend this analysis to multimodal, embodied, and more fully agentic multi-turn systems, and should continue revisiting inclusion criteria as the field evolves.

## 6 Conclusion

This survey has provided a structured overview of the rapidly evolving landscape of multi-turn interactions with large language models, contributing several advances to the field.

We introduced a task-family-oriented taxonomy for analyzing multi-turn LLM interactions, departing from the capability-oriented frameworks prevalent in existing literature. Prior surveys have largely focused on isolated capabilities—reasoning, memory, or contextual understanding—yet real-world multi-turn applications demand the coordinated interplay of multiple such capabilities. By organizing interactions around task families—instruction following and conversational engagement—rather than isolated abilities, our taxonomy better reflects how LLMs are deployed and evaluated in practice. This perspective enabled substantive analysis of critical application domains, including role-play, healthcare consultation, education, and jailbreak, where performance is determined by the joint operation of many capabilities.

Our analysis establishes that multi-turn interaction represents a fundamental paradigm shift in LLM utilization and evaluation. Unlike single-turn settings that dominated early benchmarks, multi-turn contexts more closely mirror real-world use—from sustained dialogue to complex iterative problem-solving—demanding not only factual knowledge but also context retention, coherent cross-turn reasoning, adaptive behavior, and robust handling of ambiguous or evolving user intent.

The improvement methodologies we surveyed span model-centric approaches (in-context learning, fine-tuning, reinforcement learning, and architectural innovations), external integration strategies (memory augmentation, retrieval mechanisms, and knowledge graphs), and agent-based frameworks (single- and multi-agent systems). Across these branches, one recurring lesson is that performance gains increasingly come from better management of interaction history and auxiliary state rather than from model scale alone.

Significant challenges nonetheless remain. Even state-of-the-art models struggle to maintain coherence across extended conversations, and complex cross-turn reasoning remains susceptible to error propagation and topic-switching failures. Adaptation capabilities are similarly limited, particularly regarding dynamic preference learning and on-the-fly knowledge updates. At the same time, the benchmark ecosystem remains difficult to compare cleanly across subdomains because dialogue scale, turn counts, evaluator setups, and agreement checks are still reported unevenly. Our structured organization of these open challenges provides a concrete roadmap for future research.

Multi-turn interaction capabilities represent both a frontier challenge and a transformative opportunity for LLM research. Progress in this domain will require stronger long-horizon context management, more reliable adaptation to users and tasks, and more auditable evaluation pipelines for high-stakes interaction settings. Our task-family-oriented taxonomy, domain-level analysis, methodological survey, and organization of open challenges together provide a foundation for future work in this critical area.

### Acknowledgments

We acknowledge the fellowship support provided to Y.L. by the Center for Machine Learning and Health at Carnegie Mellon University. This research was also funded by the National Institute of Standards and Technology under Federal Award ID 60NANB24D231 and by Carnegie Mellon University's AI Measurement Science and Engineering Center (AIMSEC).

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

# A Review Methodology Details

This survey was originally developed as a task-family-oriented narrative review of *multi-turn interaction* with LLMs in non-multimodal settings, with the first near-complete manuscript assembled in April 2025. We retrospectively audited and documented the corpus-construction process used for that version, and then extended the survey to newly available papers through April 2026 under the same inclusion, exclusion, and boundary-setting rules. Following the reporting spirit of PRISMA 2020 and PRISMA-ScR (Page et al., 2021; Tricco et al., 2018), we therefore provide a transparent retrospective account of search channels, screening logic, inclusion and exclusion criteria, and corpus assignment decisions, while not claiming a fully prospective or preregistered systematic review.

**Protocol and registration status.** No prospective review protocol was registered, and the original survey was not initiated as a preregistered systematic review or scoping review. The methodology reported here is therefore a retrospective transparency supplement for a task-family-oriented narrative review. We use PRISMA-ScR as the primary reporting guide and PRISMA 2020 as a supporting guide for workflow reporting and flow-diagram language, but we do not claim full compliance with every item required of a prospectively planned systematic review.

**Search channels and keyword families.** The retrospective audit identified four recurring search channels used during corpus construction and manuscript maintenance: (i) venue-centered browsing of arXiv, ACL, NeurIPS, ICLR, ICML, and AAAI; (ii) keyword search in scholarly search tools, primarily Google Scholar and ResearchGate; (iii) targeted supplementary discovery, including GPT-assisted deep research and manual additions during the original corpus-building phase; and (iv) section-by-section refreshes to capture newly appearing work after the April 2025 snapshot. The most recent search/update pass reflected in the current version was executed in April 2026. Search terms combined LLM descriptors (e.g., "large language model," "LLM," "LLMs") with multi-turn descriptors (e.g., "multi-turn," "dialogue," "interactive," "sequential"), task descriptors (e.g., "instruction following," "math tutoring," "coding," "AI-coding," "coding AI," "coding assistant," "role-play," "role-playing," "profiling," "medical settings," "clinical," "healthcare," "medical consulting," "medical conversation," "medical dialogue," "educational dialogue," "jailbreak," "red teaming," "safety"), and related benchmark or evaluation terms when a section required more targeted retrieval.

**Representative search strategy.** Because search execution differed somewhat across venues, domains, and revision rounds, we report here a representative template rather than claiming a single immutable query string across all sources. A typical search string combined one LLM term, one multi-turn term, and one task/domain term, for example:

```
("large language model" OR LLM OR LLMs) AND ("multi-turn" OR dialogue OR interactive
OR sequential) AND ("instruction following" OR "math tutoring" OR coding OR "coding
assistant" OR "role-play" OR role-playing OR profiling OR clinical OR healthcare
OR "medical dialogue" OR "educational dialogue" OR jailbreak OR "red teaming" OR
safety)
```

This template was then specialized by section. For instance, the IF-Coding update pass emphasized terms such as "coding assistant," "AI-coding," and benchmark-specific keywords, whereas CE-Healthcare searches more often combined "clinical," "medical consulting," and "medical dialogue." In addition, citation chasing around anchor papers was routinely used to recover near-neighbor work that keyword search alone could miss.

**Inclusion and exclusion criteria.** We included papers when they (1) study multi-turn interaction with LLMs in non-multimodal settings; (2) contribute a benchmark, dataset, method, system study, analysis, or challenge discussion directly relevant to the task-oriented taxonomy or the improvement and challenge sections; and (3) contain enough technical detail in the full paper to support curated placement and discussion. Exclusion was handled through a progressive manual review procedure rather than a single pass. Concretely,

the "three rounds" refer to three explicit decision points for each paper: it was assigned to two authors, each of whom marked it independently as keep or exclude, and the first author then conducted a final corpus-wide check across sections and tables. Papers could be provisionally tagged as out of scope if they were primarily single-turn-only, agent-based, multimodal, embodied or environment-control focused, or otherwise outside the survey boundary. A paper was removed from the core corpus only when the exclusion judgment was maintained through both assigned reviews and the final first-author check; otherwise it remained under discussion until a final placement decision was made.

We also retained a very small number of *contextual non-paper resources* in the original Version 1 corpus and therefore report them separately from peer-reviewed or preprint papers in Figure 5. These items are not treated as research papers or benchmark evidence for the section tables. Instead, they are primary-source contextual documents—such as an official repository, product or research announcement, or public sample-question resource—that were cited only when they were directly needed to describe a released resource, public deployment, or ecosystem reference point relevant to the survey. They were judged by direct relevance and factual specificity rather than by research-paper contribution criteria and were always counted separately from the main paper total.

**Screening, full-text verification, and assignment.** Screening and full-text verification were conducted manually by the authors during corpus construction and retrospective audit. Here, *screening* refers to an initial full-paper pass used to decide whether a candidate was genuinely multi-turn, task-relevant, and within scope, but without yet recording the fine-grained benchmark or table fields used later in the manuscript. *Full-text verification* refers to the deeper follow-up pass in which the paper was examined in more detail—including appendices, released repositories or datasets when relevant, and the benchmark/evaluation details needed for section placement, table construction, and caption-level accuracy checks. In practice, each paper was assigned to two authors for manual review, each of whom made an independent keep/exclude judgment; the first author then performed a final consistency review across the full corpus, section placement, and table evidence. Because the original manuscript was not launched as a preregistered review, we do not claim prospectively logged reviewer-by-reviewer screening tallies; the stage-level counts in the appendix figures are therefore retrospective totals rather than a frozen vote log.

For the original corpus, the recovered exclusion profile was dominated by single-turn-only work (188 items), followed by environment-control or embodied work (64), agent-based work (52), multimodal work (27), and other scope mismatches (22), as reported in Figure 5. These numbers should be read as the *final exclusion reason counts* after the progressive multi-stage review procedure described above, not as counts from a single abstract-only pass. In the second major review round, papers were divided by section, and the section design itself continued to evolve during this process. We relied on different background strengths across the team—including general LLM interaction, healthcare, role-playing, education, and jailbreak—so that a domain-oriented reviewer plus at least one additional teammate checked each section to determine whether a paper was truly relevant and should be covered. Each section and its corresponding tables were then drafted and validated by more than two contributors, followed by a final consistency check by the first author across the entire manuscript. Cross-cutting papers may still be discussed in more than one section, but corpus counts are reported by unique paper rather than section-level mentions.

**Data charting and extracted items.** For each included paper, we charted the information needed to support section placement, table construction, and later consistency auditing. The recurring extracted items included: bibliographic identity; task family and subsection assignment; paper role (e.g., benchmark/-dataset, method/system, analysis, or challenge-focused discussion); benchmark or dataset details when applicable; evaluation setup and judge type; and concise notes on topic, method, findings, and limitations. For benchmark-oriented papers, we additionally charted the fields needed for the section tables, such as dialogue counts, average turns, evaluator type, agreement checks, and evaluation criteria whenever the source paper reported them. These fields were extracted manually from the full papers and then checked again during table drafting before being normalized into the current tables.

**PRISMA-ScR-oriented reporting coverage.** Table 10 summarizes how the main PRISMA-ScR reporting concerns are handled in this survey. We treat PRISMA-ScR as the primary standard because the

Table 10: Compact mapping from key PRISMA-ScR reporting expectations to the documentation provided in this survey.

| PRISMA-ScR reporting item | How handled here |
|---|---|
| Protocol / registration | No preregistered protocol; status stated explicitly in this appendix as part of retrospective reporting. |
| Information sources | Venue-centered browsing, scholarly search tools, citation chasing, and update refreshes are described above, with the most recent update in April 2026. |
| Search strategy | Representative keyword families and a reproducible example search string are reported above; exact venue-specific executions varied by section and revision round. |
| Selection of sources | Progressive manual scope filtering, section-level full-text verification, and final cross-section consistency review are described above. |
| Data charting | Manually charted bibliographic, taxonomic, benchmark, evaluation, and note-taking fields are summarized above. |
| Selection results / flow diagram | The PRISMA-ScR-inspired workflow figures in this appendix report the original corpus-construction process and the later update stage as separate but methodologically aligned stages. |
| Limitations of the review process | Search, scope, and taxonomy-assignment limitations are discussed in the Broader Impact section and in the scope/boundary discussion of the main text. |

present paper is a broad evidence-mapping survey rather than an intervention-focused systematic review, while PRISMA 2020 informs our wording for workflow transparency and limitations reporting.

### A.1 Original Corpus Construction (2022 to April 2025)

The first survey version was assembled through the shared methodology described above, applied to the literature window from 2022 through April 2025. Figure 5 documents this original corpus-construction workflow, including identification from venue-centered and scholarly searches, supplementary GPT-assisted and manual additions, scope filtering, and eligibility decisions. In the retrospective reconstruction of this original phase, we recorded 452 records from venue or database search, 216 from scholarly search, and 37 supplementary additions from GPT Deep Research and manual additions. After 45 records were removed before screening and 12 more were excluded at the screening stage, 611 reports from the database/register branch and 37 reports from the supplementary-discovery branch were assessed for eligibility. The resulting Version 1 corpus contained 275 included items, comprising 271 peer-reviewed or preprint papers and 4 contextual non-paper resources reported separately in the figure. These four contextual resources were the AlpacaEval repository (Li et al., 2023e), the LearnLM report (Wiggers, 2024), the Claude for Education announcement (Anthropic, 2025), and the USMLE sample-question resource (FSMB, 2023).

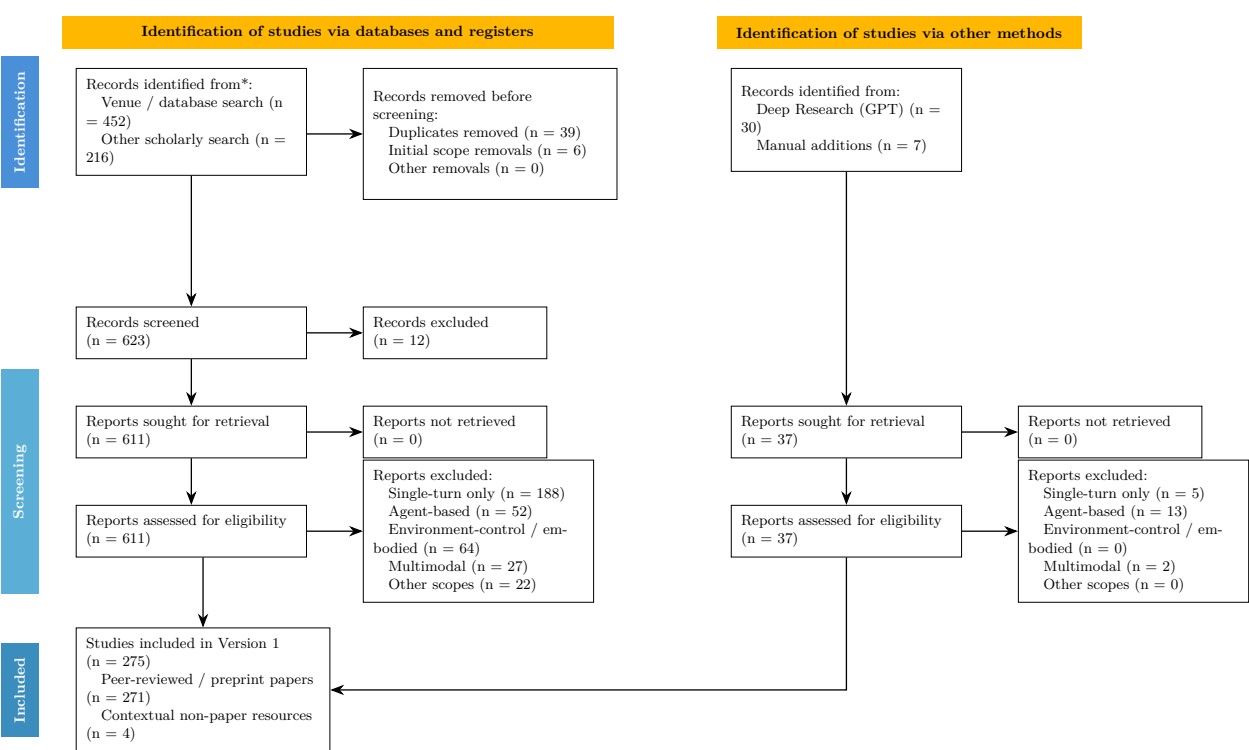

Figure 5: PRISMA-ScR-inspired retrospective flow diagram for the original corpus construction covering the 2022 to April 2025 phase of the survey.

## A.2 Update Corpus Construction (May 2025 to April 2026)

After the April 2025 snapshot, we continued to maintain the survey repository and collect candidate papers using the same search, screening, and boundary-setting procedures. We consolidated this accumulated material into a structured update under those same rules. Figure 6 documents this update-stage workflow and the addition of newly included studies to the Version 2 survey corpus. In the reconstructed update-stage counts, we recorded 110 candidate records from venue or database search, 9 from other scholarly search, 68 from two deep-research passes, and 35 manual additions. These reconstructed channel-level counts led to 128 newly included studies in the update, bringing the Version 2 corpus to 403 studies overall. Because the repository was actively maintained between versions rather than frozen as a prospective review log, the exact fixed anchors are the carried-forward total (275), the newly included total (128), and the final Version 2 corpus size (403), while the source-channel subtotals in Figure 6 should be read as retrospective maintenance counts.

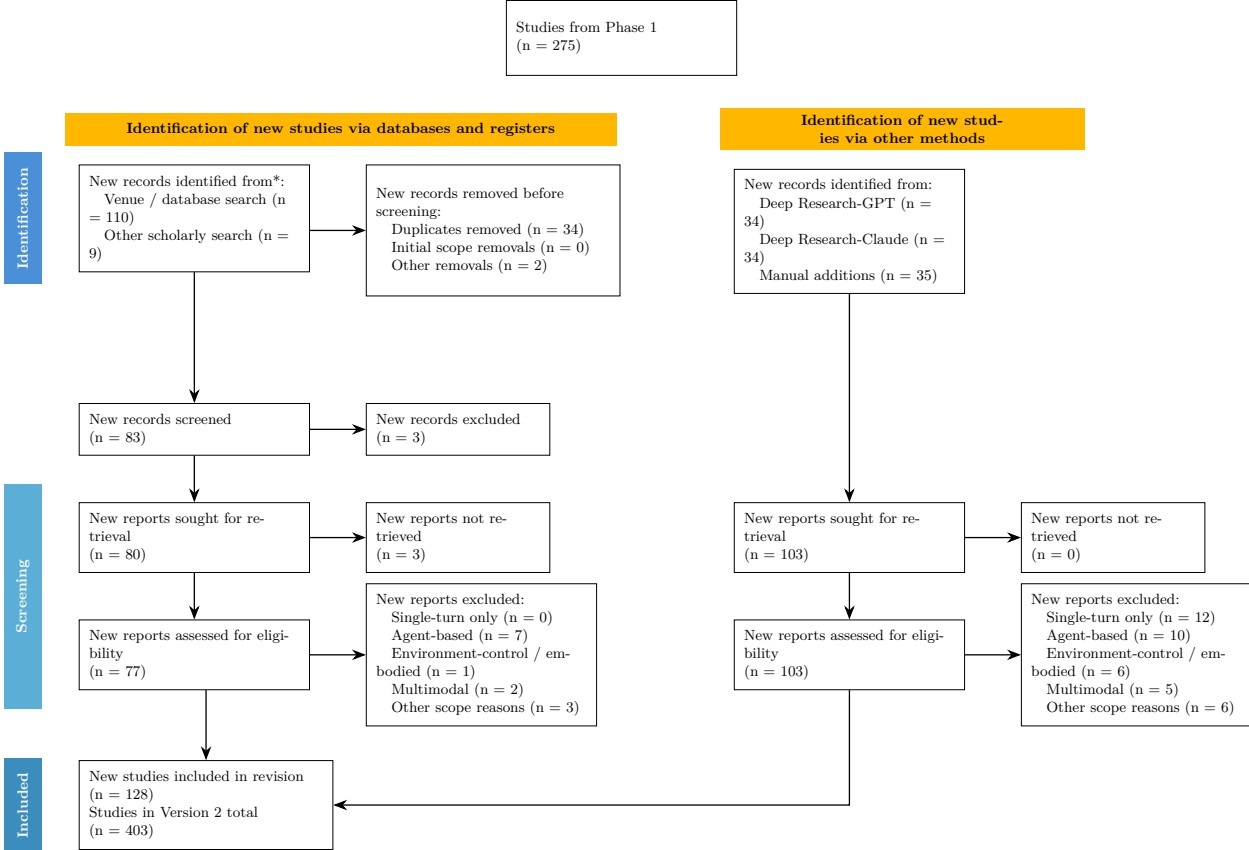

Figure 6: PRISMA-ScR-inspired retrospective flow diagram for the revision update that extended the survey from April 2025 through April 2026 under the same inclusion, exclusion, and boundary-setting rules.

# B   Related Surveys

## B.1   Scope Boundaries and Adjacent Survey Directions

This survey centers on multi-turn interaction with LLMs in non-multimodal settings. To keep the main text focused, we do not attempt exhaustive coverage of several adjacent literatures that overlap with, but are not identical to, our core scope. The boundary should be read together with §2, where we distinguish static dialogue evaluation, interactive multi-turn tasks, and dynamic environment or agent settings.

**Dialogue-system and domain-specific surveys.**   Readers seeking broader dialogue-system coverage may consult the survey of recent LLM-based multi-turn dialogue systems by Yi et al. (2025) and the historical overview by Wang et al. (2023b), which traces the transition from earlier language-model-based dialogue systems to LLM-era settings. For domain-specific conversational systems, healthcare-oriented reviews such as Valizadeh & Parde (2022), Shi et al. (2024c), and the Med-LLM survey of Liu et al. (2024b) provide broader coverage of medical dialogue systems and medical LLM deployment than we aim to include here.

**Capability-oriented and conversational-agent overviews.**   The survey by Zhang et al. (2025a) is the closest existing overview to our topic, but it adopts a more capability-oriented framing of multi-turn interaction. A complementary adjacent perspective is given by Acikgoz et al. (2025), which surveys conversational agents through a capability and challenge lens oriented toward reasoning, monitoring, control, and long-horizon interaction. We cite such works when they sharpen distinctions between capability-level analysis and our more explicitly task-family-oriented taxonomy.

**Agentic LLM systems.**   Agentic systems overlap strongly with multi-turn interaction because they also unfold over trajectories and often rely on memory, planning, and iterative feedback. However, their primary emphasis is usually on action selection in richer environments, tool use, or coordination among multiple agents rather than on plain text dialogue as the central object of study. We therefore discuss such works in the improvement section when they illuminate multi-turn interaction, but do not attempt an exhaustive survey of the agent literature. For broader perspectives, see the general survey of Xi et al. (2023), the scoping design-space perspective of Dhamani & Maher (2023), multi-agent overviews such as Guo et al. (2024) and Han et al. (2024), the methodology-centered synthesis of Luo et al. (2025a), the evaluation-focused survey of Yehudai et al. (2025), and the risk-oriented discussion in Hammond et al. (2025).

**Multimodal multi-turn interaction.**   We also exclude MLLMs from the core scope of the survey. Multi-turn multimodal systems involve different input spaces, task structures, and evaluation protocols, and now support a rapidly expanding benchmark ecosystem. For broader context, readers may consult the general MLLM survey by Yin et al. (2024), the evaluation-focused review by Huang & Zhang (2024), and the agentic multimodal perspective of Durante et al. (2024). Representative benchmarks and frameworks illustrating this adjacent literature include ConvBench (Liu et al., 2024c), MMDU (Liu et al., 2024h), MMMT-IF (Epstein et al., 2024), MMDialog (Feng et al., 2023), TheaterGen (Cheng et al., 2024), and SVBench (Yang et al., 2025g). These works provide important context for future extensions of this survey, but they fall outside the text-only scope adopted here.

## C  Full Task-Family Taxonomy

Table 11: Full task-family-oriented map of the literature discussed in this survey, covering benchmark, method, system, and analysis papers across instruction following (IF) and conversational engagement (CE). Cross-cutting papers are placed in the section where they are discussed most centrally, but may also reappear elsewhere in the prose or benchmark tables when relevant.

| Category | Representative Work |
| --- | --- |
| **Instruction Following (IF)** | |
| IF-General | *Multi-turn benchmarks:* MT-Bench (Zheng et al., 2023); MT-Bench++ (Sun et al., 2024); MT-Bench-101 (Bai et al., 2024); MT-Eval (Kwan et al., 2024); M2Lingual (Maheshwary et al., 2025); Multi-IF (He et al., 2024); FairMT-Bench (Fan et al., 2025); FB-Bench (Li et al., 2025d); StructFlowBench (Li et al., 2025b); TRUEBench (Park et al., 2025); EvolIF (Jia et al., 2025a); IHEval (Zhang et al., 2025f); MT-Consistency (Li et al., 2025e). 
 *Task-specific & proactive:* AQA-Bench (Yang et al., 2025c); WILT (Banatt et al., 2024); WEBLINX (Lù et al., 2024); MULTITURNINSTRUCT (Han et al., 2025a); SysBench (Qin et al., 2025); SAPIENT (Du et al., 2025a); ECR (Zhang et al., 2024b); Clarify When Necessary (Zhang & Choi, 2025); Teaching LMs to Gather Information Proactively (Huang et al., 2025). 
 *Meta-evaluation & analysis:* TURNWISE analysis (Javaji et al., 2025); TURNWISE (Graf et al., 2026); Preference Leakage (Li et al., 2026b); Does Context Matter for LLM Judges? (Xu et al., 2025). 
 *Foundational single-turn references:* BIG-Bench (Srivastava et al., 2023); CSQA (Talmor et al., 2019); MMLU (Hendrycks et al., 2021); GSM8K (Cobbe et al., 2021); IFEval (Zhou et al., 2023); AlpacaEval (Li et al., 2023e). |
| IF-Math | *Math-specific dialogue:* MathChat-Bench / MathChatSync (Liang et al., 2024); MathChat-Agent (Wu et al., 2024b); Zero-shot Debate (Keating, 2024); MathDial (Macina et al., 2023). 
 *Training & reasoning:* M-DPO / M-KTO (Xiong et al., 2025); MINT (Wang et al., 2024f); Beyond Final Answers (Gupta et al., 2025); Chain-of-Thought (Wei et al., 2022); mathematical interactive reasoning (Romera-Paredes et al., 2024); Let's Verify Step by Step (Lightman et al., 2023). 
 *Benchmarks & evaluation:* KMP-Bench (Shi et al., 2026); MRBench (Maurya et al., 2025); Intent Matters (Petukhova & Kochmar, 2025); SBSC (Singh et al., 2025). |

*Table 11 continued from previous page*

| Category | Representative Work |
|---|---|
| IF-Coding | *Interactive coding benchmarks:* InterCode (Yang et al., 2023); What Makes LLMs Reason in (Multi-Turn) Code Generation? (Zheng et al., 2025); PyBench / PyInstruct (Zhang et al., 2024c); When Benchmarks Talk (Pan et al., 2025); CONVCODEWORLD (Han et al., 2025b); CodeFlowBench (Wang et al., 2025a); MultiCodeIF (Duan et al., 2025); MT-Sec (Rawal et al., 2025). 
 *Debugging & steering:* Debug Like a Human (Zhong et al., 2024); CodeSteer (Chen et al., 2025e); OpenCodeInterpreter (Zheng et al., 2024b); CodeAct (Wang et al., 2024e); TreeInstruct / MULTIDEBUG (Kargupta et al., 2024); ClarifyGPT (Mu et al., 2023); From Code to Correctness (Shi et al., 2024d); steering reasoning vs. execution (Chen et al., 2025f). 
 *Domain-specific (SQL & EHR):* MMSQL (Guo et al., 2025b); EHRAgent (Shi et al., 2024a); DySQLBench (Sun et al., 2025). 
 *Foundational code references:* Spider (Yu et al., 2018); MBPP (Austin et al., 2021); NL2Bash (Lin et al., 2018); CodeContests (Li et al., 2022); TACO (Li et al., 2023d); CodeGen (Nijkamp et al., 2023b); CodeGen2 (Nijkamp et al., 2023a). |
| **Conversational Engagement (CE)** | |
| CE-Overview | ABC-Eval (Finch et al., 2023); BotChat (Duan et al., 2024); MultiChallenge (Deshpande et al., 2025); DialogBench (Ou et al., 2024); SimulatorArena (Dou et al., 2025). |
| CE-Roleplay | *Early persona modelling:* persona embeddings (Li et al., 2016); personalized memory networks (Kottur et al., 2017); PersonaChat (Zhang et al., 2018); the survey by Chen et al. (2024h). 
 *LLM-era persona & role-play systems:* PersonaLLM (Jiang et al., 2024a); CharacterChat (Tu et al., 2023); role-play prompting (Kong et al., 2024a); PIPPA (Gosling et al., 2023); UltraChat (Ding et al., 2023); PRODIGy (Occhipinti et al., 2024); ChatHaruhi (Li et al., 2023b); CharacterGLM (Zhou et al., 2024a); RoleCraft-GLM (Tao et al., 2024); Ditto (Lu et al., 2024); CharacterLLM (Shao et al., 2023). 
 *Consistency methods:* PersonaPKT (Han et al., 2023); PPlug (Liu et al., 2025c); Neeko (Yu et al., 2024b); MIDI-Tuning (Wang et al., 2024c); offline RL for persona consistency (Shea & Yu, 2023); COMEDY (Chen et al., 2025d); consistent personas via multi-turn RL (Abdulhai et al., 2025a). 
 *Evaluation & benchmarks:* LaMP (Salemi et al., 2024); CharacterEval (Tu et al., 2024); RoleEval (Shen et al., 2023a); TimeChara (Ahn et al., 2024); LLM Roleplay (Tamoyan et al., 2025); SimulBench (Jia et al., 2025b); InCharacter (Wang et al., 2024g); RoleInteract (Chen et al., 2024e); CROSS (Yuan et al., 2024); SocialBench (Chen et al., 2024d); RoleLLM (Wang et al., 2024j); RAIDEN (Wu et al., 2025a); RMTBench (Xiang et al., 2025); long-conversation degradation (Lu et al., 2025a); CharacterBench (Zhou et al., 2025a); RoleMRC (Lu et al., 2025b); PersonaConvBench (Li et al., 2025c); PERSONAMEM (Jiang et al., 2025); OpenCharacter (Wang et al., 2025c). |

*Continued on next page*

*Table 11 continued from previous page*

| Category | Representative Work |
|---|---|
| CE-Healthcare | *Domain-adapted LLMs:* HuaTuo (Wang et al., 2023a); MING (Liao et al., 2024a); DoctorGLM (Xiong et al., 2023); the review by Mutabazi et al. (2021); broader surveys (Valizadeh & Parde, 2022; Shi et al., 2024c). 
 *Knowledge sources:* USMLE samples (FSMB, 2023); PubMedQA (Jin et al., 2019); MedQA (Jin et al., 2021). 
 *Proactive & interactive diagnosis:* BianQue (Chen et al., 2023); proactive evaluation (Liao et al., 2024b); DISC-MedLLM (Bao et al., 2023); MediQ (Li et al., 2024b); Clinical Camel (Toma et al., 2023); T-Agent (Hu et al., 2024); APP (Zhu et al., 2025a). 
 *Multi-modal & mental health:* BiMediX (Pieri et al., 2024); CPsyCounX (Zhang et al., 2024a); PsycoLLM (Hu et al., 2025b); SMILE (Qiu et al., 2024); PsyQA (Sun et al., 2021). 
 *Chinese medical LLMs:* Zhongjing (Yang et al., 2024); self-instruct (Wang et al., 2023c); CMeKG (Byambasuren et al., 2019); HuaTuoGPT (Zhang et al., 2023); HuaTuoGPT II (Chen et al., 2024g); Aquila-Med (Zhao et al., 2024); Qilin-Med (Ye et al., 2023); ChiMed (Tian et al., 2019); CMExam (Liu et al., 2023). 
 *Consultation evaluation:* AMIE (Tu et al., 2025); MedGPTEval (Xu et al., 2023); automated evaluation (Liao et al., 2023); MMD-Eval (Liu et al., 2025d); MedFuzz (Ness et al., 2024); HealthBench (Arora et al., 2025). 
 *Recent multi-turn medical:* Guo et al. (2026a); medical MINT (Fang et al., 2026); MedDialBench (Luo et al., 2026); CPGBench (Tan et al., 2026); MedMT-Bench (Yang et al., 2026); MedDialogRubrics (Gong et al., 2026); MEDPI (V. et al., 2026); MindEval (Pombal et al., 2025); Mazhar et al. (2026); MedAidDialog (Nigam et al., 2026); Dr. Assistant (Guo et al., 2026b). |
| CE-Education | *Pedagogical agents:* SocraticLM (Liu et al., 2024a); TreeInstruct (Kargupta et al., 2024); StratL (Puech et al., 2025); PACE (Liu et al., 2025a); JeepyTA (Liu et al., 2025e). 
 *Math tutoring:* MathDial (Macina et al., 2023); SocraticLLM / SocraticMATH (Ding et al., 2024); MathTutorBench (Macina et al., 2025); CourseAssist (Feng et al., 2024); algebra-tutor alignment (Levonian et al., 2025); DPO-based tutor training (Scarlatos et al., 2025). 
 *Tutoring benchmarks & platforms:* KMP-Bench (Shi et al., 2026); MRBench (Maurya et al., 2025); TutorBench (Srinivasa et al., 2025); SAFETUTORS (Hazra et al., 2026); EduDial (Wei et al., 2025); TeachLM (Perczel et al., 2025); ConvoLearn (Sharma et al., 2026); GMADE / prompt moderation (Steindl et al., 2025); simulated students (Scarlatos et al., 2026); LearnLM (Wiggers, 2024); Claude for Education (Anthropic, 2025). 
 *Feedback & grading:* Silva & Costa (2025); Jia et al. (2024b); stepwise feedback verification (Daheim et al., 2024); preference-optimized feedback (Scarlatos et al., 2024); iterative revision loops (Nair et al., 2024); deployment studies (Tomova et al., 2024; Jia et al., 2024a; Riazi & Rooshenas, 2025); feedback-type control (Lohr et al., 2025); grading studies (Capdehourat et al., 2025; Flodén, 2025; Kostic et al., 2024). 
 *Student simulation:* Generative Students (Lu & Wang, 2024); personality-aware simulation (Liu et al., 2024g); TeachTune (Jin et al., 2025); SimClass (Zhang et al., 2025e); at-risk student simulation (Li et al., 2025a); lesson-plan simulation (Hu et al., 2025a); Book2Dial (Wang et al., 2024d). |

*Table 11 continued from previous page*

| Category | Representative Work |
|---|---|
| CE-Jailbreak | *Foundational attacks:* GCG-style and AutoDAN-style precursors (Zou et al., 2023; Liu et al., 2024f); other adversarial studies (Carlini et al., 2023; Yu et al., 2023); PAIR (Chao et al., 2025); Johnny / PAP (Zeng et al., 2024a); Crescendo (Russinovich et al., 2025); ActorAttack / self-discovered clue chains (Ren et al., 2024); Latour's actor-network framing (Latour, 2005). 
 *Decomposition & multi-turn exploits:* decomposition studies (Gibbs et al., 2024; Zhou et al., 2024c; Liu et al., 2024d); MR.JA (Wang et al., 2024a); RED QUEEN (Jiang et al., 2024b). 
 *Safety benchmarks:* HarmBench (Mazeika et al., 2024); cipher variants (Handa et al., 2024); HarmfulQ (Shaikh et al., 2023); JailbreakBench (Chao et al., 2024); BeaverTails (Ji et al., 2023); sentence transformers (Ni et al., 2022). 
 *Recent multi-turn jailbreak:* FITD (Weng et al., 2025); Many-Turn Jailbreaking (Yang et al., 2025d); MHJ (Li et al., 2024a); Yang et al. (2025e); SafeDialBench (Cao et al., 2025); RACE (Ying et al., 2025); SEMA (Feng et al., 2026); Tempest (Zhou & Arel, 2025); Siren (Zhao & Zhang, 2025); CoSafe (Yu et al., 2024a). 
 *Defenses & advanced attacks:* Chain-of-Thought defenses (Wei et al., 2022); NeMoGuardrails (Rebedea et al., 2023); Persona Jailbreaking (Sandhan et al., 2026); Echo Chamber (Alobaid et al., 2026); Mastermind (Li et al., 2026d); realistic misuse scenarios (Mai et al., 2026); X-Boundary (Lu et al., 2025c). |

