# OpenReview forum: "Beyond Single-Turn: A Survey on Multi-Turn Interactions with Large Language Models"
_TMLR — Accepted by TMLR_

### Review · Reviewer_6Sai · 2026-03-12

**Summary Of Contributions:**

This survey provides a comprehensive review of the shift from single-turn LLM evaluation to the complex, real-world multi-turn dialogue. By pivoting from a capability-oriented view to a task-oriented taxonomy, the authors offer a framework that better aligns with actual deployment scenarios in domains like healthcare, education, and coding.

**Audience:**

Yes

**Audience Explanation:**

TMLR’s audience is naturally interested in the evolution of LLMs from performance to real-world utility. This paper explicitly addresses a major gap in current research: while LLMs have excelled in single-turn benchmarks, real-world deployment requires multi-turn interaction. Researchers focused on making AI more practical and reliable would find the analysis of sustained dialogue dynamics essential.

**Claims And Evidence:**

Yes

**Claims Explanation:**

The claims are supported by a vast body of evidence, primarily consisting of existing benchmarks, datasets, and cross-disciplinary studies. The evidence is structured across task-oriented domains such as mathematics, coding, and healthcare to demonstrate why multi-turn interactions are more complex than single-turn prompts.

**Requested Changes:**

- The authors exclude MLLMs to maintain focus, which ignores the increasingly multi-modal nature of real-world multi-turn interactions. This should be considered.
- While it identifies "User Perception" as a challenge, the survey lacks a deep dive into long-term human-AI bonding and its ethical implications beyond basic safety.
- The paper acknowledges LLM agents as a distinct but related field and chooses to keep them mostly outside the primary scope, potentially missing critical architectural overlaps where the "model" ends and the "agent" begins.

---

> ### Author Response · Authors · 2026-04-09
>
> Thank you very much for the thoughtful and constructive review. We are grateful for your positive assessment of the paper and for recognizing the value of a task-oriented perspective for multi-turn LLM evaluation.
>
> ---
>
> Regarding the point on MLLMs, we agree this is an important and increasingly relevant direction. However, the intended scope of this survey is specifically $\textbf{multi-turn interaction with LLMs}$, and our goal is to provide a focused task-oriented survey in this setting. We will clarify this scope definition more explicitly in the revision to better explain what is included and excluded.
>
> We also greatly appreciate your suggestions on expanding the discussion of long-term human–AI interaction and ethical implications, as well as further clarifying the relationship between LLMs and agent-based systems. We would be glad to incorporate them in the revision.
>
> We will start working through these suggestions in detail once all reviews are released and revise the paper accordingly. Thank you again for the helpful feedback!

---

> ### Author Response · Authors · 2026-04-20
>
> We thank the reviewer for the constructive feedback. Over the past week, we have revised the manuscript in response to these comments and are pleased to submit the revision. In the revised version, we address the scope and boundary concerns more explicitly:
>
> >RC1:MLLM exclusion.
>
> We now state much more clearly that the survey centers on multi-turn interaction with LLMs in non-multimodal settings. We also added an appendix discussion of adjacent survey directions that explicitly situates multimodal multi-turn interaction as an important but out-of-core-scope neighboring literature, with representative surveys and benchmarks cited there for context.
>
> >RC2: Long-term human-AI bonding / user perception.
>
> We expanded the manuscript's treatment of user perception, dependence, attachment, and related safety concerns, and connected these issues more clearly to the broader challenge and broader-impact discussions rather than leaving them as a brief challenge keyword.
>
> >RC3: Agents vs. core scope.
>
> We clarified the agent boundary in both the main text and the appendix. The revised paper now explains more explicitly that agentic systems overlap with multi-turn interaction through trajectories, memory, planning, and iterative feedback, but are treated as adjacent rather than core when action in richer environments becomes the primary object of study. At the same time, we retain discussion of agent-related methods where they materially inform the improvement section.

---

### Review · Reviewer_HsSW · 2026-04-08

**Summary Of Contributions:**

This paper provides a task-oriented survey of multi-turn interactions with Large Language Models (LLMs), categorizing the field into instruction-following and conversational engagement tasks. It reviews current benchmarks, improvement methodologies (model-centric, external integration, and agent-based), and identifies open challenges.

### Strengths
- The shift from capability-oriented to task-oriented analysis reflects real-world deployment challenges.
- Provides deep dives into specialized high-impact fields like healthcare, where multi-turn proactivity is essential for diagnosis, and education, where Socratic tutoring is a key goal.
- Table 1 provides a valuable chronological overview of the instruction-following landscape.
- Open Challenges Framework: Identifies six critical areas for future research

### Weaknesses
- Sparse Technical Depth in RL for Roleplay: Section 2.2.1 only discusses one primary paper regarding RL (Shea & Yu, 2023). Given the importance of RL, this section feels underemphasized compared to the SFT discussion.
- Limited Benchmark Comparison: Table 1 is restricted to "Instruction Following". The survey would benefit from a similar comparative table for other subsections.

**Audience:**

Yes

**Audience Explanation:**

The findings of this paper are of high relevance to the TMLR audience, which includes researchers in Natural Language Processing (NLP), Human-Computer Interaction (HCI), and AI Safety.

**Claims And Evidence:**

Yes

**Claims Explanation:**

The claims made in this survey are supported by an extensive and systematic review of contemporary literature. The evidence is convincing because it moves beyond generalized LLM capabilities to focus on task-oriented outcomes in specific high-impact domains like healthcare, education, and coding.

**Requested Changes:**

- The survey currently lacks a formal problem formulation for multi-turn interaction. This makes it difficult for readers to differentiate between distinct problem settings (e.g., static dialogue evaluation vs. dynamic environment interaction). Please provide a a mathematical or symbolic representation (e.g., using MDP or State-Space notation) to define the multi-turn interaction for each problem.
- The organization of the sub-sections in Section 2.2 is inconsistent, which hinders the comparative utility of the survey.
Roleplay (2.2.1): Organized by technique (ICL, SFT, RL). Healthcare/Education (2.2.2–2.2.3): Organized by application sub-domains. To provide a unified benefit to readers, please reorganize these sections to follow a consistent internal structure.
- Clarify Interactive Terminology: Formally define what constitutes an interactive task versus a static dialogue task. The distinction between an interactive task and a static dialogue is currently blurred. In many RL and Agent literature, "interactive" implies the agent changes the state of an environment.
- Expand RL Literature in Conversational Tasks: Conduct a deeper literature search for RL applications in Roleplay or Education beyond the few currently cited.
- Expanded Benchmark Tables like Table 1.

---

> ### Author Response · Authors · 2026-04-09
> **Response to Reviewer HsSW**
>
> We sincerely thank the reviewer for the thoughtful and constructive feedback, as well as for the positive assessment of the paper’s relevance and contribution. We are encouraged that the task-oriented framing and the discussion of high-impact domains were found valuable.
>
> ---
>
> We appreciate the reviewer’s concrete and thoughtful suggestions regarding:
> (1) adding a clearer formal problem formulation for multi-turn interaction,
> (2) improving the consistency of the organization in Section 2.2,
> (3) clarifying the distinction between interactive tasks and static dialogue tasks,
> (4) expanding the discussion of RL literature in conversational settings, and
> (5) enriching the benchmark comparisons with additional summary tables.
>
> These are all very helpful suggestions, and we will carefully incorporate them in the revision. We will begin working through these points in detail as soon as all reviews are released, and will revise the paper accordingly.
>
> We thank the reviewer again for the constructive feedback!

---

> ### Author Response · Authors · 2026-04-20
>
> We thank the reviewer for the positive assessment of the task-oriented framing and for the concrete suggestions. Over the past week, we have revised the manuscript in response to these comments and are pleased to submit the revision. The revised version directly addresses the main structural requests.
>
> >RC1: Formal problem formulation.
>
> We added a Background section with a compact symbolic formulation for multi-turn interaction, together with explicit discussion of dialogue history, evolving state, and task trajectories. This was added precisely to separate multi-turn interaction from static evaluation setups.
>
> >RC2: Interactive vs. static dialogue terminology.
>
> We now define this distinction explicitly in both prose and figure form. A new early figure contrasts single-turn with multi-turn interaction, and the background text clarifies the difference between static dialogue evaluation, interactive text-based tasks, and richer environment/agent settings.
>
> >RC3: Section 2.2 organization.
>
> Rather than forcing all CE subsections into one artificial template, we designed it on purpose and now clarified the organizing principle in the revised CE introduction. The role-play literature is still most naturally organized by technical approach, while healthcare, education, and jailbreak are better organized by application domain. The revised text now makes that design choice explicit so readers are not left to infer it.
>
> >RC4: RL literature in conversational tasks.
>
> We have expanded the citation pool and added 12 new works on roleplay and 9 on education. In the task-oriented section, these additions make the application and CE task discussions more comprehensive. We also provide a deeper treatment of RL in the improvement section, specifically under model-centric approaches, where we cover both single-turn and multi-turn RL across domains.
>
> >RC5: Expanded benchmark tables.
>
> We greatly appreciate this suggestion, which significantly improves the presentation of the paper. We have addressed this request carefully and directly. The revised manuscript now includes section-specific benchmark and dataset tables across all major IF and CE subdomains, rather than concentrating benchmark comparisons in a single instruction-following table.

---

### Review · Reviewer_BNSo · 2026-04-10

**Summary Of Contributions:**

The manuscript provides a literature review on the topic of multi-turn interactions with large language models (LLMs), with a focus on specific tasks. Its stated main contributions are organising the literature on the topic of multi-turn interaction with LLMs in terms of tasks addressed, categorising and providing details on the methods identified in the literature (in terms of source of improvements), and discussing open challenges in this field.

Strengths:
- Provides a literature survey on a timely topic, since multi-turn interactions are the de facto way people interact with LLMs
- Covers many different aspects relevant to multi-turn interactions with LLMs, including method characteristics that drives improvements

Weaknesses:
- The authors propose to conduct a literature review, but do not provide basic details about how this review is structured. Please see detailed comments below.
- One does not clearly know what is new in this review and what is not and has been covered in previous reviews (in terms of covered papers, for example)
- Survey focus, organisation, and coherence needs a lot of improvement. The survey claims to be focusing on tasks, but Table 1 (the only Table in the review) does not mention tasks. Figure 1 is the main figure in the paper with a taxonomy organising the works, which also does not mention tasks. See detailed comments below.

**Audience:**

Yes

**Audience Explanation:**

The topic of the literature review is clearly on topic and I believe most TMLR readers would benefit from learning about multi-turn interactions with LLMs.

**Broader Impact Concerns:**

There is no Broader Impact Statement, and I believe this should be included. In Section 4.5 "Ethical and Safety" some ethics-related aspects are discussed, but authors should discuss the ethical implications of their work rather than of the works of others. For example, are there biases in the authors inclusion / exclusion / search criteria? Please integrate these aspects (ethical implications of your work and the works of others).

**Claims And Evidence:**

No

**Claims Explanation:**

I believe this submission fails at conforming with the basics of what a literature review article requires. The authors propose to conduct a literature review, but do not provide basic details about how this review is structured. I strongly recommend using guidelines for surveys such as PRISMA for systematic reviews [1] and PRISMA-ScR for scoping reviews [2]. Even if the authors decide that they are not doing neither a systematic nor a scoping review (and therefore are simply doing a narrative review), I urge the authors to use PRISMA to report the search strategy (databases searched, keywords used, number of records removed), screening strategy (how many authors screen title/abstract/full text), and inclusion of studies. Please use the 4-phase PRISMA flow diagram and consider adapting the 27-item PRISMA 2020 checklist to increase transparency [1,2].

You should also clearly differentiate your work from previous works. Right now that is blurry, even if one reads the previous surveys and tries to actively do that. For example, authors state the closest existing survey to theirs is the pre-print by Zhang et al. (2025). In Zhang et al. (2025)'s Figure 1, they provide a taxonomy to organise the retrieved articles by "capability". In the submitted manuscript's Figure 1 (in page 22), we have a figure with similar purpose, the main difference being that this organisation is done by "type of improvement". However, there are subparts of these two taxonomies that should match but don't. For example, "External integration approaches" focusing on memory / RAG in this manuscript, and "Context Memory / External" in Zhang et al. (2025). However, there are zero papers in the intersection between these two groups. In fact, among all the papers appearing "Context Memory / External" in Zhang et al. (2025), I found none anywhere in the taxonomy of this manuscript's Figure 1. If we don't know how you select papers to include in your review, unfortunately there is no way we can judge the merits of this survey.

[1] https://www.prisma-statement.org/
[2] https://www.prisma-statement.org/scoping

**Requested Changes:**

- Use PRISMA / PRISMA-ScR checklists to structure your literature review.
- Clearly define the important terms and concepts used in your taxonomy. You could create a Background section for this purpose, or unambiguously define any terms/concepts early in the section where they appear.
- You frame your survey's novelty in the introduction as focusing on tasks, and not capabilities (as in Zhang et al., 2025). However, your Figure 1 does not cover tasks, but improvements. I suggest either having two separate figures, Fig.1 with a taxonomy for tasks and Fig.2 for a taxonomy for improvements, or adding the tasks as an extra dimension in the taxonomy in Fig.1.
- Your Table 1, the only Table in the manuscript mentioned early in Section 2, lists the recent benchmark/datasets and evaluation methods, but you do not position your review as surveying datasets / evaluation methods. It is odd to see the only table in the manuscript providing details about many things, but not the tasks directly which in principle is your manuscript's focus. Why is that Table 1 relevant for your survey? I suggest focusing on tasks directly in Table 1.
- Explain how this taxonomy in Figure 1 works: should it be seen as a decision rule, meaning that for a paper, one traverses the taxonomy from root to leaf, so that a paper can only appear in one leaf node in your taxonomy? Or are there multiple dimensions in the taxonomy, and a single paper can (or must?) in fact traverse different branches of the taxonomy (e.g., it seems that "Model-Centric Approaches / In-Context Learning" and "External Integration Approaches / RAG" are not exclusive options, but on the contrary are complementary). Please clarify all these details and restructure your taxonomy as necessary.
- Authors state the closest existing survey to theirs is the pre-print by Zhang et al. (2025). In Zhang et al. (2025)'s Figure 1, they provide a taxonomy to organise the retrieved articles by capability. In the submitted manuscript's Figure 1 (in page 22), we have a figure with similar purpose, the main difference being that this organisation is done by type of improvement. However, there are subparts of these two taxonomies that should match. For example, "External integration approaches" focusing on memory / RAG in this manuscript, and "Context Memory / External" in Zhang et al. (2025). However, there are zero papers in the intersection between these two groups. In fact, among all the papers appearing "Context Memory / External" in Zhang et al. (2025), I found none anywhere in the taxonomy of this manuscript's Figure 1. Please clarify these issues.

---

> ### Author Response · Authors · 2026-04-10
> **Response to Reviewer BNSo**
>
> We thank the reviewer for the detailed and constructive feedback. We are encouraged that the reviewer finds the topic timely and valuable for the TMLR audience, and we appreciate the recognition of the breadth of aspects covered in our survey.
>
> We agree that the current manuscript lacks sufficient methodological transparency. In the revision, we will add a dedicated Literature Review Methodology section following PRISMA/PRISMA-ScR principles, including the search strategy, data sources, inclusion/exclusion criteria, and screening process. We will also include a PRISMA checklist in the appendix to improve clarity and reproducibility.
>
> We also acknowledge the need to better position our work with respect to prior surveys. We will strengthen the comparison with Zhang et al. (2025) and other related works, explicitly clarifying both the overlaps and the key differences in taxonomy design and paper coverage.
>
> For Table 1 concern, we would like to clarify that Table 1 is currently designed to summarize datasets and evaluation protocols specifically for the task of $\textbf{instruction following}$. Based on all reviewers' comment on this point, in the revision, we will:
> (i) explicitly state that Table 1 focuses on instruction following,
> (ii) extend this structured summarization to other conversational tasks discussed in the paper, and
> (iii) improve the presentation to make the task dimension more explicit.
>
> Finally, we will improve clarity by defining key concepts more explicitly (e.g., memory, RAG, in-context learning) and add a broader impact discussion, including potential biases in our survey methodology and limitations of our coverage.

---

> > ### Comment · Reviewer_BNSo · 2026-04-14
> >
> > Dear authors, thank you for your quick reply. I wanted to note that the concerns I have are rather substantial, and I would like to / expect to see at least one revision of the manuscript to be able to judge its merits.
> >
> > Kind regards,
> > Reviewer BNSo.

---

> > > ### Author Response · Authors · 2026-04-14
> > >
> > > Dear Reviewer BNSo,
> > >
> > > Thank you for your follow-up. We fully understand that the concerns raised are substantial and that a concrete revision is needed to properly address them.
> > >
> > > We are currently working on a revised manuscript that incorporates the changes outlined in our previous response including the PRISMA-based methodology section, restructured taxonomy and tables, clearer differentiation from prior surveys, and a broader impact discussion. We look forward to submitting the revised version for your evaluation.
> > >
> > > Best regards,
> > > Authors

---

> ### Author Response · Authors · 2026-04-20
>
> We sincerely thank the reviewer for the detailed and constructive feedback. Over the past week, we have carefully revised the manuscript in response to these comments and are pleased to submit the revised version. The revision directly addresses the methodological, organizational, and positioning concerns raised above.
>
> >RC1: Review methodology / PRISMA transparency.
>
> We added a paragraph in the Intro section (Review Methodology) for the process transparency plus a dedicated Appendix A, *Review Methodology Details*, which now reports protocol status, search channels, a representative search query, inclusion/exclusion criteria, screening and assignment procedures, data-charting fields, a compact PRISMA-ScR mapping table, and two PRISMA-ScR-inspired retrospective flow diagrams. We explicitly frame this as a retrospective transparency supplement rather than a preregistered systematic review.
>
> >RC2: Definition of key terms and concepts.
>
> We added a Background section that now defines the core terminology used throughout the paper, including multi-turn interaction, interactive task, static dialogue evaluation, memory/context notions, and utterance-level terminology. We also added a compact symbolic formalization to distinguish interaction trajectories from static dialogue settings. plus an illustration (Fig.1) for the demonstration.
>
> >RC3: Task taxonomy vs. improvement taxonomy.
>
> We separated these more cleanly in the revision. The main text now includes a dedicated task-taxonomy figure for the two core task families, instruction following (IF) and conversational engagement (CE), while the improvement taxonomy is presented separately later in the paper. This directly resolves the earlier mismatch between the task-oriented framing and the figure structure.
>
> >RC4: Role of Table 1 and benchmark coverage.
>
> The earlier table design in V1 was too narrow and did not match the task-oriented framing. In the revision, the manuscript no longer relies on a single benchmark/evaluation table to carry the survey. Instead, benchmark and dataset coverage is distributed into section-specific audit tables across IF-General, IF-Math, IF-Coding, CE-Overview, CE-Roleplay, CE-Healthcare, CE-Education, and CE-Jailbreak.
>
> >RC5: How the taxonomy should be read.
>
> We clarified the distinction between the task taxonomy and the improvement taxonomy. The task taxonomy is used to organize the literature by deployment-level task family and subsection; the improvement taxonomy is not intended as a mutually exclusive decision tree for papers, and we now make this distinction clearer in the text.
>
> >RC6: Positioning relative to prior surveys
>
> We strengthened the related-survey comparison in two places: (i) the introduction now includes a direct comparison table against closely related surveys, and (ii) the appendix adds a dedicated section on related surveys and adjacent survey directions. We now make explicit that Zhang et al. (2025) is the closest prior survey but is capability-oriented, whereas our revision is task-oriented and couples that task framing with harmonized benchmark tables and a transparent retrospective methodology supplement. We no longer imply a paper-by-paper one-to-one correspondence between their capability clusters and our improvement clusters.
>
> > Broader impact and limitations.
>
> We added a Broader Impact section and also discuss methodological limitations in Appendix A. This now includes explicit acknowledgment of retrospective reconstruction, scope boundaries, and possible biases introduced by inclusion/exclusion rules and taxonomy assignment decisions.

---

> ### Comment · Reviewer_BNSo · 2026-04-20
> **Comments on revision (1/2)**
>
> - RC1-RC6 and Broader Impact Statement: Thank you for these updates, they considerably improved the manuscript.
>
> - RC1: Review methodology / PRISMA transparency
>     - I thank the authors for this, I find it easier now to gauge the contributions of the manuscript. I have still a few clarification points.
>         - The main part that is still not clear to me is the exact procedure authors used to exclude papers deemed out of topic for this review. (1) In App.A, "Inclusion and exclusion criteria" you state that out of scope papers were "marked during review", and the paper would be excluded only if it had been "marked for exclusion in three rounds". (2) You later mention in "Screening, full-text verification and assignment" that papers were first filtered for the "original corpus" including numbers of filtered out papers (e.g. for single-turn-only work). (3) Then you mention a "second major review round" whereby two people checked the relevance of each paper, and at least two others double-checked the Tables in which this paper appears.
>         - Could you please clarify? It's unclear what it means to mark a paper "during review" or what are the "three rounds" (in 1). You also mention a number of papers filtered out by type (e.g., single-turn-only work , 188 items). It is unclear how these were filtered out, e.g., by how many authors? How? (in 2). Finally, (3) is the part that looks like a standard review practice, whereby you have multiple authors screening the same paper and disagreements are discussed.
>         - You also don't explain what you mean by screening: do you mean reading the title + abstract only? Or do you mean something else?
>         - Later in A.1 you mention "4 contextual non-paper resources reported separately in the figure". Are these associated with a technical report of any kind? It is unclear what you mean by "non-paper resources". If they are "non-paper", how do you judge them?
>
> - RC3: Task taxonomy vs. improvement taxonomy.
>     - Authors state their focus is task-based, which differs from previous work, and propose to taxonomise works as "instruction following" and "conversational engagement" tasks. For me, a task is question answering, or text summarisation, or information extraction, etc. I'd suggest a small change in wording, since these do not seem to be tasks themselves, but rather types of tasks. I think the manuscript would read much better if authors referred to these as "task families" or "task types", for instance in the header of Section 3 and elsewhere. Later, you already say that, e.g., "By focusing on well-defined task families such as (…)", but I suggest aligning that everywhere in the manuscript (and referring to "task families" or something similar, rather than "tasks" only in terms of what you address in your review). Even if you go one step further and look at the next level of your taxonomy, you are still not talking about tasks but about types of tasks (e.g., "General", "Math", "Role-play"). Or even further, in "Conversational engagement" you clearly mix "tasks types" with domain (e.g., "healthcare" or "education"). For clarity and precision in your message, I strongly suggest finding better terms for your taxonomy, since even only mentioning "task types" would be misleading since "healthcare" or "education" are not task types.
>     - In Fig.3, you are only showing each paper in its "main branch". I suggest showing papers everywhere they apply, clearly differentiating between the papers appearing in that section's via its "main branch" and appearing in that section for another reason. Practically speaking, you could simply have two rows in each blue box ("main", "other") and explain  both in the Figure caption.
>     - "In multi-turn conversations, this mechanism can help LLMs maintain coherent and contextually rich dialogue histories by providing a persistent, efficient memory of earlier interactions, thereby enhancing the model’s ability to reference and build upon past conversational turns." -> This is ambiguous. Do you mean it can help as in "if it were applied to conversational tasks, it could help" or "according to the experiments the authors conducted including conversational tasks, we see it can help"?
>     - In 4.2.2, you say RAG is introduced by Lewis and colleagues in 2020. Then you mention the Wizard of Wikipedia method, introduced in 2018. How is that RAG, if RAG was "invented" later? Or did they do RAG "before it was invented" somehow? A discussion here would help.

---

> > ### Comment · Reviewer_BNSo · 2026-04-20
> > **Comments on revision (2/2)**
> >
> > - RC6: Positioning relative to prior surveys
> >     - I still believe the positioning relative to previous surveys need to be strengthened. For instance, in relation to Zhang et al. (2025), see my previous comments/criticisms: as an example, why is there no overlap between "External integration approaches" focusing on memory / RAG in your manuscript and "Context Memory / External" in Zhang et al. (2025)? Is this an issue with Zhang et al. (2025) or with your taxonomy? Is that because years covered in both papers do not overlap? Because of the definition of integration / memory in each paper? I would like to see these points more clearly discussed. If this is an issue with Zhang et al. (2025), that is more evidence pointing towards the need of a better conducted review on this topic (i.e., yours). A question I find very important that is not yet addressed: how much overlap is there with Zhang et al. (2025) in terms of surveyed papers (in general, or per section)?

---

> > > ### Author Response · Authors · 2026-04-20
> > >
> > > We agree with the reviewer that this comparison needed to be made more explicit. The main issue is that the two surveys are organized around different primary axes. Zhang et al. (2025) is **capability-oriented**, whereas our revision separates (i) a **task-family oriented** taxonomy, (ii) a distinct following improvement section, and (iii) a dedicated open-challenges section. As a result, their capability buckets are not expected to map one-to-one onto our task-family or improvement branches, and the overlap is redistributed rather than concentrated in aligned categories.
> > >
> > > To make this more concrete, we examined the overlap more systematically. Our full revised survey cites 403 unique references overall, but because Zhang et al. do not organize the literature through our domain-centered CE branches, a direct comparison against all 403 items would be misleading. Excluding the 171 works whose main placement in our survey is in CE leaves a non-CE comparison pool of 232 items. Against the full 157-reference bibliography of Zhang et al. (2025), exact title-level matching yields 33 overlaps, corresponding to 14.2% of our non-CE pool and 21.0% of Zhang et al.’s full citation set. These overlaps are concentrated mainly in instruction-following and method/improvement-oriented literature, including works such as InterCode, WILT, MathChat, Parrot, WEBLINX, and several method papers such as Reflexion and CodeSteer.
> > >
> > > We also compared against the papers explicitly listed in Zhang et al.’s taxonomy trees rather than their full bibliography. Under this stricter like-for-like comparison, we find 15 exact title-level overlaps. This lower number is expected, because Zhang’s taxonomy trees emphasize capability-organized exemplars, while our survey allocates many papers to task-family branches, to a separate improvement taxonomy, or to the open-challenges discussion.
> > >
> > > This also clarifies the reviewer’s specific example. The lack of overlap between our “External Integration Approaches” and Zhang et al.’s narrower “Context Memory / External” subgroup is not primarily a year-window effect. Even when we compare that slice against our Version 1 manuscript, the exact overlap remains zero. The main reason is taxonomic. Zhang et al.’s subgroup is a narrow capability-memory slice focused on external memory mechanisms, whereas our external-integration branch groups memory augmentation together with conversational RAG and graph integration under a broader intervention-oriented category. We therefore do not interpret the zero-overlap in that slice as evidence that one survey is simply missing the other’s papers; rather, it reflects a mismatch in the organizing units themselves.
> > >
> > > Finally,  Zhang et al. do not develop a comparably explicit improvement taxonomy, and they do not include a dedicated open-challenges section of the kind we provide here. We believe this is the main reason why the overlap is partial rather than high, and why the two surveys should be understood as complementary rather than redundant.

---

> > > > ### Comment · Reviewer_BNSo · 2026-04-21
> > > >
> > > > I thank the authors for the detailed response, that really helps understand the differences.
> > > >
> > > > Just for reference, when I suggested in Figure 3 to include papers no only in their "main dimension" but also other applicable dimensions, one reason is that that would make it easier to compare "overlap" with previous works (since one can now easily visually check these references, "main" and "other", and compare to those from other papers). However, it is of course perfectly fine to keep Figure 3 as is if the authors believe that is best for their paper.

---

> > > > > ### Author Response · Authors · 2026-04-21
> > > > >
> > > > > We appreciate the reviewer’s continued engagement, and the discussion has been very helpful in clarifying these distinctions.
> > > > >
> > > > > We thank the reviewer for this helpful clarification and understand the motivation behind the suggestion. We agree that showing both “main” and “other” dimensions could make overlap with prior surveys easier to inspect. We believe both choices have value; in our case, we kept Figure 3 as is because repeated entries made it more crowded, and we instead clarify cross-branch applicability in the caption and surrounding text. We appreciate the suggestion and the reviewer’s flexibility on this point.

---

> ### Author Response · Authors · 2026-04-20
>
> Thank you for this positive feedback and for noting that the revisions have considerably improved the manuscript. We appreciate the opportunity to further clarify the review workflow.
>
> >RC1. Review methodology / PRISMA transparency
>
> To make it clear, we revised Appendix A to clarify the actual review procedure. In our workflow, the “three rounds” do not refer to three abstract-only screening passes. Rather, each paper was assigned to two authors, each of whom made an independent keep/exclude judgment, and the first author then conducted a final corpus-wide consistency review across sections and tables.
>
> We also clarified the distinction between screening and full-text verification. Screening was not a title/abstract-only pass; instead, it was an initial full-paper pass used to decide whether a candidate was genuinely multi-turn, task-relevant, and within scope, but without yet recording the detailed benchmark and table fields. Full-text verification refers to the deeper follow-up pass in which the paper was examined in more detail, including appendices, released repositories or datasets when relevant, and the benchmark/evaluation details needed for section placement, table construction, and caption-level accuracy checks. Accordingly, the exclusion counts reported by type (e.g., single-turn-only work) should be interpreted as final exclusion-reason totals after this progressive manual review process, rather than counts from a single abstract filter.
>
> We further clarified that the four contextual non-paper resources are primary-source contextual documents retained only when directly needed to describe a released resource, public deployment, or ecosystem reference point; they are counted separately from peer-reviewed/preprint papers.
> - AlpacaEval repository
> - LearnLM report / announcement
> - Claude for Education announcement
> - USMLE question resource
>
> > RC3. Task taxonomy vs. improvement taxonomy.
>
> We thank the reviewer for this precision-oriented comment and agree that “task families” is a better term than “tasks” for our taxonomy. We therefore revised the manuscript to use “task family” / “task families” consistently in the main framing of Section 3, the introduction, and the conclusion. We also added explicit clarification that the next level of the taxonomy is intentionally heterogeneous: in the instruction-following family it is divided into subfamilies such as general-purpose, mathematical, and coding settings, whereas in the conversational-engagement family it mixes interaction settings (e.g., role-play, jailbreak) with application domains (e.g., healthcare, education), because that is how the literature clusters in practice.
>
> Regarding Figure 3, we agree that the improvement taxonomy is not a mutually exclusive partition and that several papers could reasonably appear under more than one branch. In the revised manuscript, we therefore clarify explicitly that Figure 3 should be read as an organizing lens rather than as a one-paper-one-category ontology. Both the section-opening text and the figure caption now state that some papers span multiple branches, but that each work is placed under the branch most central to its primary contribution. We considered duplicating papers across all applicable branches, but decided against doing so in the figure itself because it would substantially reduce readability and make the taxonomy visually cluttered. Instead, we keep Figure 3 compact and use the surrounding prose to acknowledge cross-branch relevance where needed.
>
> Besides, we revised the sentence in the improvements section about memory mechanisms to remove the ambiguity identified by the reviewer. The text now states that these mechanisms suggest a plausible path for helping conversational systems when incorporated into them, rather than implying that every cited study directly demonstrated gains on conversational tasks.
>
> Finally, regarding the RAG description, we agree that our earlier wording could misleadingly suggest that Wizard of Wikipedia (2018) was already using the later RAG terminology introduced by Lewis et al. (2020). We therefore revised Section 4.2.2 to distinguish more carefully between (i) the now-standard RAG formulation popularized by Lewis et al. (2020), and (ii) earlier retrieval-grounded dialogue systems that implemented the same broad idea of retrieving external knowledge to support response generation before the term “RAG” became standard. In the revised text, Wizard of Wikipedia is now described as a retrieval-grounded precursor to later conversational RAG systems, rather than as an instance of the later terminology stated without qualification.

---

> > ### Comment · Reviewer_BNSo · 2026-04-21
> >
> > I again thank the reviewers for the detailed response. I would just like to clarify one remaining point that is still not clear to the reviewer. Authors say "(...) the four contextual non-paper resources are primary-source contextual documents retained only when directly needed to describe a released resource, public deployment, or ecosystem reference point (...)". I understand a released resource, but I don't really know what the authors mean by "public deployment" or "ecosystem reference point". Could you please provide a concrete example of each?

---

> > > ### Author Response · Authors · 2026-04-21
> > >
> > > We thank the reviewer for pointing out that this wording was too abstract. In our usage:
> > >
> > >
> > > >By a public deployment, we mean an officially announced public-facing system or product deployment that is relevant to the application setting discussed in the survey.
> > >
> > > - A concrete example is Anthropic’s “Claude for Education” announcement, which we cite only to document the existence of that publicly launched educational deployment.
> > > - ChatGPT Health(https://openai.com/index/introducing-chatgpt-health/)
> > >
> > > >By an ecosystem reference point, we mean an official non-paper document that anchors a widely used evaluation or application context, but is not itself treated as a technical contribution.
> > >
> > > - A concrete example is the USMLE sample-question resource from FSMB/NBME, which we cite only as an official reference set for the medical exam context discussed in the healthcare section.
> > > - Patient Handbooks from health centers
> > >
> > > Hope the explanation solve your concerns.

---

### Author Response · Authors · 2026-04-15

Dear Reviewers and AC,

Thank you very much for your patience and for your thoughtful feedback on our submission!

We would like to share a brief update that we are currently revising the paper carefully based on all the comments we received. Our goal is to address all of your concerns as thoroughly as possible, and to further improve the paper beyond the specific points raised in the reviews.

We truly appreciate your time and patience throughout this process. We expect to complete the revision by this weekend.

Best regards,
Authors

---

### Author Response · Authors · 2026-07-02

Dear Action Editor and Reviewers,

Thank you for the acceptance decision and the Survey Certification, and for your thoughtful engagement throughout the review process. We've submitted the camera ready version of the submission!

We are especially grateful for the depth and constructiveness of the feedback: Reviewer BNSo's guidance on PRISMA-ScR methodological transparency and positioning relative to prior surveys, Reviewer HsSW's suggestions on formal problem formulation, section organization, and expanded benchmark tables, and Reviewer 6Sai's points on scope boundaries and long-term human–AI interaction. These discussions substantially strengthened the survey, and many of the resulting changes — the review methodology appendix, the task-family taxonomy, and the section-specific benchmark audit tables — are now among the parts of the paper we value most.

Thank you again for your time and care. We hope the survey serves the community well, and we will continue to maintain the companion repository as the field evolves.

Best regards,

The Authors

---

### Decision · Action_Editor_ecLt · 2026-06-04

**Recommendation:** Accept as is

**Additional Comments:**

The reviewers raised a few key concerns with the paper as originally submitted. These included questions about how the literature review was structured and the paper's relationship with existing surveys, insufficient focus on tasks despite the paper's claims to the contrary, insufficient discussion of connections to reinforcement learning (RL), and limited benchmark comparisons.

The authors made a concerted effort to address these issues through discussions with the reviewers and revisions to the paper. The reviewers agree that this effort addressed their concerns and significantly strengthened the paper.

**Audience:**

Yes

**Audience Explanation:**

As all three reviewers emphasize, the topic of multi-turn interactions with LLMs is of significant interest to people across the broader TMLR community. Further, the reviewers appreciate the practical/real-world relevance of the discussion.

**Claims And Evidence:**

Yes

**Claims Explanation:**

The paper provides a survey of the current state of multi-turn interactions with large language models (LLMs). There is consensus among the reviewers that the survey in its current form provides a thorough and systematic review of existing literature that is clear in how the literature review is structured and how it is situated in the context of related survey articles.